# Multidecadal trend analysis of in-situ aerosol radiative properties around the world

Martine Collaud Coen[1], Elisabeth Andrews[2,3], Andrés Alastuey[4], Todor Petkov Arsov[5], John Backman[6], Benjamin T. Brem[7], Nicolas Bukowiecki[8], Cédric Couret[9], Konstantinos Eleftheriadis[10], Harald Flentje[11], Markus Fiebig[12], Martin Gysel-Beer[7], Jenny L. Hand[13], András Hoffer[14], Rakesh Hooda[6,15], Christoph Hueglin[16], Warren Joubert[17], Melita Keywood[18], Jeong Eun Kim[19], Sang-Woo Kim[20], Casper Labuschagne[17], Neng-Huei Lin[21], Yong Lin[12], Cathrine Lund Myhre[12], Krista Luoma[22], Hassan Lyamani[23,24], Angela Marinoni[25], Olga L. Mayol-Bracero[26], Nikos Mihalopoulos[27], Marco Pandolfi[4], Natalia Prats[28], Anthony J. Prenni[29], Jean-Philippe Putaud[30], Ludwig Ries[9], Fabienne Reisen[18], Karine Sellegri[31], Sangeeta Sharma[32], Patrick Sheridan[3], James Patrick Sherman[33], Junying Sun[34], Gloria Titos[23,24], Elvis Torres[26], Thomas Tuch[35], Rolf Weller[36], Alfred Wiedensohler[35], Paul Zieger[37,38], Paolo Laj [39,40,41]

[1] Federal Office of Meteorology and Climatology, MeteoSwiss, Payerne, Switzerland
[2] Cooperative Institute for Research in Environmental Sciences, University of Colorado, Boulder, CO, USA
[3] NOAA/Global Monitoring Laboratory, Boulder, CO, USA
[4] Institute of Environmental Assessment and Water Research (IDAEA), Spanish Research Council (CSIC), Barcelona, Spain
[5] Institute for Nuclear Research and Nuclear Energy, Bulgarian Academy of Sciences, Sofia, Bulgaria
[6] Atmospheric composition research, Finnish Meteorological Institute, Helsinki, Finland
[7] Laboratory of Atmospheric Chemistry, Paul Scherrer Institute, Villigen PSI, Switzerland
[8] Atmospheric Sciences, Department of Environmental Sciences, University of Basel, Basel, Switzerland
[9] German Environment Agency (UBA), Zugspitze, Germany.
[10] Institute of Nuclear and Radiological Science & Technology, Energy & Safety N.C.S.R. "Demokritos", Attiki, Greece
[11] German Weather Service, Meteorological Observatory Hohenpeissenberg, Hohenpeißenberg, Germany
[12] NILU – Norwegian Institute for Air Research, Kjeller, Norway
[13] Cooperative Institute for Research in the Atmosphere (CIRA), Colorado State University, Fort Collins, CO, USA
[14] MTA-PE Air Chemistry Research Group, Veszprém, Hungary
[15] The Energy and Resources Institute, IHC, Lodhi Road, New Delhi, India
[16] Empa, Swiss Federal Laboratories for Materials Science and Technology, Duebendorf, Switzerland
[17] South African Weather Service, Research Department, Stellenbosch, South Africa
[18] CSIRO Oceans and Atmosphere, PMB1 Aspendale VIC, Australia
[19] Environmental Meteorology Research Division, National Institute of Meteorological Sciences, Seogwipo, Korea
[20] School of Earth and Environmental Sciences, Seoul National University, Seoul, Korea
[21] Department of Atmospheric Sciences, National Central University, Taoyuan, Taiwan
[22] Institute for Atmospheric and Earth System Research, University of Helsinki, Helsinki, Finland
[23] Andalusian Institute for Earth System Research, IISTA-CEAMA, University of Granada, Junta de Andalucía, Granada, Spain
[24] Department of Applied Physics, University of Granada, Granada, Spain
[25] Institute of Atmospheric Sciences and Climate, National Research Council of Italy, Bologna, Italy
[26] University of Puerto Rico, Rio Piedras Campus, San Juan, Puerto Rico
[27] Environmental Chemistry Processes Laboratory, Department of Chemistry, University of Crete, Heraklion, Greece.
[28] Izaña Atmospheric Research Center, State Meteorological Agency (AEMET), Tenerife, Spain
[29] National Park Service, Air Resources Division, Lakewood, CO, USA
[30] European Commission, Joint Research Centre (JRC), Ispra, Italy

[31] Université Clermont Auvergne, CNRS, Laboratoire de Météorologie Physique (LaMP), Clermont-Ferrand, France

[32] Climate Chemistry Measurements Research, Climate Research Division, Environment and Climate Change Canada, Toronto, Canada

[33] Department of Physics and Astronomy, Appalachian State University, Boone, NC USA

[34] State Key Laboratory of Severe Weather & Key Laboratory of Atmospheric Chemistry of CMA, Chinese Academy of Meteorological Sciences, Beijing, China

[35] Leibniz Institute for Tropospheric Research (TROPOS), Leipzig, Germany

[36] Glaciology Department, Alfred-Wegener-Institut Helmholtz Zentrum für Polar- und Meeresforschung, Bremerhaven, Germany

[37] Department of Environmental Science and Analytical Chemistry, Stockholm University, Stockholm, Sweden

[38] Bolin Centre for Climate Research, Stockholm University, Stockholm, Sweden

[39] Univ. Grenoble Alpes, CNRS, IRD, Grenoble-INP, IGE, 38000 Grenoble, France

[40] CNR-ISAC, National Research Council of Italy – Institute of Atmospheric Sciences and Climate, Bologna, Italy

[41] University of Helsinki, Atmospheric Science division, Helsinki, Finland

**Correspondence**: Martine Collaud Coen (martine.collaudcoen@meteoswiss.ch)

## Abstract

In order to assess the evolution of aerosol parameters affecting climate change, a long-term trend analysis of aerosol optical properties was performed on time series from 52 stations situated across five continents. The time series of measured scattering, backscattering and absorption coefficients as well as the derived single scattering albedo, backscattering fraction, scattering and absorption Ångström exponents covered at least 10 years and up to 40 years for some stations. The non-parametric seasonal Mann-Kendall (MK) statistical test associated with several prewhitening methods and with the Sen's slope was used as main trend analysis method. Comparisons with General Least Mean Square associated with Autoregressive Bootstrap (GLS/ARB) and with standard Least Mean Square analysis (LMS) enabled confirmation of the detected MK statistically significant trends and the assessment of advantages and limitations of each method. Currently, scattering and backscattering coefficient trends are mostly decreasing in Europe and North America and are not statistically significant in Asia, while polar stations exhibit a mix of increasing and decreasing trends. A few increasing trends are also found at some stations in North America and Australia. Absorption coefficient time series also exhibit primarily decreasing trends. For single scattering albedo, 52% of the sites exhibit statistically significant positive trends, mostly in Asia, Eastern/Northern Europe and Arctic, 18% of sites exhibit statistically significant negative trends, mostly in central Europe and central North America, while the remaining 30% of sites have trends, which are not statistically significant. In addition to evaluating trends for the overall time series, the evolution of the trends in sequential 10 year segments was also analyzed. For scattering and backscattering, statistically significant increasing 10 year trends are primarily found for earlier periods (10 year trends ending in 2010-2015) for polar stations and Mauna Loa. For most of the stations, the present-day statistically significant decreasing 10 year trends of the single scattering albedo were preceded by not statistically significant and statistically significant increasing 10 year trends. The effect of air pollution abatement policies in continental North America is very obvious in the 10 year trends of the scattering coefficient – there is a shift to statistically significant negative trends in 2010-2011 for all stations in the eastern and central US. This long-term trend analysis of aerosol radiative properties with a broad spatial coverage provides insight into potential aerosol effects on climate changes.

## 1. Introduction

Climate change has been considered as a premier global problem in the scientific community for decades. Thirty years ago, the community organized to produce the first Intergovernmental Panel on Climate Change (IPCC) report (IPCC, 1990) about the state of scientific, technical and socio-economic knowledge on climate change, its impacts and future risks, and options for reducing the rate at which climate change was taking place. Aerosols have been recognized as an important active climate forcing agent since the 1970's and, in the last IPCC report (IPCC, 2013), the impact of aerosols on the atmosphere was still considered as one of the most significant and uncertain aspects of climate change projections and, for the first time, decadal trend analysis of in-situ aerosol optical properties around the world was reported.

Aerosol optical properties are the relevant parameters that determine the radiative forcing of particulate matter. While some of these optical properties are currently measured by satellite (Choi et al., 2019), airborne and ground-based remote sensing (REM) technologies (https://aeronet.gsfc.nasa.gov/, www.earlinet.org), the ground-based, in-situ measurements represent some of the longest time series, allowing assessment of the long-term time evolution of aerosol radiative properties in the lower troposphere.

The first in-situ measurement network began in the mid 1970's at several remote locations (Bodhaine et al., 1995). Through national, international programs and/or on individual organisation's initiatives, the number of stations with systematic aerosol monitoring activities in regional background locations has continued to increase since the 1990's. As of 2017 absorption has been measured for at least 1 year (y) at 50 sites, for 5 y at 37 sites and for 10 y at 20 sites, while scattering has been measured for at least 1 y at 56 sites, for 5 y at 45 sites and for 10 y at 30 sites. The companion paper (Laj et al., in review 2020) provides a historical view and a complete description of the present networks for aerosol measurements. The longest datasets cover up to 40 y of measurements (BRW (40 y), SPO (40 y), and MLO (31 y) (see Table 1 for station's acronyms), whereas some stations with long time series recently closed or moved (THD, SGP, MUK, CPT). The spatial and temporal variability of aerosol properties is extremely high due to the short lifetime of aerosol particles (on the order of days to weeks), the wide variety of sources, as well as the chemical and microphysical processing occurring in the atmosphere; a dense network of stations is consequently required to obtain a global view of aerosol changes. The growing number of stations with long-term (>10 y) time series of aerosol particles optical properties - 24 in 2010 (Collaud Coen et al., 2013, hereafter referred to as CC2013) and now 52 in 2016-2018 - is a positive factor. Detracting from that growth is the continued lack of sites in South America, Africa, Oceania and Asia.

Long-term measurements are the only possible approach for detecting change in atmospheric composition resulting from either changes in natural or anthropogenic emissions and/or changes in atmospheric processes and sinks. However, detecting long-term trends of aerosol optical properties remains a challenge, due to their high natural variability, uncertainties caused by changes and biases in measurement methodology, the ill-defined statistical distribution of the parameters, the presence of high autocorrelation in aerosol parameters, as well as the occasional issues regarding traceability of historic operating procedures. Trend analysis can only be performed on time series without breakpoints or on homogenized time series that account for changes in measurement conditions (e.g. relocations, instrument calibration/repair/upgrades, inlet changes) (CC2013). Once homogenized data sets are available, appropriate techniques must be used to identify potential trends. The trend analysis methodology must take into account the non-normal distribution of most aerosol parameters, the high autocorrelation of the parameters, and the presence of gaps and negatives in the datasets.

In this current analysis, a considerable effort was made to detect time series break points, to find explanations for them in the logbooks and station history and, if possible, to correct or homogenize the time series. These homogenized time series were then subjected to an array of statistical tests to identify trends. These tests include: (1) the non-parametric seasonal Mann-Kendall test (thereafter referred to as the MK test) associated with the Sen's slope. The applied MK test is however applied with a new pre-whitening method (Collaud Coen et al., in review 2020), (2) a

Generalised Least Squares (GLS) method associated with a Monte-Carlo bootstrap algorithm and (3) the Least-Mean Squares fit (LMS). While the MK test with pre-whitening was considered the most robust method, the other tests were included to allow a comparison between various simple and frequently used methods.

The first long-term trend analyses of aerosol optical properties, number concentration and particle size distribution (CC2013 and Asmi et al., 2013) covered 2001-2010 as the shortest period and longer periods if data were available. The main observations were: (1) a general statistically significant (ss) -at 95% confidence level- decrease of number concentration, scattering and absorption coefficients in North America, (2) a ss decrease of number concentration in
northeastern Europe, (3) no ss trends in central Europe for any of the parameters and (4) no ss scattering coefficient trends but increasing 10 y absorption coefficient and number concentration trends in polar regions. These trends were related to the decrease in anthropogenic primary aerosol emissions and in precursors of secondary aerosol formation. The high altitude station Mauna Loa (MLO) in the Pacific was unique in exhibiting increasing optical properties trends that
were mostly attributed to long-range transport from Asia. The results in CC2013 are in line with the 1996-2013 trend analysis at the BND and SGP stations in North America (Sherman et al., 2015) showing decreasing scattering coefficient and sub-micron scattering fraction and increasing backscattering fraction. More recently, Pandolfi et al. (2018) presented the long-term trends of in-situ surface aerosol particle optical properties (scattering) measured in Europe until 2015. The ss
decreasing trends of aerosol particle scattering observed in Europe at around 40% of the stations (mostly in Nordic and Baltic countries and southwestern Europe) were attributed to the implementation of continental to local emission mitigation strategies. Pandolfi et al. (2018) also reported that the scattering Ångström exponent decreased at around 20% for the European stations included in their study (at remote Nordic and Baltic locations and at two mountain sites
in central and eastern Europe), whereas an increase was observed at 15% of the stations (one urban site in southwestern Europe and one in central Europe). In the same study, the backscattering fraction was observed to increase. Trends in horizontal visibility synoptic observations over 1929-2013 from 4000 stations over the US, Europe and Asia (Li et al., 2016) generally agreed with extinction coefficient trends with a significant decrease in all regions but
with different evolutions of the trends. Hand et al. (2014) also found a significant drop of the ambient light extinction coefficient at all IMPROVE (Interagency Monitoring of Protected Visual Environment, http://vista.cira.colostate.edu/Improve/) stations over the 1990s through 2011 with a larger decrease in the eastern US. To our knowledge, no further trend analyses of surface in-situ aerosol optical properties involving a network of stations or several stations have been
published up to now.

This study is part of the SARGAN (in-Situ AeRosol GAW observing Network) initiative (see companion paper Laj et al. (in review, 2020)) with the objective of supporting a global aerosol monitoring network to become a GCOS (Global Climate Observing System) associated network. This trend analysis is intended to answer the following questions:

1) Are there homogeneous long-term trends in in-situ aerosol optical properties over the covered regions of the world? Do they differ as a function of the length of the data series? How do the trends evolve with time?

2) Are there regional similarities or differences in the observed trends among stations? Are there similarities or differences in trends among aerosol parameters at a regional and
continental scale?

3) How do the observed optical property trends compare with trends in other aerosol and gaseous properties reported in the literature?

The results of this study provide the best representation of change in surface aerosol optical properties considering the available in-situ aerosol optical properties datasets and highlight the
possible side effects of air pollution control policies on radiative forcing.

## 2. Experimental

### 2.1 Measurement sites

The long-term trend analysis presented in this study analyzes in-situ aerosol time series from 52 observatories worldwide shown in Fig. 1 with site information listed in Table 1. The network, which
is a subset of the station network described in Laj et al. (in review 2020), comprises 16 stations in Europe, 21 stations in North America, 5 in Asia, 2 in Africa, 6 in the polar regions and 2 in the southwest Pacific. The stations included in this study are primarily located in rural or remote areas and are expected to exhibit regional to large scale representativeness (e.g., Wang et al., 2018). Apart from MUK, all the stations are regional or global GAW (Global Atmospheric Watch,
https://gawsis.meteoswiss.ch/GAWSIS//index.html#/) sites or IMPROVE stations. The GAW aerosol data are archived at and available from the World Data Centre for Aerosol (WDCA, http://www.gaw-wdca.org) located at the Norwegian Institute for Air Research (NILU). The WDCA data repository is the database EBAS (http://ebas.nilu.no), an e-infrastructure shared with other frameworks targeting atmospheric aerosol properties, such as the Co-operative Programme for
Monitoring and Evaluation of the Long-range Transmission of Air pollutants in Europe (EMEP) and the European Aerosols, Clouds, and Trace gases Research InfraStructure Network (ACTRIS). The IMPROVE data are available from the IMPROVE website (http://vista.cira.colostate.edu/improve/Data/data.htm), and from the WDCA. To ensure that the long-term trends analysis was performed on homogeneous time series, a substantial effort of
quality control, rupture detection and homogenisation (see Sect. 2.4) was performed in close collaboration with each station's PI on the data. As has been noted in previous papers, it is critical to have outside review of data to improve the quality of long-term time series (CC2013, Asmi et al., 2013). The final time series used in this analysis are available from following DOI: https://doi.org/10.21336/c4dy-yw57.

The stations' environments were classified into four types (continental, coastal, mountain, or polar) that are represented by 22, 8, 16 and 7 time series, respectively. The type of measured aerosol at each site is further characterized by their footprints comprising 6 types (rural background, forest, desert, (sub)-urban, pristine and mixed). While the environments of Europe, North America and polar regions are fairly well represented, the number of long-term stations in
the rest of the world is currently quite low, resulting in a lack of information from the largest deserts (e.g. Sahara, Gobi, Australian, Arabian, Atacama), from many mountain ranges (e.g. Himalaya, Andes, Southern Great Escarpment, Great Dividiing Range, Ural) and from whole continents (South America (no site), Africa (one island in the Atlantic and one coastal site), and Australia (one coastal site)). Some stations from these underrepresented areas currently have 4 to 7 y of
measurements available and will potentially be used for trend analyses in the future (see table in Laj et al. (in review, 2020)).

Sites were chosen based on the following criteria: (1) availability of at least 10 y of continuous data (two sites with 9 y and one site with 8 y of data for at least one parameter have also been included to improve spatial coverage (CPT, EGB and GSN, respectively)), (2) continuous
measurements without ruptures in the aerosol light scattering and/or absorption measurement; (3) submission of quality-assured data to the WMO WDCA data repository; (4) responsiveness of site operators to questions concerning data quality and homogeneity.

The longest time series with 40 y of measurements are the Arctic and Antarctic stations of BRW and SPO, followed by the high altitude MLO station (31 y). During the 1990's NOAA began
extending their network (Andrews et al., 2019), the IMPROVE network installed numerous stations in the USA (Malm et al., 1994), and the first long-term measurements in Europe, JFJ (Bukowiecki et al., 2016) and HPB, began in 1995. To have the largest representativity and to minimize the number of stations with less than 10 y of measurement, the current long-term trends were computed from time series ending in 2016, 2017 or 2018 (whichever year was most recently
available). To obtain an overview of the long term trend evolution in the past 40 y, all stations with at least 10 y of measurements were considered (see results Sect. 3.2).

## 2.2 Instruments

The relevant instruments operating at each site are listed in Table 1 and further instrument details are given in supplemental material (Table S1). Some particular instrumental features that could influence the trend analysis or comparison between stations are briefly discussed below.

Nephelometers measure aerosol light scattering over a truncated angular range (Müller et al., 2009 and references therein) leading to non-idealities often called "truncation error". The truncation adjustment accounts for scattering over the angles outside the measurement range and non-ideality of the light source. All TSI nephelometer scattering and backscattering sets were adjusted for truncation and instrument non-idealities using the Anderson and Ogren (1998)
correction. Thus, for times when enhanced amounts of large diameter (Dp>1 µm) particles are present, the measured scattering will be lower than true scattering by a substantial amount since the truncation correction increases with particle size (Anderson and Ogren, 1998, Molenar et al., 1997). The Radiance Research nephelometer has similar truncation characteristics as the TSI nephelometer (Müller et al., 2009). The Optec nephelometer measures over a wider angular
range (Molenar, 1997) than the other nephelometers and, like the Radiance Research measurements, the scattering has not been corrected for truncation in this study. The Optec nephelometers measure at ambient conditions with no size cut (they are open air instruments) so they can sample the very large particles present due to both hygroscopic growth at high humidities and/or the occurrence of precipitation, fog, dust, pollen, etc. The Ecotech nephelometers have a
similar angular range as the TSI nephelometers, and the measurements are corrected for truncation errors using the Müller correction (Müller et. al., 2011b), adapted from the Anderson and Ogren correction.

For better comparability of aerosol properties amongst sites and to minimize the confounding effects of water associated with the aerosol, GAW recommends drying the sample air to RH<40%
(WMO/GAW report 227, 2016). While most of the nephelometer scattering time series are accompanied by sample RH measurements, this was not the case for all stations and for the entire measurement period. The calculated RH trends are therefore not always complete. Many breakpoints were detected in sample RH data and exchanges with the individual station PIs revealed that humidity sensors often suffer from artefacts, offsets, and modifications that were
not considered as problematic. These sensor problems were often not resolved due to the secondary status of this housekeeping diagnostic leading to problematic time series. Nonetheless, apart from the IMPROVE network, the majority of nephelometers appeared to have sampled at RH<40%. The IMPROVE scattering measurements were analyzed at the measurement conditions with some constraints on acceptable scattering values, although the
IMPROVE network recommends screening the data when RH>90% (Prenni et al., 2019).  For this study and according to CC2013,  the IMPROVE scattering coefficient was restricted to $\sigma_{sp}$ values lower than 500 Mm$^{-1}$ for stations in the eastern USA (ACA, GSM, MCN and SHN) and lower than 100 Mm$^{-1}$ for stations in the western USA to minimize the influence of rain, fog, snow and ice. These screening constraints minimized the issues associated with high RH but do not correspond
to a screening based on RH.

Measurement of the absorption coefficient was always performed by some type of filter-based photometer but relied on a variety of instruments. These instruments include: Multi-Angle Absorption Photometers (MAAP), Particle Soot Absorption Photometers (PSAP) and Continuous Light Absorption Photometers (CLAP), as well as various models of the Aethalometer (AE16,
AE21, AE31 and AE33). All these instruments suffer from various artefacts, from which the loading effect can influence the wavelength dependence. However, the largest uncertainty in filter-based photometer measurements lies in the effect of the multiple scattering of light into the filter matrix leading to over-prediction of absorption aerosol (e.g., Bond et al., 1999; Lack et al., 2008; Müller et al., 2011a; Collaud Coen et al., 2010, Bernardoni et al., 2019). This artefact is roughly
corrected by the multiple scattering constant $C_{ref}$, and is probably the largest for the Aethalometer and the smallest for the MAAP.

The ACTRIS community has suggested that Level 2 AE31 data submitted to EBAS utilize a multiple scattering constant $C_{ref}=3.5$; most of the analyzed AE31 time series were corrected with

this new rule. The AE33 adds a simultaneous measurement of the light transmission through a second filter spot sampling the same air at a different flow rate associated with a real-time compensation algorithm. This two spots technique allows for correction of the filter loading artefact. This improvement, however, has no effect on the largest artefact (multiple scattering artefact) and, as of yet, there is no agreed upon correction for the AE33 by the aerosol community. Previous AE models used a white light diode (AE10 and AE16) and a $C_{ref}= 1.6$ is usually applied. At FKL, the AE21 used a $C_{ref}=1.8$ and the AE33 a $C_{ref}= 3.0$. The various versions of the Aethalometer require then different corrections, whereas the real $C_{ref}$ value depends on the filter and on the aerosol type. For background rural aerosol, the real $C_{ref}$ value is between 2.5 and 4.5 (Collaud Coen et al., 2010, Bernardoni et al., 2019), the Asian plume has a relatively high $C_{ref}$ between 4 and 5.5 (Kim et al., 2018), in the Arctic $C_{ref}$ is suggest to be 3.45 (Backman et al., 2017), whereas pure mineral dust leads to lower $C_{ref}$ of 1.75-2.56 (Di Biagio et al., 2017).

The MAAP measures not only the light transmission through the filter but also the light backscattered at two different angles. This design takes into account the scattering and multiple scattering artefacts (see Collaud Coen et al., 2010), which are two of the most significant artefacts for filter-based absorption photometers so that no correction is needed ($C_{ref}=1$). The MAAP measured absorption coefficient is consequently more reliable.

The CLAP was developed by NOAA as a replacement for the PSAP (Ogren et al., 2017). The CLAP was designed to have the same optical characteristics as the PSAP so that either the Bond et al. (1999) correction along with the Ogren (2010) update for wavelength and spot size correction or the Virkkula et al. (2005, 2010) corrections can be applied to account for scattering artifacts at multiple wavelengths as well other instrument non idealities (e.g., filter loading artefacts, variability in spot size and flow calibrations). These correction algorithms rely on co-located scattering measurements from a nephelometer and may have issues in the presence of large, primarily scattering aerosol such as sea salt or dust (e.g., Bond et al., 1999) and also may not work well when organic aerosol is abundant (e.g., Lack et al., 2008).

The differences in instrumentation, measurement conditions and post-processing data treatment do not allow the absolute values of aerosol optical parameters for all sites to be compared; however, because there was consistency of data treatment for each individual time series, the trends across the different sites can be compared.

### 2.3 Aerosol optical properties

The data used in this paper consist of hourly-averaged, quality-checked, spectral light scattering ($\sigma_{sp}$), backscattering ($\sigma_{bsp}$) and absorption ($\sigma_{ap}$) measurements. The quality checks correspond to the Level 2 requirements of EBAS (Laj et al., in review 2020). After further visual quality control by the authors, the hourly data were aggregated into daily medians with the requirement that at least 25% of the daily data be valid. The median was chosen to minimize the effect of extreme values on the average since the measured parameters are strongly not normally distributed and most of the calculated parameters also do not follow a normal distribution. Such a low requirement for data coverage was chosen since 6 hourly measurements a day corresponds to half of the potential data coverage at many of the NOAA stations, where the operation mode consists of alternating between PM1 and PM10 size cutoff on a sub-hourly basis (Andrews et al., 2019).

All the nephelometers and the multi-wavelength absorption photometers measure at a green wavelength (~525-550 nm), which is the channel for which the parameters are reported. For the AE31 and AE33 models, the 520 nm channel was chosen. At several sites, the light absorption was measured by white light (~840-880 nm) Aethalometers (AE16), two channel Aethalometers (AE21) using 370 nm and 880 nm or by MAAPs (Multi-Angle Absorption Photometer) at 637 nm (Müller et al., 2011a), requiring the use of another wavelength, typically a red wavelength. In some cases, the blue or red wavelength was preferred due to inhomogeneities or gaps in the green data. Since the trend analysis is not sensitive to the multiplication by a constant, the data series used to determine scattering and absorption trends were not adjusted to 550 nm.

In addition to the measured parameters, the following parameters were computed when the appropriate measurements were available:

- backscatter fraction, $b = \sigma_{bsp} / \sigma_{sp}$

- scattering Ångström exponent, $\mathring{a}_{sp} = -\ln(\sigma_{sp,1}/\sigma_{sp,2})/\ln(\lambda_1/ \lambda_2)$

- absorption Ångström exponent, $\mathring{a}_{ap} = -\ln(\sigma_{ap,1}/\sigma_{ap,2})/\ln(\lambda_1/ \lambda_2)$ or by a linear fit between the logarithm of the 7 absorption coefficients as a function of the logarithm of the 7 wavelengths of the Aethalometer (AE31 and AE33).

- single scattering albedo, $\omega_0 = \sigma_{sp}/(\sigma_{sp}+\sigma_{ap})$

where $\sigma_{sp,i}$ is the scattering coefficient at wavelength i, $\lambda_i$ is the wavelength i, $\sigma_{bsp}$ is the hemispheric backscattering coefficient, and $\sigma_{ap}$ is the absorption coefficient.

$\mathring{a}_{sp}$ and $\mathring{a}_{ap}$ were usually computed from the blue (~450 nm) and green wavelengths, because the red channel of the nephelometers was frequently less stable and more prone to rupture in the time series due to calibrations or instrument changes. However, in some cases, other wavelength pairs were used to utilise the longest time series. $\mathring{a}_{ap}$ computed from the AE31 and AE33 is always more homogeneous if fitted on the 7 wavelengths, so that the fitted $\mathring{a}_{ap}$ was always chosen for these two instruments.

The single scattering albedo was computed from $\sigma_{sp}$ and $\sigma_{ap}$ after $\sigma_{ap}$ was adjusted to match the nephelometer green wavelength with an assumed absorption Ångström exponent of one (i.e., $1/\lambda$ dependence). In order to maintain similar data treatment for absorption instruments with single or multiple wavelengths, the measured absorption Ångström exponents were not used for the wavelength adjustment for the $\omega_0$ calculation.

It should be recalled that all parameters calculated using ratios of the $\sigma_{sp}$, $\sigma_{bsp}$ and/or $\sigma_{ap}$ may have higher uncertainties for two reasons: (1) the ratio of two similar values has a larger uncertainty than the $\sigma_{sp}$, $\sigma_{bsp}$ or $\sigma_{ap}$ uncertainties and (2) the $\sigma_{sp}$ difference between the wavelengths depends on the nephelometer calibration that is performed independently for each wavelength. These uncertainties are particularly enhanced for clean locations with low aerosol loading.

### 2.4 Discontinuities, data consistency and homogenisation

Long-term climate analyses require homogeneous time series to be accurate. A homogeneous climate time series is defined as one where variations are caused only by variations in weather and climate (Conrad and Pollak, 1950) and in emissions of aerosol particles and their precursor gases. Long-term climatological time series can be affected by a number of non-climatic factors called breakpoints (e.g. relocation, instrument upgrades, inlet changes, calibrations, nearby pollution sources) that mask the real climate variations. The breakpoints can be detected either by subjective visual inspection or by objective statistical methods (Peterson et al., 1998, Beaulieu et al., 2007) and must correspond to an event recorded in logbooks describing the station/instrumental history. Many statistical methods are only suitable for normally distributed data and cannot therefore be applied to aerosol optical properties measurement without data transformation (Lindau and Venema, 2018). Moreover, they are often applied not only to the data but to ratios or differences between various time series that are not systematically available at all the measuring sites of this study.

Visual inspection was used to detect breakpoints and to assess the validity of the time series to be used for climatic trend analysis. For this study, each measured and calculated (see Sect. 2.3) parameter at all wavelengths, as well as all the possible ratios between measured parameters (including the number concentration if available), at each station were visually inspected in linear and logarithmic time series plots. The treatment of minimum and maximum values, of outliers and negatives along with the consistency of seasonal cycles were looked at closely when inspecting the time series plots. In addition, the data owners responded to a questionnaire about

potential breakpoints, providing metadata that could be used to confirm/dismiss possible break points or to accurately locate them. The identified breakpoints were discussed with the data owners leading to corrections, homogenisation, invalidations or splitting of the time series into two parts. In one case (absorption data from SUM measured by AE16 and CLAP), the two time series were homogenised by multiplying the AE16 data by the median of the ratio between both data sets during the 10.5 months of simultaneous measurements. Only data sets considered as homogeneous by the authors and the data owners were analyzed in this study.

In the older networks, several modifications likely lead to inhomogeneities that occurred at sites in the network around the same time. Some of these include:

1) Two of the longest running NOAA stations changed their TSP (Total Suspended Particles) inlets for PM10 size cuts in the middle of the multi-decade time series (MLO: 2000, BRW: 1997).  Some other stations outside the NOAA network also modified the measurement size cuts over their long-term measurement period. Usually this change of size cut (TSP to PM10) did not generate a breakpoint for aerosol optical properties so that the time series could be considered as homogeneous. A differentiation between periods of sampling in- or outside of clouds was not made, even though TSP and PM10 could respond differently in these situations. In contrast, the modification of TSP or PM10 size cuts to PM2.5 or PM1 cutoffs usually led to visible breakpoints. PAL is the only station where changes between PM10, PM5 and PM2.5 did not induce a visually obvious breakpoint, likely due to the minimal presence of supermicron particles at this site.

2) The NOAA stations used the single green wavelength PSAP until the years 2005-2007 when they replaced them with a three wavelength (3w) PSAP (see Table S1). This instrumental change usually did not induce a visually obvious breakpoint.

3) A further instrument change for the absorption coefficient at NOAA sites occurred in 2013-2015 through the introduction of the 3w CLAP. The 3w PSAP to 3w CLAP change usually induced no breakpoint in the green absorption coefficient. The red channel sometimes exhibited a visible breakpoint (APP and BND), resulting in breakpoints in the absorption Ångström exponent. In those cases, calculation of the absorption Ångström exponent with the blue and green channels was preferred.

4) The long time series from MLO and JFJ were subject to the removal of negative values during the first years of measurements until 2000 and 1999, respectively. The raw data prior to these years were not archived by the data providers for either site. This change in minimal values does not seem to produce a clear breakpoint in the sense that the computed trends were not affected strongly enough to modify the climatic trends.

To compare long-term trends between stations from various networks, instruments and operators, instrumentation, measurement conditions and data treatment consistency is critical, but some lenience amongst stations was deemed acceptable. Specifically, some of the discretion was allowed including whether the data sets had the same corrections applied (e.g., truncation or not), how the sites dealt with sample RH and very low aerosol amounts, and inlet size cuts. Table 1 includes columns indicating information about the size cuts and RH conditions at the various sites. No screening or analysis as a function of cloud amount/clear sky conditions were done since these criteria/flagging were not available at all stations. Below, the impact of sample RH, size cut and of general instrument conditions and corrections on trend evaluation are briefly discussed:

1) Humidity: One important factor affecting all aerosol measurements is the relative humidity (RH) at which the measurements are made. For $\sigma_{sp}$, measurements at controlled RH enable minimization of the confounding effects of aerosol hygroscopic growth resulting in increases in the amount of scattering aerosol (Nessler et al., 2005, Fierz-Schmidhauser et al., 2010; Burgos et al., 2019). The disadvantage of making measurements at low RH is that aerosol hygroscopic properties must be measured or assumed in order to adjust the aerosol optical properties to ambient conditions. As noted above (see Sect 2.2), within the GAW program,

recommendations have been given to measure $\sigma_{sp}$ at low (RH<40%) humidities. Apart from the IMPROVE and CPR nephelometers, the instruments typically operated at RH < 50%, with only six stations having a RH 95[th] percentile value larger than 50% (AMY, CMN, EGB, GSN, IPR and SGP) but with a median clearly much lower than 50%. In contrast, the IMPROVE network instruments measure at near ambient conditions (Malm et al., 1996). The scattering restriction method (see Sect 2.2) was chosen in order to maintain the highest data coverage - simply removing scattering values associated with RH>50% from the ambient IMPROVE data set would have eliminated most of the summertime measurements, particularly for the eastern USA locations. For all stations with some contribution of scattering made at RH values larger than 50%, the dry scattering and backscattering coefficients were calculated by removing values corresponding to hourly RH median >50%.

Ensuring a low humidity in the nephelometer reduces but does not suppress the potential influence of the hygroscopic growth on nephelometer measurements (Zieger et al., 2013). Therefore, if RH data were available, the RH long-term trends were also computed and their potential effect on the trend of $\sigma_{sp}$, $\sigma_{bsp}$, b and $\mathring{a}_{sp}$ was evaluated (see Sect 4.1).

The filter-based absorption photometers are also sensitive to rapid RH changes (e.g., Anderson et al., 2003), but daily absorption averages are usually not biased by such rapid fluctuations (Bernardoni et al., 2019). Very high sample RH could lead to higher uncertainties but absorption measurements at GAW stations are usually connected to inlets with some sort of conditioning intended to reduce sample RH (e.g. diffusion or membrane dryers, dilution with dry air and in some cases heating). Additionally, CLAPs are gently heated to ~37 C to minimize RH effects. In this study, stations with high sample RH in the nephelometer sample (Table 1) are also the most likely to have issues with high sample RH in the collocated absorption photometer.

2) Size cut: As described in Table 1, the size cuts differ amongst the stations, but most of the sites measure TSP or PM10. The GAW program generally recommends a PM10 size cut, except for stations in extreme environments (clouds etc.) where a whole air inlet is recommended (WMO/GAW report 227, 2016; GAW/WCCAP recommendations https://www.wmo-gaw-wcc-aerosol-physics.org/files/WCCAP-recommendation-for-aerosol-inlets-and-sampling-tubes.pdf). Many stations in the NOAA Federated Aerosol Network measure at a second size cut (PM1) as well. PAY and SUM are the only stations that have no measurement of coarse mode aerosol with only a PM2.5 inlet. As reported previously, the amount of aerosol particles larger than 10 micrometers is usually sufficiently low to enable consideration of TSP and PM10 results as in the same category. Moreover, the trend results of PM10 and PM1 sampling are found to be quite similar for all stations with both size cuts, so that the results of TSP/PM10 size cut will be presented in this study and, if not specified, PM1 results can be assumed to be similar to those of the larger size cut (PM10 or TSP).

3) Absorption filter photometers artefacts: The first main point to consider is that all filter-based absorption photometers suffer from various measurement artefacts and that continuous reference measurements to assess the absolute $\sigma_{ap}$ values are not available at long-term monitoring sites. If the variability and the long-term trends of absorption coefficients are to be analysed with large confidence, the $\sigma_{ap}$ absolute value is necessary to compute the $\omega_0$. As stated in Sect 2.2, the real $C_{ref}$ values can potentially vary by a factor of 4 (1.5 to 5.5). Using an erroneous $C_{ref}$ value can influence the magnitude of the $\omega_0$ trends. Similarly, an applied correction depending on the wavelengths can affect the absorption Ångström exponent calculation and its trends. Both $\omega_0$ and $\mathring{a}_{ap}$ long-term trends therefore must be interpreted with greater care.

4) Nephelometer truncation correction artefacts: as explained in Sect. 2.2, the various types of nephelometer measure at different truncated angular ranges that were corrected by several algorithms or even not corrected. The absence of truncation correction leads to lower scattering and backscattering coefficients than the true values and the correction algorithm effects are known to increase with particle size. The most important requirement that was verified for this trend analysis is the coherent treatment of nephelometer data for each time

series. The bias leading to a higher contribution of Aitken and accumulation modes than the coarse mode is difficult to estimate, but the minimal differences in PM1 and PM10 results (see Sect 4.2) suggest this artefact is small. The effect of the humidity on the nephelometer measurements is regarded as the most significant artefact.

Finally, in order to minimize the potential artefacts in the determination of the long-term trends in the case of large seasonal variability (de Jong and de Bruin, 2012), only full start and end years of the time series, that is, without gaps in the data, were considered. For some stations, we did allow gaps of up to 4-6 weeks without measurements after checking that the removal of the whole year led to similar trend results.

The differences in instrumentation, measurement conditions, and post-processing data treatment do not allow the absolute values for all sites to be compared; however, because there was consistency of data treatment for individual sites, the trends can be compared.

## 2.5 Trend analyses

The aerosol extensive parameters ($\sigma_{sp}$, $\sigma_{bsp}$ and $\sigma_{abs}$) are not normally distributed and they exhibit varying degrees of autocorrelation. They can be represented approximately by a lognormal distribution but are usually better fitted by a distribution in the Johnson distribution family (Johnson, 1949). The intensive parameters (b, $\mathring{a}_{sp}$, $\mathring{a}_{ap}$ and $\omega_0$) also exhibit distributions that differ to varying degrees from the normal distribution. We chose, therefore, to rely mostly on the non-
parametric seasonal Mann-Kendall (MK) test associated with the Sen's slope. The MK test does not require normally distributed data. Additionally, as described under Sect. 2.5.1, the MK test was adapted to correctly handle autocorrelated datasets. To allow a comparison with other studies, the trends were also computed with the Generalized Least Squares analysis associated to the autoregressive or block bootstrap confidence intervals (GLS) and the least-mean square
(LMS) fit applied to the data logarithms.

### 2.5.1 Mann-Kendall test and the Sen's slope estimator

This non-parametric method based on rank (Gilbert, 1987; Sirois, 1998) is the most appropriate test to compute optical properties trends because it can be applied regardless of missing values, statistical distribution and presence of negatives or below detection limits values in the data set.
The MK test determines if a monotonic increasing or decreasing long-term trend exists, the slope and the confidence limits are then computed by the Sen's slope estimator that is based on the median of the slopes calculated from all possible data pairs. For this study, the MK test was applied on daily medians.

The MK test is designed for serially independent data and is, consequently, influenced by
autocorrelation in the time series leading to inflated type 1 error; that is, there is increased probability of rejecting the no-trend hypothesis (i.e., a false positive). Several correction schemes for the MK test were proposed to correctly handle autocorrelated datasets and the problems induced by autocorrelation and its various corrections have been clearly described (Wang and Swail, 2001, Yue et al., 2002, Zhang and Zwiers, 2004, Bayazit and Önöz, 2007, Blain, 2013,
Wang et al., 2015). A new method has been used for this study that tends to minimize the type 1 and 2 error (type 2 error is non-rejection of a false null hypothesis, i.e., a false negative). The new method also minimizes issues with the modification of the slope due to pre-whitening procedures by the application of three pre-whitening methods (Collaud Coen et al., submitted 2020). The standard pre-whitening (PW) by removing the first lag autocorrelation (von Storch, 1995) has a
very low type 1 error but also a low test power, whereas the so-called trend-free pre-whitening procedure published by Yue et al. (2002) (called TFPW-Y in Collaud Coen et al., submitted, 2020) restores the test power at the expense of the type 1 error. Both these prewhitening procedures were applied prior to the MK test to assess the statistical significance of the trend. A trend was then considered as ss only if both PW and TFPW-Y were ss at the 95% confidence level or if PW
is ss but not TFPW-Y (false negative). Among the trends of all parameters at all stations calculated for this paper, none was ss for the PW but not for the TFPW-Y, meaning that the PW procedure

was always powerful enough. In contrast, many trends were not ss when PW was applied, but were ss with the TFPW-Y procedure, leading to false positives and showing that the TFPW-Y rejection rate of the no-trend hypothesis is too high.

After having determined the statistical significance, a third prewhitening procedure, the variance-corrected trend-free prewhitening procedure (VCTFPW) allowing an increase in the slope accuracy (Wang et al., 2015) was applied prior to the Sen's slope estimation. The confidence limits of the Sen's slope were computed at the 90% confidence level.

Since many of the time series exhibited clear seasonal cycles, the modified seasonal MK test (Hirsch et al., 1982) was always applied to the four meteorological seasons. The annual trends were considered only if the slopes of the four seasons were homogeneous at the 90% confidence level (Gilbert, 1987; Sirois, 1998).

Figure 2 presents three examples of seasonal MK results and Sen's slopes of $\sigma_{sp}$. At JFJ, $\sigma_{sp}$ has ss negative annual trends for all of the analyzed periods, with the most recent 10 y period having a larger negative slope than the longer periods. Spring and autumn are the seasons at JFJ with the strongest ss trends; winter has tiny ss negative trends. MRN also exhibits $\sigma_{sp}$ annual negative trends for all of the analyzed periods, but only the 15 y, 20 y and 25 y trends are ss and their slopes are more negative for the longest periods. At MRN, summer and autumn are the seasons with the largest trends and that is true for all the trend periods of all lengths (10 y to 25 y), while spring and winter have more scattered and less significant slopes. Finally, MLO has annual trends that are ss negative for the last 10 y, not ss for the last 15 y, and ss positive the longest periods (20 y, 25 y and 30 y). The spring season at MLO exhibits a not ss negative trend for the last 10 y and positive trends for the longest periods with only 25 y and 30 y trends being ss.

### 2.5.2 Least Mean Square analysis (LMS)

Following the Weatherhead procedure (Weatherhead et al., 2000), the trend is estimated by fitting the following frequently used statistical model for monthly data with an LMS approximation:

$$Y_t = m + C_t + \rho \cdot (t/12) + M_t, \qquad\qquad t = 1..n, \qquad\qquad (3)$$

where m is a constant term, $C_t$ is a seasonal component, and $\rho$ is the magnitude of the trend per year. The unexplained noise term $M_t$ is modeled as an [AR(1)] process $M_t = \phi \cdot M_{t-1} + \epsilon$, where $\phi$ is the autocorrelation coefficient of the data noise. For this study, either the logarithm of the monthly medians or the monthly medians were taken for all the parameters. Due to the non-normal distribution of the studied parameters, the LMS method applied on the logarithm is considered as the standard method according to previous trend analyses (CC2013 and Asmi et al., 2013). A trend is considered as ss at the 95% confidence level if $|\rho/\sigma_\rho| > 2$, $\sigma_\rho$ being the standard deviation of the slope. Fig. 3 a and c show the LMS trends and statistics for MLO $\sigma_{sp}$, respectively. The LMS results are similar to the MK analysis, the last 10 y trend is negative but ss at only the 90% confidence level, the 15 y and 20 y trends are not ss and the 25 y and 30 y trends are ss positive. The normal probability plot of the residue (Fig. 3c) shows that the use of the logarithm of the data results in normally distributed residues as required by this statistical tool.

### 2.5.3 Generalized Least Square associated with autoregressive bootstrapping method (GLS/ARB)

A similar Generalized Least Squares (GLS) method based on the minimization of the least square errors similar to ordinary least squares fitting (including similar sensitivity to outliers), but taking into account the autocorrelation in the covariance matrix was also used in this study. The GLS uses an autoregressive bootstrapping algorithm (ARB) to evaluate the potential differences in the GLS trends arising from the noise terms (Asmi et al., 2013). The ARB methodology was used to produce 1000 realizations of the original time series, with randomized noise terms, and the resulting set of trends was used to determine the 5th to 95th percentile confidence intervals (ARB CLs) of the GLS trends. If the ARB CLs did not include a zero trend, we considered the GLS trend to be ss. The GLS and ARB methodologies were adapted from Mudelsee (2010) and applied to

both daily and monthly medians. The previous trend analyses (CC2013 and Asmi et al., 2013) used daily medians.

Fig. 3b and f shows the GLS/ARB results for MLO $\sigma_{sp}$ for daily and monthly medians. Here again the results are similar to the MK analysis, where the 10 y trend is positive but not ss, the 15 y trend is not ss while the longer periods exhibit ss positive trends. As with many other stations included in this study, the use of daily or monthly medians did not result in normally distributed residues (Fig. 3f); and, in fact, the residues of the daily and monthly medians appeared to represent different types of distributions. It is also obvious that the seasonality fits (fits from monthly and daily medians in red and orange on Fig. 3b) are different for the two time granularities, with similar shape but higher absolute trend values if fitted from daily medians. The timing of the winter minima is also more precisely defined with the daily data.

## 3. Results

### 3.1 Long-term trends ending in the present-day (2016-2018)

To assess the aerosol optical properties long-term trends, the largest number of stations around the world were included in this study. This overview takes into account the 10 y (or longer) trends ending in 2016, 2017 or 2018. The results shown here comprise not only the 10 y trends, but also the longer periods for 15 y to 40 y in 5-year increments also ending in 2016-2018. The results are presented for the MK analysis and a comparison between the trend analysis methods will follow in Sect. 3.3. Complete results for all the other methods can be found in the supplemental material.

### 3.1.1 Total scattering and hemispheric backscattering coefficients

Long-term trend analysis of $\sigma_{sp}$ has been performed on 37 data sets. Since some nephelometers only measure $\sigma_{sp}$ (Optec and Radiance Research nephelometers) and $\sigma_{bsp}$ was determined to be unusable for several other sites due to various discontinuities (see Sect. 2.4), the hemispheric backscattering coefficient trends were computed on only 28 data sets. The detailed results of MK trend analyses are given in Table 2 while the overall picture for $\sigma_{sp}$ is presented in Fig. 4. The results for $\sigma_{bsp}$ are very similar to those for $\sigma_{sp}$ for sites where both measurements existed; corresponding figures for $\sigma_{bsp}$ can be found in the supplement material (Fig S1, S2 and S7).

The $\sigma_{sp}$ ss trends are predominantly negative: 20 stations have ss negative 10 y trends, 5 stations ss positive trends and 12 stations no ss trends dispersed across all continents. Eight (nine) stations with time series longer than 10 y have ss negative 15 y (20 y) trends and none (two) of the 15 y (20 y) trends are ss positive. The MK slopes range between -2.45 to +0.39 $Mm^{-1}y^{-1}$ with a mean of -2.19 $Mm^{-1}y^{-1}$. The main results are as follows:

- Over North America, all the $\sigma_{sp}$ trends for periods longer than 10 y are ss negative and the most recent 10 y trends are generally ss negative. Three stations have not ss trends: (1) EGB's 9y time series does not allow for a ss trend (too short) but was included as one of only two Canadian sites, (2) MRN is an IMPROVE station on the west coast of the USA with very high humidity leading to condensation that can disturb the humidity measurement. This makes it difficult to know if the ss positive RH 10 y trend (Table S4) is real or due to measurement artefacts and uncertainties. If the ss positive RH trend is real, it could mask a decreasing $\sigma_{sp}$ trend resulting in a not ss trend. The time coverage for the dry $\sigma_{sp}$ ($\sigma_{sp}$ restricted to RH<50%) for MRN is too low to be representative for trend analysis. It should be mentioned that the 10 y trends for MRN ending in 2014-2018 are all not ss (see Sect. 3.2.1) so that the absence of $\sigma_{sp}$ trends seems to be a real phenomenon. (3) GLR is also an IMPROVE station with high humidity. The RH trends at GLR are also not ss, and the dry $\sigma_{sp}$ has a ss negative trend, similar to other stations in its vicinity.

  In the previous decadal trend paper (CC2013), the trends in scattering for the arid state of Arizona were not consistent (ss positive: IBB, ss negative: SIA, PAZ, not ss: HGC, SCN).

Four of the five Arizona sites (IBB, PAZ, SIA and SCB) were closed in 2010 and HGC now exhibits a ss decreasing scattering trend. MZW, the other IMPROVE station with ss positive scattering trends in 2010, also closed in 2010.

- Most (seven out of eleven) of the European sites have present day ss decreasing $\sigma_{sp}$ trends. The other four stations have not ss trends: (1) one urban station also influenced by Saharan dust (UGR), (2) two sites in Eastern European countries (KPS and BEO) and (3) a high altitude station in the Central Range in France (PUY). The ss negative scattering trends of the Scandinavian stations have lower absolute slopes than in central Europe. PAL, the northernmost station, has a ss positive trend. PAL is geographically situated in Europe but it can be climatologically considered an arctic station (Schmeisser et al. 2018). PAL (slope=0.06 Mm$^{-1}$/y) has a similar trend as ZEP (slope=0.05 Mm.$^{-1}$./y), the nearest Arctic station, with the largest ss trend in summer (JJA) when PAL is largely influenced by Arctic air masses. The increasing trend at PAL may be due to increasing biogenic secondary organic aerosol formation related to emissions from the surrounding boreal forest (Lihavainen et al., 2015a), changes in circulation patterns or a larger influence of open water with increasing concentration of sea salt aerosol.

- Sites in polar regions exhibit two ss positive $\sigma_{sp}$ trends. In addition to ZEP and PAL, SPO also has a ss positive present day 10 y trend but with lower slope, whereas no ss trend is found for the other Antarctic site (NMY). BRW and ALT both exhibit ss negative 10 y trends. The BRW 15 y $\sigma_{sp}$ trend is not ss, whereas longer periods up to 40 y lead to ss negative trends. SPO also has very long time series but with alternating trend slopes, from ss positive for the shortest periods (10-25 y) to ss negative for the longest periods (35-40 y), with some not ss trends in between. The aerosol load is very low at BRW and SPO leading to scattering coefficients near the instrumental detection limits, so that the measurement uncertainties are proportionally larger than for middle latitude stations.

- CPR, a site on the Caribbean island of Puerto Rico, has a ss positive $\sigma_{sp}$ trend. At CPR, the largest scattering trend is found in summer and the scattering trend of PM10 trend is five times larger than the PM1 trend. The most probable explanation is increased Saharan dust transport over the Atlantic ocean; more dust transport has been reported at an IMPROVE site in the Caribbean (Hand et al., 2017, Hand et al., 2019).

- The only two stations representing the Pacific region are MLO and CGO. The recent MLO 10 y $\sigma_{sp}$ trend is ss decreasing, the $\sigma_{sp}$ 15 y trend is not ss, whereas the trends for the longer time periods (20-30 y) are ss positive (see Fig. 2). In the previous decadal trend paper (CC2013), MLO exhibited a ss positive trend for the 10 y period ending in 2010. MLO $\sigma_{sp}$ trends changed from previously ss positive to currently ss negative trends. The recent 10y trend at CGO is found to be positive and quite homogeneous with the seasons, with fall being the only season without a ss trend.

- The $\sigma_{sp}$ trends are mostly (70%) not ss for stations at middle to high altitudes. From the 10 stations higher than 1100 m a.s.l., only SPO in Antarctica has a present day ss positive 10 y trend and only JFJ in the European Alps and HGC in Arizona and GBN in Nevada exhibit ss negative 10 y trends. In contrast, only 26% of the stations lower than 1100 m a.s.l. have not ss trends. New particle formation (NPF) and growth are favored at high altitudes (> 1000 m and up to 5000 m) due to low temperatures, high solar radiation and low pre-existing particle concentrations leading to limited condensational sinks for nucleation precursor gases (Sellegri et al., 2019). This higher frequency of nucleation at high altitude leads to a high contribution of secondary particles to the total number concentration that largely contributes to the total scattering coefficient. The decreasing $\sigma_{sp}$ trends from anthropogenic pollution in the planetary boundary layer can, consequently, be masked by the presence of NFP at high altitude stations.

The seasonal MK results for $\sigma_{sp}$ are presented in Fig. 5. Spring is the season with the largest number of ss decreasing trends and winter with the lowest. ZEP and PAL exhibit ss positive trends only in summer and BRW has ss negative trends only between December and May. The

SPO annual trend is ss positive whereas it is not for NMY. Both Antarctic stations exhibit, however, a coherent seasonality with ss positive trends only in spring. While the 25 y and 30 y trends at MLO are all ss positive with the largest slope in spring when MLO is influenced by Asian long-range transport (CC2013), the most recent 10 y-20 y trends are not ss for the individual seasons.

### 3.1.2 Absorption coefficient

The analysis of $\sigma_{ap}$ long-term trends has been performed on 33 datasets (see Fig. 6 and Tables 2 and 3). The long-term trends are ss decreasing (21 stations) or not ss (12 stations) for all stations around the world leading to a mean decreasing trend of -3.05 Mm$^{-1}$y$^{-1}$. No ss $\sigma_{ap}$ positive trends are measured for any of the stations. The other main results are:

- In North America the number of $\sigma_{ap}$ datasets is much lower than the number of $\sigma_{sp}$ datasets (IMPROVE sites do measure aerosol absorption, but with a different instrumental setup (White et al., 2016)). From the five sites with long-term aerosol absorption, APP and BND, two continental rural sites, and the marine Caribbean island (CPR) station have ss negative trends. The other three stations representing continental rural US (SGP, EGB) and marine west coast of the US (THD) exhibit not ss trends in $\sigma_{ap}$.

- In Europe, most (12 stations) of the 10 y $\sigma_{ap}$ trends are ss negative. Only three stations, one in Scandinavia (PAL), one eastern rural continental (KPS) and one coastal Mediterranean (FKL) station exhibit no ss trends. The 15 y $\sigma_{ap}$ trends at JFJ and FKL are ss negative.

- In Asia, both the high altitude stations of LLN in Taiwan and WLG in China exhibit annual ss decreasing $\sigma_{ap}$ trends. The South Korean coastal station of AMY has no ss annual trend.

- For polar regions, the Antarctica site of NMY, the American Arctic site of BRW and the Russian Arctic site of TIK have slight ss negative $\sigma_{ap}$ trends, whereas SUM, ALT and ZEP have no ss trends. Thus, there is no common clear $\sigma_{ap}$ trend in polar regions.

- In the southwest Pacific, the high altitude station of MLO has a ss decreasing trend for the last 10 y but no ss trend for the last 15 y, whereas the coastal station of CGO in Australia exhibits not ss $\sigma_{ap}$ trends.

- In contrast to the $\sigma_{sp}$ trends, $\sigma_{ap}$ trends at high altitude stations (> 1100 m a.s.l) are mostly (6 out of 8) ss decreasing, the trends at the other two high altitude stations are not ss.

The seasonal trends are more strongly negative and more ss in spring than in summer (see Fig S3). Winter is the season with the smallest number of ss decreasing trends in Europe (only 2/15) and with the only ss positive trend (ZSF), the others being not ss, whereas fall seems to be the season with the least ss trend in North America.

### 3.1.3 Single scattering albedo

As described under Sect. 2.4, $\omega_0$ trends have to be considered with greater caution since the $\sigma_{ap}$ absolute values suffer from a certain uncertainty related to filter-based absorption photometer artefacts.

The $\omega_0$ trends depend directly on both the magnitude and the sign of the $\sigma_{sp}$ and $\sigma_{ap}$ trends. If expressed in %/y, a $\sigma_{ap}$ trend larger (smaller) than the $\sigma_{sp}$ trend will result in an increasing (decreasing) $\omega_0$ trend, respectively (see Fig. S8 and related estimation of $\omega_0$ uncertainty due to measurement and $C_{ref}$ errors). The $\omega_0$ trends are consequently much more diverse than the $\sigma_{sp}$ and $\sigma_{ap}$ trends with 52% of ss positive (relatively more scattering), 22% of ss negative (relatively more absorption) and 26% not ss trends (see Fig. 7 and Table 2). One peculiarity is that all $\omega_0$ ss negative trends are found between latitude 30 and 50, but this is perhaps due to the low spatial coverage outside of North America and Europe. The main results are:

- The $\omega_0$ is decreasing at three stations in North America (BND, SGP and THD), whereas APP and CPR exhibit ss positive $\omega_0$ trends. The CPR $\omega_0$ increasing trend can perhaps be related to increased Saharan dust load. The seasonal $\omega_0$ trends at CPR are, however, ss not only in summer when Saharan influence is the greatest, but for every season except spring (Fig. 8). EGB has no ss trend.

- European stations exhibit ss increasing $\omega_0$ trends at the urban station of UGR and at most eastern and Scandinavian stations (KPS, SMR, PAL) and at the mid-altitude station of HPB. These ss positive $\omega_0$ trends in eastern and northern Europe are the strongest in summer (Fig. 8), when MEL and BIR are also ss positive, and the weakest in winter when only PAL is ss positive (possibly related to increased particle formation from biogenic emissions, as mentioned above). In central Europe, JFJ, IPR and MSY have ss negative $\omega_0$ trends for the entire year as well as for all seasons. PUY, a station at 1465 m in France's central range, has a ss positive annual trend due to strong positive trends in autumn and winter, even if a strong ss negative trend is found in summer. Because the site is located at a mid-range elevation (1465 m asl), PUY has a large probability of being influenced by different air masses as a function of the season, with a large impact of the planetary boundary layer in summer (Collaud Coen et al., 2018; Hervo, 2013 ).

- The high altitude stations of LLN and WLG in Asia have a strong and a weak ss positive annual $\omega_0$ trends, respectively. This pattern is also observed for all seasonal trends at LLN but only in autumn at WLG. The coastal station of AMY has ss decreasing annual $\omega_0$ trend that is due to decreasing trends in MAM, SON and DJF. AMY is located in an agricultural and touristic region that is influenced not only by these regional aerosol sources (e.g. traffic, field burning), but also by long-range transported plumes with high aerosol load.

- The Arctic stations of ALT and ZEP have ss positive $\omega_0$ annual trends, which are due to ss positive trends from December to August for ALT and from December to May for ZEP. The two polar stations (BRW and NMY) exhibit no ss $\omega_0$ annual trends, although there is a ss positive trend in summer at BRW for the most recent 10 y time series.

### 3.1.4 Backscattering fraction and scattering Ångström exponent

The present-day trends for the backscatter fraction b are mostly ss positive (65%) across all regions (Fig. 9).  This suggests a shift in the size distribution towards smaller accumulation mode aerosol. The two stations with ss negative trends are CPR in Puerto Rico and BEO, located on a summit in the Balkan range. Not ss trends are mostly found in eastern and northern Europe (KPS, MEL, BIR and PAL), in Antarctica (NMY), as well as at BND and AMY for the last 10 y. The Arctic sites (ALT, BRW and ZEP) all exhibit ss positive b trends. CPR's seasonal trend is ss negative only in fall, trends in b for the other seasons at CPR are not ss (see Fig S4). Similarly, the BEO b seasonal trend is ss negative only in summer, and not ss otherwise. PAL has ss positive b trend in spring and summer and ss negative b trends for autumn leading to an annual not ss trend.

The scattering Ångström exponent ($å_{sp}$) trends exhibit a higher variability than the trends in other parameters with 33% of ss positive, 37% of ss negative trends and 30% of not ss trends. There are ss positive and negative trends in North America, Europe and Polar Regions, and the various trends cannot be attributed to specific regions or environments. It should be recalled, however, that $å_{sp}$ is affected by higher uncertainties (see Sect 2.3) that may contribute to the larger observed variability. The seasonal results also exhibit high variability, with summer being the season with the least number of ss $å_{sp}$ trends (10 out of 26 sites), while spring and fall are the seasons with the largest number of ss positive and negative trends in $å_{sp}$  (8 out of 26 sites), respectively (see Fig. S5).

### 3.1.5 Absorption Ångström exponent

The number of stations with long-term $å_{ap}$ measurement is low, with only 14 time series available. Seven stations situated in various geographical regimes exhibit ss positive trends: polar regions

(ALT and ZEP), a Carribean coastal station (CPR), high altitude stations in the remote Pacific (MLO) and in continental (WLG) and coastal (LLN) Asia, and in rural continental North America (SGP). SMR and JFJ, two stations in Europe but with very different environmental footprints and altitudes, exhibit ss decreasing $\text{å}_{ap}$ trends, the six other stations, consisting of 3 coastal and 2 continental sites, have no ss trends.

While CPR and SGP $\text{å}_{ap}$ trends are ss positive and JFJ ss negative for all four seasons, the other stations exhibit higher variability as a function of the meteorological seasons (see Fig. S6). The absorption Ångström exponent is principally a function of the particle chemical composition and material properties but its assignment to an aerosol type is not uniquely defined and also depends on the particle size, with larger particles corresponding to lower $\text{å}_{ap}$ values (Liu et al., 2016, Schmeisser et al., 2017). For example, $\text{å}_{ap}$ >2 corresponds to mineral dust in the case of big particles, and to brown carbon in the case of small particles. In contrast, $\text{å}_{ap}$ <1 corresponds to large particles with small absorption like sea salt dominated aerosol in the case of big particles, and to BC dominated aerosol in the case of small particles. Following these observational constraints, the JFJ and SMR aerosol tends to represent the category "mixed BC/BrC" according Schmeisser et al. (2018). CPR absorption has a strong contribution from mineral dust and sea salt, whereas at MLO, SGP, ALT and ZEP contributions to absorption are from mixed sources including various light-absorbing carbon species and dust. Ideally, direct chemical composition measurements would provide more precise information on the aerosol type, but the necessary chemical composition measurements are not yet readily available at many sites.

### 3.2 Time evolution of 10 year trends

The previous section describes the present day trends for different periods extending from 10 y to 40 y. Another interesting analysis is to follow the evolution of the trends in time and space. For this purpose, all the possible 10 y trends were computed and plotted as a timeline for each station. In what follows each point on the timeline represents a 10 y trend ending in the year it is located on the graph. For example, in Figure 12 the two black points for AMY represent the 10 y trends covering the periods 2008-2017 and 2009-2018, respectively. These timelines can be presented as a function of the latitude, longitude, altitude or environment. Depending on the results, the most interesting representation has been chosen for each parameter.

#### 3.2.1 Scattering and backscattering coefficients

Figure 12 presents the $\sigma_{sp}$ 10 y trend timelines as a function of the longitude of the stations. The $\sigma_{bsp}$ 10 y trend timelines are similar to the $\sigma_{sp}$ trend timelines (see Fig. S7). The polar stations (in both the Arctic and Antarctic) have been gathered to the bottom of the figure just after the two Pacific stations of MLO and CGO. The main result is that the sites in eastern and central North America (longitude between -68° and -112°) have ss negative $\sigma_{sp}$ 10 y trends ending after 2009-2012 regardless of their altitude (200 to 2200 m a.s.l.) and their environments. This is a clear signature of continental scale modification due to air quality regulations and this very clear feature relates to the sulfate-dominated aerosol in the eastern US and to large $SO_2$ reductions in power plants emissions (Hand et al., 2014, McClure and Jaffe, 2018). Almost all the $\sigma_{sp}$ 10 y trends in the southwest US (MZW, SCN and HGC) ending before 2011 are ss positive as published in the previous trend analysis (CC2013). MLO also exhibits ss positive trends for the same period. These four stations are also high altitude sites (2000-3400 m) so that it is possible that all of them were influenced by long-range transport of highly polluted air masses from Asia (CC2013). It is further interesting to note that the high altitude site JFJ (3580 m) in Europe also exhibited a ss positive 10 y trend ending in 2005-2008.

The evolution of the European $\sigma_{sp}$ 10 y trends does not show a clear time for trend modification as is seen in North America, probably due to variable timing in implementation of abatement policies in each individual country. Apart PAL (which can be considered, to some extent, as a polar station), the $\sigma_{sp}$ 10 y trends in Europe ending after 2008 are all ss negative or not ss. The four stations in Asia have not ss trends ending in the last 5 years. The two African stations exhibit

no ss trend. For polar sites, BRW, ALT and NMY have mostly not ss trends, whereas SPO exhibits alternating ss positive and negative trends, with the oldest 10 y trends being not ss. In contrast, ZEP exhibits positive trends for all three 10 y periods, which is similar to the 10 y trends at PAL for the same time periods. Due to the very low aerosol concentrations at these sites and, thus, larger measurement uncertainty, it is difficult to interpret the evolving $\sigma_{sp}$ polar trends. They could be related to increased influence from the boreal forest and/or changed circulation patterns modifying the sea/ice influence.

The GSN dataset only covers 8 years with some missing periods due to the destruction of the station by a typhoon. Due to the very low number of long-term measurements in Asia, GSN was included in this study. While GSN $\sigma_{sp}$ summer trends are not reliable (low data coverage and issues in humidity control), the ss negative winter-spring trends corresponding to the dry season are valid and in line with the PM10 decreasing trends in Korea (Kim and Lee, 2018; Nam et al., 2018).

### 3.2.2 Absorption coefficient

The lengths of the $\sigma_{ap}$ time series are much shorter than for $\sigma_{sp}$ (Fig. 13). This means that the oldest 10 y trends cover the period 1998-2007 (BRW and BND), followed by MLO (2001-2010) and JFJ (2002-2011).  For these four stations, the most recent 10 y trends are either not ss or ss negative. The present day (i.e., trends covering 2009-2018) ss negative 10 y trends (JFJ, BND, MLO and BRW) are preceded by not ss trends. The $\sigma_{ap}$ 10 y trends evolution of each station is usually homogeneous with either ss negative or not ss 10 y trends in Asia, Europe, Africa and North America. ALT $\sigma_{ap}$ 10 y ending in 2017 is ss positive. The BRW polar station and MLO high altitude station exhibit also some ss positive 10 y trends ending between 2010 and 2014. Unfortunately, only MLO has a long enough $\sigma_{ap}$ time series to compare with the ss positive $\sigma_{sp}$ 10 y trends at high altitude sites (Fig. 12). At MLO, the series of ss positive $\sigma_{sp}$ 10 y trends ended in 2008, while the series of ss positive $\sigma_{ap}$ 10 y trends occurred for the period ending 2009-2013.

Figures 12 and 13 suggest that mid-latitude $\sigma_{sp}$ and $\sigma_{ap}$ sequential 10 y trends were ss positive for some periods between 2000 and 2013, followed by not ss trends and ending in the present day with ss negative trends. The evolution from increasing to decreasing $\sigma_{sp}$ and $\sigma_{ap}$ trends appears to be not simultaneous, with the $\sigma_{sp}$ inflection points occurring some years before those for the $\sigma_{ap}$ trends. The sparse number of stations with long enough time series does not allow generalisation of this result.

### 3.2.3 Single scattering albedo

Because it is limited by the length of $\sigma_{ap}$, time series, the $\omega_0$ 10 y trends evolution also only covers the last decade. The following results can be seen in Fig 14:

- All stations at longitude > 10° have ss positive $\omega_0$ 10 y trends except for AMY, which exhibits a ss positive 10 y trend ending in 2018, and MUK with a not ss 10 y trend ending in 2013. For European sites, ss positive $\omega_0$ 10 y trends exist for all stations at latitude >46.8° apart from BIR which has a not ss trend ending in 2018. This suggests that the decreasing $\sigma_{ap}$ trends in Asia and in eastern and northern Europe are proportionally larger than the decreasing $\sigma_{sp}$ trends.

- The central and western European sites exhibit mostly ss negative or not ss $\omega_0$ 10 y trends. At JFJ and IPR, a shift between not ss to ss negative 10 y trends occurred in 2013-2014. The JFJ time series is moreover long enough to monitor a ss positive $\omega_0$ 10 y trends ending in previous years (2010).  The urban station of UGR in Spain, exhibits an increasing trend in $\omega_0$ (decrease in contribution of absorbing aerosol) for the most recent 10 y period (2009-2018), possibly relating to long-term effects of the 2008 financial crisis (e.g., Lyamani et al., 2011).

- In North America, the $\omega_0$ 10 y trends ending after 2013 are ss negative or not ss, apart from CPR in Puerto Rico and APP. The sites with the longest series of $\omega_0$ 10 y trends (BND and THD) exhibit ss positive trends followed by not ss and ss negative trends. In

contrast, MLO $\omega_0$ 10 y trends shifted from ss negative trends (10 y trends ending in 2010-2015) to not ss trends (10 y trends ending 2017) to ss positive trend in 2018. This is consistent with the observed increase of $\sigma_{ap}$ 10 y trends ending in 2010-2012.

● The two polar sites of BRW and NMY exhibit mostly not ss $\omega_0$ 10 y trends, whereas ALT and ZEP, similar to northern European sites, exhibits ss positive $\omega_0$ for all the 10 y trends.

### 3.2.4 Backscattering fraction and scattering Ångström exponent

Both the b and $å_{sp}$ 10 y trends in Asia and Africa exhibit similar 10 y trend patterns that are either ss positive or not ss (Fig. 15). In this context, similar means that the b and $å_{sp}$ trends are never ss when opposite signs of the slope are observed. These results suggest that particle average size tends to decrease at the Asian and African sites. In Europe, b and $å_{sp}$ 10 y trends have a majority of ss negative or not ss trends in the northeast (longitude > 10°). At lower European longitudes, there is a discrepancy between b and $å_{sp}$ 10 y trends, with IPR, JFJ and PUY having opposite ss trends for b and $å_{sp}$, b trends being often ss positive and $å_{sp}$ trends often ss negative. The discrepancy in the signs of the trends for b and $å_{sp}$ may be related to shifts in both the fine and coarse modes of the aerosol size distribution - this is discussed more below (see Sect. 4.2).

In North America, the b 10 y trends ending after 2012 are almost all ss positive whereas the previous 10 y trends are ss negative at BND and MLO. As in western America, one can see discrepancies in the sign of the slope for b and $å_{sp}$ 10 y trends, with APP, CPR, MLO, SGP and THD having at least one 10 y period with opposite signed ss trends. Also, as in western Europe, the b trends are usually ss positive and the $å_{sp}$ trends ss negative. BND records four 10 y periods with opposite ss trends. In contrast to the other stations, BND $å_{sp}$ trends are ss positive while BND b exhibits ss negative trends.

In polar regions, the b 10 y ss trends ending after 2014 are all ss positive, whereas the older trends are primarily ss negative. Here again, the discrepancy between b and $å_{sp}$ 10 y trends is large, with all the 10 y b and $å_{sp}$ trends for ZEP and ALT being ss with opposite signs. In contrast, BRW exhibits trends with the same sign for both parameters that can be interpreted as an increase of average particle size for early years followed by a decrease after 2014.

### 3.2.5 Absorption Ångström exponent

The $å_{ap}$ time series are not very long, because the first generation of absorption photometers used either white light or only one wavelength (Fig. 16). The longest time series of $å_{ap}$ begins in 2002 at JFJ and exhibits a continuous ss $å_{ap}$ decrease. Similar to the results for JFJ, most of the stations have consistently ss negative (JFJ, SMR), ss positive (ALT, ZEP, IPR, SGP, WLG, LLN, MLO and CPR) or not ss 10 y trends (TIK, BRW, THD, APP, GSN, MUK and CPT). The not ss trends of TIK and GSN may be due to datasets shorter than 10 y.

### 3.3 Comparison of the trends among methods

As described under Sect. 2.5, the long term trends were computed with three methods (MK, GLS and LMS), where GLS was used on both daily and monthly medians and LMS with and without taking the logarithm of the monthly medians. These methods are thereafter called GLS/day, GLS/month, LMS/log and LMS/lin, respectively. Tables S2 and S3 give the GLS/day and LMS/log results for all parameters and stations. Table 3 presents an overview of the number of present day 10 y trends that are ss with each method. Due to the reasons described in Sect. 2.5.1, MK is considered as the most appropriate method for aerosol optical parameters. The agreement between the three methods used in CC2013 (MK, GLS/day and LMS/log), between all five methods are then also reported in Table 3, as well as the number of cases with either GLS/day or LMS/log agreement with MK and with both GLS/day and LMS/log disagreement with MK. The following conclusions can be derived:

- Generally, the trends computed by the various methods agree very well with another. Among all parameters, all stations and all periods, none of the present-day trends presents ss results with opposite slope for different methods. In all cases, the differences among

methods relates either to the degree of the ss or to the sign of the slopes for not ss trends. This implies that the main conclusions of this study would not have been fundamentally different if the other methods were used.

- GLS applied on daily medians is the method that has the largest number of ss trends for all parameters.

- The three methods applied on monthly data have lower number of ss trends for all the computed parameters ($\omega_0$, b, $\mathring{a}_{sp}$ and $\mathring{a}_{ap}$)

- The three methods used in 2013 have similar statistical significance (comprising cases with no ss trend) in 44% to 86% of the cases, whereas the five methods used here exhibit consistency in 37% to 82% of the cases. The measured parameters, which are less uncertain than the calculated parameters, always exhibit largest agreements amongst the methods (> 69% for the three methods used in 2013 and > 63% for the five methods utilized here). $\omega_0$ is always the parameter with the largest dissimilarity among the methods and $\sigma_{bsp}$ the parameter with the largest similarity among methods.

- The MK statistical significance is similar to at least one of the methods applied in 2013 in more than 90% of the cases for all of the parameters apart from $\omega_0$ (88%) and $\mathring{a}_{ap}$ (78%). This lower level of agreement can be explained by the fact that $\omega_0$ and $\mathring{a}_{ap}$ are almost normally distributed so that the use of the LMS/log is not appropriate.

The boxplots of the slopes computed by the various methods (Fig. 17) show first that the application of the logarithm to transform to a normal distribution for $\omega_0$ and $\mathring{a}_{ap}$ (not shown) is not suitable and leads to very large interquartile ranges. While the measured parameters are clearly not normally distributed, the derived parameters usually have distributions that more closely approximate normal distributions. No systematic rule could be deduced, since the distributions of each computed parameter largely depends on the individual stations. It seems however that $\omega_0$ and the Ångström exponents are closer to the normal distribution than to the lognormal distribution.

The Sen slope estimator applied to the variance corrected prewhitening (Wang et al., 2015) leads, in almost all cases, to a median of the slope nearer to zero than the other methods. The VCTFPW method was developed specifically to get rid of the falsely increased slope by the trend free prewhitening process (Collaud Coen et al., submitted 2020). The LMS/log method sometimes results in lower absolute slope medians and this effect is probably due to the almost normal distribution of the data (= log of the monthly median). Both the GLS (GLS/day and GLS/month) and the LMS/lin method lead to higher absolute slopes, probably due to misuse of statistical methods developed for normally distributed data.

The GLS/day method leads to broader range of slopes than the GLS/month method. This larger variance may be due to (1) the larger variability of daily data leading to a less distinct seasonal cycle and, consequently, to a worse fit of the seasonal variation, and (2) a higher autocorrelation in the daily time series with, possibly, an autocorrelation order larger than one.

**4. Discussion**

### 4.1 Considerations related to measurement humidity

As explained in the instrumental section, GAW protocol suggests that the $\sigma_{sp}$ and $\sigma_{bsp}$ be measured at low and controlled humidity, and that is the case for almost all stations considered here, except for those in the IMPROVE network which measure at ambient conditions due to their different monitoring goals. Temporal cycles and variations of RH with time are observed in a number of datasets. There are also some clear breakpoints in measurement RH that have been identified at several stations (e.g., an insulating jacket was installed on the nephelometer at THD in late 2012 resulting in a clear decrease in sample RH due to warmer nephelometer

temperatures). It is evident that high RH will enhance particle diameters and, consequently, increase $\sigma_{sp}$, $\sigma_{bsp}$ and $\omega_0$ while resulting in decreased b and $\mathring{a}_{sp}$. This particle diameter enhancement depends not only on the RH values but also on the particle hygroscopicity, which is a function of the aerosol size distribution and chemical composition.

Similar to the previous aerosol optical properties trend study (CC2013), dry $\sigma_{sp}$ was calculated by removing data when measurement RH was higher than 50% in order to minimize the impact of aerosol hygroscopicity on the scattering trends. However, hygroscopic growth can occur for RH < 50%; for example, for sea salt aerosol, up to 25% of the scattering could be due to water at RH=40% (e.g., Figure 5 in Zieger et al. 2013). The confounding effects of aerosol water impact

the reported scattering values and, hence, the trends presented here to a greater or lesser extent. The effect of hygroscopic growth at RH<50% on the reported trends would depend on the temporal variability in sample RH, composition and size; investigating the interactions amongst those parameters is beyond the scope of this study.

   For this study, if RH was frequently larger than 50% at a station, relationship between RH and

aerosol parameters trends were analysed as follows. In the case of the RH trend being not ss, the aerosol parameters trends were considered to be independent of the RH variation. In the case where a ss RH trend was detected (see Table S4 in supplement), an attempt was made to try to determine the influence of RH trend on each aerosol parameter by considering the following situations: (1) if all aerosol trends follow the RH trends, (2) if $\sigma_{sp}$ at all measurement RH and dry

$\sigma_{sp}$ trends are similar, and, finally, (3) the features of $\sigma_{ap}$ trends, which are less likely to be influenced by long-term RH variation. The distinct patterns exhibited by the evolution of the 10 y trends was very helpful in this analysis. Below we describe the assumed implications for scattering trends at sites where trends in RH were observed for several cases:

- Trends in RH are the opposite of both $\sigma_{sp}$ and $\sigma_{bsp}$ trends: this implies that the aerosol
optical properties trends are real and not influenced by humidity (SMR, SHN, MRN)

- Trends in RH are ss but trends in $\sigma_{sp}$ are not ss: this implies that the absence of statistical significance for the $\sigma_{sp}$ trends is real if the slopes of the RH and $\sigma_{sp}$ trend have the same sign (IZO, LLN) or can be partially induced by the RH trend if the slopes have opposite signs (EGB, PUY, UGR)

- Trends in RH and $\sigma_{sp}$ are similar, the overall and dry (RH<50%) $\sigma_{sp}$ trends are similar, and $\sigma_{sp}$ and $\sigma_{ap}$ exhibit similar trends: This implies that the $\sigma_{sp}$ trends are probably influenced by RH but also have an intrinsic aerosol trend (APP, BIR, MZW, SGP).

- Trends in RH and $\sigma_{sp}$ are similar, but the dry (RH<50%) and overall $\sigma_{sp}$ trends are dissimilar and the trends in $\sigma_{sp}$ and $\sigma_{ap}$ are also dissimilar: this implies that the RH
influence is major (THD, CPR). THD and CPR are coastal stations with a dominant influence of sea salt. At THD on the North America west coast, RH, $\sigma_{sp}$ and $\sigma_{bsp}$ trends are ss decreasing, whereas b and $\mathring{a}_{sp}$ trends are ss increasing and $\sigma_{ap}$ trends are also ss decreasing but with lower slope and ss than $\sigma_{sp}$. Further, the PM1 trends were less ss and exhibited much lower slopes, suggesting that the large sea salt particles are probably
sensitive to the RH decrease, leading to the decreasing $\sigma_{sp}$ trend. The 10 y trends show that RH decreasing trends are particularly important until 2015 and likely explain the $\omega_0$ ss positive 10 y trends ending in 2012 and 2013. At CPR, a coastal site in the Caribbean, RH, $\sigma_{sp}$ and $\sigma_{bsp}$ trends are ss increasing, but the $\sigma_{ap}$ 10 y trends do not have the same shape or statistical significance as the $\sigma_{sp}$ trends. As observed at THD, the PM1 trends at
CPR are less ss and have much lower slopes than PM10 trends.

   As mentioned in the instrumental section, RH trends measured by the nephelometers have to be considered with caution. Because the measurement RH is only a secondary parameter, the instrument humidity sensors are typically not maintained or calibrated with the same care as the scattered light detectors. The influence of humidity variations on the optical properties trends

presented here can generally be considered as low, apart from the cases of very hygroscopic particles like sea salt (e.g., at THD and CPR). A better knowledge of the particle hygroscopic

growth at low RH (< 40%) would be valuable in order to interpret $\sigma_{sp}$ and $\sigma_{bsp}$ trends as well as trends in $\omega_0$, b and $å_{sp}$.

## 4.2 Particle size trends

Both the scattering Ångström exponent and the backscattering fraction are indicators of the
particle's average size, with the general interpretation that lower values of b and $å_{sp}$ correspond to the presence of larger particles albeit at different parts of the aerosol size distribution (Collaud Coen et al., 2007). However, the relation between b and $å_{sp}$ is not uniquely defined for several reasons. First, the scattering efficiency has an oscillating response to particle size rather than a constant increase. Second, the measured particle size distribution is usually composed of several
modes. Since the sensitivity of scattering to the mode depends on the size parameter (proportional to the ratio: diameter/wavelength), b (here usually taken at 550 nm) and $å_{sp}$ (here usually computed with the 450-550 nm pair) do not always exhibit similar sensitivity to the various size modes. Further, the extinction Ångström exponent (analogous to $å_{sp}$) was found to be more sensitive to fine mode volume fraction if computed from long wavelengths and to fine mode
effective radius if computed from short wavelengths (Schuster et al., 2006). Lastly, the relation between b and $å_{sp}$ also depends on the refractive index and consequently on the absorption coefficient (Hervo, 2013): for a constant particle diameter, an increase of the refractive index real part will decrease $å_{sp}$ but increase b.

In this analysis, some stations exhibit b and $å_{sp}$ trends with the same sign (BRW, CPT, SMR, LLN,
SPO, UGR), while for other stations b and $å_{sp}$ trends are in opposite directions (ALT, APP, BEO, BND, EGB, JFJ, MLO, MSY, PAL, PUY, SGP, SPO, ZEP). The plots showing the evolution of the 10 y trends (Fig 15) demonstrate that b and $å_{sp}$ can exhibit either similar or opposite trends depending on the considered periods (CPR, IPR, MLO, THD). The plots showing the evolution of the 10 y trends also suggest that the variations of the 10 y slopes are often identical in sign but
different magnitude (e.g., shifted towards larger trend values for b (see for example MLO, Fig. 18)).

We can attribute both b and $å_{sp}$ ss positive trends (ALT, BRW, SMR, LLN, THD, UGR) to a shift of the accumulation mode towards smaller sizes and a decrease of the coarse mode particle concentration. In contrast, ss negative trends (BEO, CPR) for both b and $å_{sp}$ suggest a shift to
bigger sizes, specifically an increase in the coarse mode particle concentration and perhaps also a shift towards larger diameters of the accumulation mode. At a boreal forest site in northern Europe (SMR), size distribution data suggests that seasonal variation of b and $å_{sp}$ was caused by a shift in the accumulation mode and not by changes in the coarse mode fraction (Luoma et al., 2019). Trends towards smaller particle size might be due to an increase of near anthropogenic
sources of pollution, to an increase in new particle formation, to a decrease of long-range transport of anthropogenic pollution, to increased scavenging of larger particles due to changes in atmospheric conditions, to a modification of atmospheric chemistry (Banzhaf et al., 2015) or to a change in both primary and secondary natural aerosol (e.g. an increase of biogenic secondary aerosols and their precursors as demonstrated by Ciarelli et al., 2019). Trends towards bigger
particles can relate to a decrease of near anthropogenic emissions, to larger influence of mineral dust caused by variation in desert emissions or dust transport, to changes in agricultural activities or to an increase of humidity.

For stations with opposite b and $å_{sp}$ trends, the chemical composition may play an important role in identifying reasons for the changing trends. It is however out of the scope of this paper to study
these kind of dependencies.

## 4.3  Single scattering albedo trends

The single scattering albedo is the most important variable determining the direct radiative impact of aerosol so that its trend analysis - derived for the first time for a large number of stations - has a high relevance. The filter-based absorption photometers artefacts lead to uncertain absorption
absolute values that have no effect on $\sigma_{ap}$ trends but impart higher uncertainties to $\omega_0$ trends. The results of $\omega_0$ trends depend directly on the relative values of $\sigma_{sp}$ and $\sigma_{ap}$ trends. The global picture

is nuanced, with about half of ss positive, 1/5 of ss negative and a 1/4 of not ss trends leading to annual positive median trend of 0.02%/y (Table 4). The median of ss trends are increasing in Asia, in Arctic and in the Pacific, but decreasing in Europe and North America. The largest median slopes are found in Asia and in the Pacific (+0.13-0.14%/y), whereas the decreasing median slopes in other regions are relatively small (< 0.01%/y). The beginning of the decrease of the aerosol burden varies with region; the earliest decrease is found in Europe in the 1980's (Tørseth et al., 2012), followed by North America in the 1990's (Bodhaine and Dutton, 1993, Hand et al., 2012) and by Asia some 10-15 years ago (Sogacheva et al., 2019, Zhao et al., 2019, Paulot et al., 2018). The median slope of the $\omega_0$ trends seems to be proportional to the length of the mitigation efforts, which for some relevant pollutants (e.g., black carbon, $SO_2$ and $NO_x$) are still ongoing. In Europe, the diversity of the timing of abatement policies with earlier impact in Western Europe than in Eastern Europe (Vestreng et al., 2007, Crippa et al., 2016, Huang et al. 2017) is also directly visible in the decreasing and increasing $\omega_0$ trends(Fig. 7 and 14), respectively.

These results suggest that policy regulations induced first a $\omega_0$ increase (cooling effect) and, in a second phase, a $\omega_0$ decrease (warming effect). The Emission Database for Global Atmospheric Research (EDGAR V4.3.2) (easy accessible via https://eccad.aeris-data.fr/) shows that both the black carbon (BC) and $SO_2$ emissions decreased rapidly during the 1990s, and that currently emission reductions of $SO_2$ are larger than the reductions for BC. From this we conclude that the reduction of primary particles, such as BC, leads first to the $\omega_0$ increase, whereas the reduction of $SO_2$, a precursor of secondary particle formation, tends to result in a $\omega_0$ decrease. Moreover, emission changes can lead to modification of the atmosphere chemistry. Banzhaf et al. (2015) shows, for example, that sulfate and nitrate formation have increased in efficiency by factors between 20-25% between 1990 and 2009. The decrease in sulfate and total nitrate concentrations nitrate is consequently smaller than expected (non-linear response), leading to lower trends than the trends in precursor emissions and concentrations. This different timing and evolution in primary and secondary aerosol concentrations could explain the evolution of the 10 y $\omega_0$ trend at IPR, JFJ, BND and THD (Fig. 14) but the time series are not long enough to properly assess this change.

These observed $\omega_0$ trends are in line with the modelled impact of aerosol on climate (Zhao et al., 2019). They found a global cooling effect of -0.41K due to growth of aerosol burden caused by an increase in energy use in the northern hemisphere (particularly in Asia) is counterbalanced by a global warming of +0.10 K caused by the decreased aerosol emissions due to technology advances particularly in North America and Europe. This illustrates the complex nexus of environmental pollution regulations which have positive effects for health and the environment (air pollution is a primary cause of premature deaths in much of the world, Landrigan et al., 2018) but may have an adverse effect on efforts to reduce climate change.  Ideally, abatement policy aimed at decreasing atmospheric pollutant levels would take into account both climate and health impacts.

### 4.4 Comparison with other trends and causality

The current study has focused on surface in situ aerosol optical properties at point locations, primarily in North America and Europe, but also in Asia and Polar Regions. Comparison with reported trends from other long-term measurements of aerosol properties (e.g., surface aerosol mass concentrations, surface chemical mass concentrations, ground-based and satellite column optical properties, etc), can provide a more holistic and global view of changes in the atmospheric aerosol. Model simulations of aerosol trends can also supply insight into global impacts of emission changes. We, thus, present a (non-exhaustive) comparison of the trend results from this study with some other relevant aerosol trend studies in the literature. The supplemental materials of Li et al. (2017) include a summary of trends reported in the literature for AOD, PM2.5 and several aerosol constituents (e.g., sulphate, BC, etc.).

There are some important caveats to keep in mind when comparing aerosol trends across platforms and instruments. First, they represent different aspects of the aerosol (chemical, physical, or optical), at different conditions (dry or ambient), different wavelengths (300-1100 nm), different techniques (in-situ, REM) and different locations (ground-based, airborne or satellite).

Second, there are differences in the statistical methodologies, both in terms of methods used and data treatment. Third, the periods covered often overlap, but are not the same. Further, some REM measurements can only be made under certain conditions (e.g., daylight and cloud-free conditions versus continuous sampling, over land versus over ocean, etc.), meaning temporal coverage may be quite different. Because of all these differences, we only discuss general tendencies rather than absolute values when comparing trends from different studies. Below we first compare our results with trends from other surface in-situ measurements and REM observations. Finally, we discuss causes of the observed trends and speculate specifically on some of the trends in intensive aerosol properties, which have received less attention in the literature than properties related to aerosol loading.

### 4.4.1 Comparison with other surface, in-situ aerosol trends

A comparison of the present day trends derived here to our previous trend ending in 2010 (CC2013) demonstrates that the larger number of stations, particularly in Europe, permits a more detailed view of regional trends. The current wide coverage across continental Europe shows decreasing present-day trends. Decreasing $\sigma_{sp}$, $\sigma_{bsp}$ and $\sigma_{ap}$ trends were confirmed for individual stations (e.g., SMR (Luoma et al., 2019), PAL (Lihavainen et al., 2015b), ARN (Sorribas et al., 2019)), as well as at ACTRIS sites including JFJ, HPB, IPR, IZO, PAL, PUY, SMR and UGR (Pandolfi et al. (2018)).There are some discrepancies in the trends between our current study and Pandolfi et al. (2018) that seem to be principally due to differences in the analyzed periods. Three additional years of data were included in this study and some older periods included in Pandolfi et al. (2018) were invalidated following the evaluations described in Sect. 2.4. The European b and $\mathring{a}_{sp}$ trends computed by Pandolfi et al. (2018) are similar to the results of this study for most of the stations, in that they also found a general ss increase of b and variable $\mathring{a}_{sp}$ trends. In North America the ss decreasing trends in aerosol extensive properties observed in CC2013 are found to have continued in this work with the extended data sets. These results are confirmed by the two other trend studies for in-situ aerosol optical properties in North America. While the methodology and time period of Sherman et al. (2015) were different, the sign and ss of their $\sigma_{sp}$, b,and $\mathring{a}_{sp}$ trends for BND and SGP were the same as reported here. White et al. (2016) found a decreasing trend in absorption coefficient (estimated from light transmittance measurements on 24 h filter samples) at 110 IMPROVE stations for the 2003-2014 period. SPO $\sigma_{sp}$, b and $\mathring{a}_{sp}$ trends for the 1979-2014 period (Sheridan et al., 2016) do agree with CC2013 results, whereas the 1979-2018 trends reported in this study suggest an evolution towards more ss positive trends. The very low aerosol concentrations in Antarctica and the difference in the MK algorithm could however also explain the differences amongst these three analyses.

There have been multiple trends studies on carbon species (also referred to as black carbon (BC), elemental carbon, equivalent black carbon, brown carbon or other terms) which is closely related to aerosol absorption. A decreasing trend in BC concentration is found in Europe (Singh et al., 2018, Kutzner et al., 2018, Grange et al., 2019) related primarily to traffic emission decreases rather than changes in wood burning and/or industrial emissions. Similarly, Lyamani et al. (2011) noted a decrease in BC in southern Spain due to the 2008 economic crisis. In contrast, Davuliene et al. (2019) reported an increasing trend in equivalent black carbon (eBC) for the Arctic site of TIK. In North America, White et al. (2016) found that the decreasing elemental carbon trend at IMPROVE sites was larger than the aerosol absorption trend at the same sites due to the impact of Fe content in mineral dust.  BC trends in the Arctic have been extensively studied (e.g., AMAP, 2015; Sharma et al., 2019; and references therein) and suggest a decreasing trend. This is consistent with our general trend in absorption for polar regions (Table 4), although for individual stations most trends were statistically insignificant.

Particulate mass (PM) and visibility are other metrics for atmospheric aerosol loading that can be most readily compared with our trends in aerosol scattering. Tørseth et al. (2012) detailed decreases in PM across Europe while Hand et al. (2014, 2019) report significant decreases in PM2.5 mass across the US with larger trends in eastern than in western US. Both these trends were also confirmed by the PM trend analysis in Mortier et al. (2020) and are consistent with our

reported scattering trends. Li et al. (2016) used visibility to assess trends in atmospheric haze and aerosol extinction coefficient around the world. The time delay in when the trends switch sign between North America (late 1970s), Europe (early 1980s) and China (mid 2000s) correlates with $SO_2$ trends and the trend differences between eastern and western part of US and Europe are consistent with what is presented in our study.

Many atmospheric aerosols are formed in the atmosphere rather than being directly emitted, so understanding trends in aerosol precursors is also relevant for understanding changes in the atmospheric aerosol. Our study found similar results for scattering as have been found for sulphate trends (Aas et al., 2019), i.e., decreasing sulphate trends across Europe and the US, albeit with the sulphate decrease in Europe beginning before the decrease was observed in the US. Aas et al (2019) also describe potential increases in sulphate in India and increases followed by decreases in SE Asia. Vestreng et al. (2007) monitored the sulphur dioxide emission reduction in Europe and concluded that $SO_2$ emission reductions were largest in the 1990s with a first decrease in Western Europe in the 1980s followed by a large decrease in Eastern Europe in the 1990s. Similarly Crippa et al. (2016) simulated a larger impact of policy reduction in Western than in Eastern Europe for $NO_x$, CO, $PM_{10}$ and BC between 1970 and 2010. Likewise, Huang et al. (2017) simulated the non-methane volatile organic compounds emissions and found a rapid decrease in Europe and in North America since the 1990s, whereas the emission of Africa and Asia clearly increased between 1970 and 2012.

### 4.4.2 Comparison with remote sensing trends

A significant advantage of many REM platforms is their global coverage. Satellites often provide coverage over both land and ocean and the major ground-based REM network AERONET (Holben et al., 1998) is more globally representative than the sites used in this study. However, there are some inherent limitations in comparing aerosol optical property trends from REM retrievals with surface in-situ trends. Our study used aerosol optical measurements made at low RH (typically RH<40%) at the surface, while column aerosol optical retrievals are made at ambient conditions and represent the atmospheric column including layers aloft. Only in the situation of a well-mixed atmosphere, will it be reasonable to compare trends in surface in-situ optical properties with those obtained by ground-based or satellite retrievals. It has also to be mentioned that satellite measurements are less sensitive to the near ground layers containing the greatest aerosol load. Thus, while our trends can be compared with those for column aerosol properties, there is no reason to expect them to be in complete agreement. Below we discuss trends in PM, AOD, column $\sigma_{ap}$ and column SSA.

Satellites have been used to assess the decreasing PM trends in North America and Europe and also to estimate PM trends in other regions with sparse surface measurements. For example, Nam et al. (2017) evaluated the trend in satellite-derived PM10 over Asia and reported mixed annual trend values depending on the subregion they looked at. Li et al. (2017) found satellite-derived PM2.5 to continuously increase in some parts of Asia (e.g., in India) for the 1989-2013 period - we also find an increasing trend (for aerosol absorption) at the one site we studied in India (MUK). For China, Li et al. (2017) report that the PM2.5 trend transitions from an increasing to a decreasing trend with the transition occurring in the 2006-2008 time period similar to the sulphate trend pattern reported by Aas et al. (2019). The in-situ measurements from China (WLG) and Taiwan (LLN) used in our study are not long enough to detect this transition.

Multiple ground-based REM studies (e.g., Yoon et al., 2016, Wei et al., 2019, Mortier et al., 2020,) report decreasing trends in AOD over the US and Europe with larger decreasing trends over Europe than over the US, which is the case in our study (see Table 4) as well. The lack of measurements in many regions similar to the lack of representativeness in the surface in-situ aerosol sites discussed in this study (Asia, Africa, South America, etc) are also emphasized. Ningombam et al. (2019) analyse AOD 1995-2018 trends from 53 remote and high altitude sites, of which 21 had ss negative trends. Regionally, Ningombam found primarily negative trends at sites in the US, Europe and polar regions. Their findings for sites in China and India suggested

mixed trends with some being positive and some negative in those regions. Some of the sites in Ningombam et al. (2019) were also involved in our study. The trends they find for AOD at LLN and MLO are similar to ours (i.e., not ss trends) at SPO (i.e., ss increasing) and at SGP (i.e., decreasing (note: they refer to SGP as 'car')). Their results are different for IZO (we found no ss trends for scattering while they reported ss decreasing AOD) and at BRW and BIR (we found ss decreasing scattering trends but they found not ss AOD trends).

Satellite retrievals can offer an even more global picture of aerosol trends than the surface based REM data. Various satellite trend analyses present a picture of trends in aerosol optical depth for different regions of the world that is quite consistent across satellite (and ground-based) AOD datasets. For example, for the satellite literature that we surveyed, all found decreases in AOD over the US and Europe (e.g., Hsu et al., 2012, Mehta et al., 2016, Zhao et al., 2017, Alfaro-Contreras et al., 2017, Wei et al., 2019) consistent with what we have reported for the AOD from ground-based, REM instruments. As we note above, this is also consistent with surface in-situ scattering trends.

There are some discrepancies in the various satellite derived AOD trends over Asia that are likely due to differences in time period of analysis, trend methodology, regional definitions and/or perhaps satellite data product. Nam et al. (2017) found AOD trends varied depending on what part of Asia was being evaluated. Zhao et al. (2017) reported an increasing then decreasing trend over China, which was also suggested by others (e.g., Sogachova et al., 2019; Alfaro-Contreras et al., 2017). Wei et al. (2019) found a slightly negative but statistically insignificant AOD trend for China. Our study found statistically insignificant trends in aerosol loading for both the high-altitude surface site in China (WLG) and in Taiwan (LLN), perhaps because measurements at both these sites span the AOD increase/decrease periods mentioned by Zhao et al. (2017). Over India, increasing trends in satellite AOD were reported by all the literature we surveyed (e.g., Wei et al. 2019; Mehta et al., 2016; Hsu et al., 2012; Alfara-Contreras et al., 2017). This is consistent with our finding of an increasing trend for aerosol absorption for the one Indian site (MUK) in our study.

The satellite measurements also enable evaluation of aerosol loading changes in regions with few to none long-term surface in-situ aerosol optical property measurements. The Middle East exhibited an increasing trend in AOD, while South America exhibited variable trends (e.g., Wei et al., 2019; Metha et al., 2016, Hsu et al., 2012; Alfaro-Contreras et al., 2017). Wei et al. (2019) found a statistically insignificant trend in South America and suggested it was due to complex and changing aerosol sources. Mehta et al. (2016) looked specifically at Brazil and found a decreasing annual AOD trend, but an increasing AOD trend in springtime. Decreasing AOD trends were found over central Africa (Wei et al., 2019), over the African deserts (Metha et al., 2016) and on African coasts (Alfaro-Contreras et al., 2017) regardless if they are dominated by smoke aerosols (southwest) or dust (northwest).

In addition to AOD, trends for other column aerosol property such as column $\sigma_{ap}$ and column SSA can be considered. While there appear to be many investigations focusing on trends in column aerosol properties other than AOD at individual sites, there are only a few papers that take a more global, multi-site approach (e.g., Li et al., 2014; Zhao et al., 2017; Mortier et al., 2020). There have been several studies related to changes in column $\sigma_{ap}$ using AERONET REM retrievals. For example, Li et al. (2014) suggest an increase in column $\sigma_{ap}$ over the US and a decrease over Europe and at most sites in Asia. More recently, Mortier et al. (2020) found ss decreasing $\sigma_{ap}$ trends in Europe, North America and ss increasing $\sigma_{ap}$ trends in Asia and Africa. Zhao et al. (2017) used satellite retrievals and reported decreasing trends of column $\sigma_{ap}$ over both the US and Europe and a not ss column $\sigma_{ap}$ trend over China. Nam et al. (2018) suggested there was an increasing trend in column extinction Ångström exponent over Asia based on satellite observations. These findings are mostly consistent with our results (Table 4) which indicated decreasing $å_{sp}$ trends in the US and Europe, but perhaps an increasing trend in Asia.

Comparisons of in-situ and column $\omega_0$ trends are more fraught, because, in addition to the above mentioned caveats related to comparing surface and column measurements, column $\omega_0$ can only be obtained from REM techniques under higher aerosol loading conditions. For example, Kahn and Gaitley (2015) indicate that MISR SSA retrieval requires AOD>0.15-0.2. Similarly, AERONET

retrievals require AOD (at 440 nm) > 0.4 (Dubovik et al., 2000). This limits the sites for which column SSA can be retrieved. Andrews et al. (2017) present a plot derived from global model simulations suggesting more than 80% of the globe has annual AOD values below 0.2, and, indeed, many of the surface in-situ sites discussed here are in remote locations with annual AOD consistently below 0.2. Andrews et al. (2017) also suggest there is a systematic variability of SSA with loading that might result in column SSA biases if retrievals are constrained to higher levels of AOD. With these caveats in mind, we can compare our surface $\omega_0$ trend results with satellite column $\omega_0$ trends.

Li et al. (2014) studied 2000-2013 trends in column $\omega_0$ at select AERONET sites. Their findings suggest that column $\omega_0$ is increasing in the US, Europe and Asia. However, they noted the uncertainty in these trends is high because they used level 1.5 data (AOD<0.4) in order to have enough data points for their analysis. Zhao et al. (2017) utilized satellite retrievals and reported decreasing trends in column $\omega_0$ over the eastern US and Europe and a not statistically significant trend over China for the 2001-2015 period. Their results over the US and western Europe are consistent with the overall regional $\omega_0$ trends reported in this study (i.e., Table 4), although Figure 7 suggests there is a fair amount of variability in the surface $\omega_0$ trends at the individual sites in these two regions. Our study found an increasing trend in $\omega_0$ at the surface in Asia (based on 3 sites), which is consistent with Li et al.'s column $\omega_0$ trend but not with the lack of trend in column $\omega_0$ over China suggested by Zhao et al. (2017). But, as noted above, remote sensing retrievals of column SSA should be considered with caution and, clearly, further effort in column SSA trend analysis is warranted.

### 4.4.3 Causality

While it is beyond the scope of this effort to explore in depth the causes of the observed trends of aerosol optical properties, some general comments can be made. First, tendencies in regional trends for variables representing aerosol loading (e.g., surface in-situ aerosol scattering, PM, and AOD) are generally consistent across multiple datasets. Overall, the main cause of observed decreasing trends in loading is likely strong reduction of both primary aerosols and precursors of secondary aerosol formation connected to mitigation strategies on regional to continental scales (e.g., Huang et al., 2017; Crippa et al., 2016; Pandolfi et al., 2016; Vestreng et al., 2007). Detailed analysis of PM reductions and composition changes in Europe and the US have enabled attribution of the trends to changes in source types and emission levels (e.g., Hand et al., 2019; Pandolfi et al., 2016, Ealo et al., 2018).

The explanations of the trends based on long-term measurements are supported by modelling efforts. Like many satellite retrievals, model simulations also provide global coverage and, in addition, can be used to investigate reasons for observed changes in aerosol. Model simulations described in Li et al. (2017) suggested that the decrease in PM2.5 in western and central Europe is principally due to sulfate, whereas in eastern Europe decreases in organic aerosol also plays a role. The EMEP status report (2019) notes that the difference in emissions trends between western and eastern Europe has become more significant since 2010. Further, the EMEP status report suggests that estimated increasing emissions of all pollutants since 2000 in the eastern Europe are mainly influenced by emission estimates for the remaining Asian areas in the EMEP modelled domain. Similarly, Zhao et al. (2019) used a model to attribute the AOD, $\omega_0$ and $\mathring{a}_{sp}$ decreases in North America and Europe to considerable emission reductions in all major pollutants except in mineral dust and ammonia.

For Asia, modelling by Li et al. (2017) suggests aerosol changes are principally related to increases in organic aerosol and secondary inorganic aerosol, whereas the increases in BC, nitrate and ammonium are comparably moderate. Yoon et al. (2016) use a model to ascribe the observed increases in AOD over India to increases in BC and water soluble materials - both related to anthropogenic emissions. Over China, Yoon et al. (2016) observe a disconnect between the model chemical composition and the measured AOD which they explain by noting that the measurement sites they rely on in the region are far from the population centers where most of the emissions occur. Zhao et al. (2019) use a model to attribute the increase in AOD followed by

a decrease in AOD to emission increases induced by rapid economic development until 2008-2009 followed by decreases in both anthropogenic primary aerosols and aerosol precursor gases.

Zhao et al. (2017) suggest that the larger reductions in aerosol precursors (e.g., $SO_2$ and $NO_x$ emissions) rather than primary aerosols, including mineral dust and black carbon can explain the decreases in $\omega_0$ and $\mathring{a}_{sp}$ observed over Europe and the US. This is because the secondary aerosols formed from such precursors tend to be primarily scattering, so less secondary aerosol would change the relative balance between scattering and absorption driving $\omega_0$ down. Similarly, secondary aerosol particles tend to be small so a decreasing trend in secondary aerosol would change the relative contribution of small to large particles in the aerosol size distribution and lead to a decreasing trend in $\mathring{a}_{sp}$. In contrast, in Asia simultaneous increases in aerosol precursors and BC before 2006, and a simultaneous decrease after 2011 explains the trends $\omega_0$ and $\mathring{a}_{sp}$ they observed there. Modifications in emissions of aerosol precursors also impact the atmospheric chemistry leading to non-linear response of the formation of secondary inorganic aerosol (Banzhaf et al., 2015).

While regional changes in emissions are one driving factor in trends, because of long-range transport, out of region changes in sources also have the potential to affect trends. For example, Saharan dust impacts CPR, IZO and UGR (e.g., Denjean et al., 2016; Rodriguez et al., 2011; Garcia et al., 2017; Lyamani et al., 2008) and its emissions may change (decrease) in a warmer world (Evan et al., 2016). Other examples of sites clearly impacted by long range transport include IZO impacted by North African pollution due to developing industries (Rodriguez et al., 2011), and the high altitude station of MLO which is impacted by Asian pollution (e.g., Perry et al., 1999). Mountainous stations can also be affected by modifications of the planetary boundary layer or of the continuous aerosol layer heights responding to ground temperature or mesoscale synoptic weather changes (e.g., Collaud Coen et al., 2018 and references therein).

The oscillation in trend sign for several variables at the Arctic sites is potentially caused by the very low aerosol loading, but the Arctic region is changing rapidly and the impact of evolving transport patterns, atmospheric removal processes or local sources cannot be excluded (e.g., Willis et al., 2018) and requires closer study.

While both increasing and decreasing levels of aerosol due to changes in anthropogenic emissions have been observed, the role of non-anthropogenic sources may become more important in the future. For example, climate change also affects soil drought and the positive feedback between drought and wildfires can also affect aerosol optical properties (Hallar et al., 2017, McClure and Jaffe, 2018). The number and intensity of wildfires is increasing in several regions (e.g., Moreira et al., 2020; Turco et al., 2018; Hand et al., 2014). McClure and Jaffe (2018) confirmed an increasing trend of $PM_{2.5}$ 98 percentiles in northwest US due to an increase in wildfires superimposed on the global decrease in anthropogenic emissions. Yoon et al. (2016) also note an increase in extreme AOD events in the western US, which they hypothesize could be due to wildfires. Another example of potential changes in natural aerosol may take place in the Arctic, where decreases in sea ice coverage might play a role in natural aerosol increases in the region (e.g., Willis et al., 2018) (decreases in sea ice coverage may also lead to enhanced anthropogenic emissions due to increased human activity (e.g., Aliabadi et al., 2015)). Whether such changes in natural aerosol emissions lead to observable changes in overall aerosol trends or trends at the extremes of aerosol loading is something to look for in future trend analyses.

Detailed studies at each station are necessary to discriminate between direct causes like changes in anthropogenic emissions, and indirect causes related to general climate changes such as drought, changes in surface albedo, biogenic aerosol concentration, atmospheric chemistry, sea ice coverage or atmospheric circulation patterns. The availability of the homogenized data set from this study will provide a useful tool for these types of analyses.

In order to get a truly global overview of aerosol trends, surface in-situ measurements need to be paired with model simulations and satellite observations. This will enable evaluation of the uncertainty in regional and global trends based on deficiencies in spatial and/or temporal

coverage. Satellites and models are able to fill the gaps in coverage from ground-based measurements, but both rely on surface measurements for ground truth.

## 5. Conclusion and recommendations

This second long-term trend analysis of in-situ aerosol measurements derived from stations with large spatial representation leads to a more coherent picture of aerosol radiative properties around the world. Results from this study provide evidence that the aerosol load has significantly decreased over the last two decades in North America and Europe. The low number of stations in the other continents means global tendencies cannot be assessed and the results are more variable. The mean extensive property trends are decreasing for all parameters ($\sigma_{sp}$, $\sigma_{bsp}$ and $\sigma_{ap}$) and all regions apart from the $\sigma_{sp}$ trend in the South Pacific and in polar regions (see Table 4). These decreases in aerosol burden are assumed to be a direct consequence of decreases in primary particles and particulate precursors such as $SO_2$ and $NO_x$ due to pollution abatement policies. This assumption is supported by trend results for the USA where the inflection point between not ss and ss decreasing $\sigma_{sp}$ 10 y trends consistently occurred over the same time period (2009-2012) for all central and eastern stations. While the annual $\sigma_{ap}$ decrease (-2.5 to -5%/y for the ss trends in all regions) is larger than that for $\sigma_{sp}$, the $\sigma_{ap}$ time series are not long enough to detect the beginning of $\sigma_{ap}$ decreasing 10 y trends.

The single scattering albedo trend analysis - derived for the first time from a large number of stations - has the greatest climatic relevance. The uncertainty of the $\omega_0$ trend is higher than for the other aerosol parameters due to uncertainties in absorption coefficient absolute value. The general picture is nuanced with ss positive trends mostly in Asia and Eastern Europe and ss negative in Western Europe and North America leading to annual positive median trend of 0.02%/y. It appears that the historical abatement policies for gaseous species and primary aerosol particles (e.g., in Western Europe in the 1980s) have resulted in present-day decreasing $\omega_0$ trends in the western hemisphere, whereas more recent regulations (Asia) are leading to increasing $\omega_0$ trends. Again, this suggests it is necessary to consider how regulatory policies designed to improve health and environmental outcomes impact climate change efforts and vice versa.

The backscattering fraction and scattering Ångström exponent trends relate mostly to the average particle size distribution and to the relative concentrations in the accumulation and coarse modes of the size distribution, but the mean refractive index also plays a role. The interpretation of the results for these parameters is less straightforward as, depending on the site, the trends for b and $\mathring{a}_{sp}$ may have the same or opposite signs. The causes of particle size change encompass not only the primary aerosol emission but also the emission of secondary aerosol precursors, the particle chemistry and condensation rate, the hygroscopic growth and the humidity condition during the measurement. In general, the interpretation of b, $\mathring{a}_{sp}$ and $\mathring{a}_{ap}$ trends is more difficult, and the effects of global climate change on aridity, wildfire frequency and intensity, planetary boundary layer height, transportation patterns or natural oscillation must also be investigated in order to find the causality of aerosol optical properties changes.

This study was limited by the lack of information from many WMO regions. Since 2010, the number of stations with time series longer than 10 y has doubled (24 in 2010, 52 currently) so that the spatial coverage is improved and various additional environments are covered in Europe, North America and in polar regions. A first result of this study is that, while aerosol exhibits a very weak spatial and temporal homogeneity, general features can be deduced with the present station density in Europe and North America, while the picture in polar regions is less clear. The few stations in Asia, Africa, South America and in Oceania/Pacific region cannot, however, be considered as representative for their continents/regions, first, because of their small number and, second, because mountainous and coastal environments are overrepresented relative to the continental environment with rural, forest or desert footprints. According to information from the GAWSIS metadata base, more stations located in underrepresented regions are now in operation, which promises a better spatial coverage in a few years, however, sustaining these operations is still an open issue (the longest time series in India closed in 2016) and not all stations are actually providing their data in open access, with the proper associated metadata. Even in developed countries, the financial resources needed to operate long-term monitoring are not always secure,

leading to the closing of stations, to a decrease of time series quality and/or to a delay in data submission to the international data banks.

In this study, a number of datasets were not used or were only partially used due to the occurrence of break points following instrumental or inlet changes or even calibration shifts. High quality data rely on attention to international recommendations for measurements, on a regular maintenance schedule, participation in inter-comparison efforts and on high-level quality control. The existence of metadata, logbooks and station's history is crucial for determining causes of any detected break points and necessary to enable the generation of a final homogenized time series for trend analysis. This homogenization process provides us with an important finding: a critical review of the data by others outside the measurement network is very important in improving the quality of the reported data. This study has resulted in a large improvement to the EBAS database and in the quality of the reported datasets.

Based on the results of this study and with a view toward future trend analyses, the following recommendations concerning the improvement of aerosol optical time series are raised:

- The station history, metadata and logbooks have to be detailed and handled with great care, since they are absolutely necessary to evaluate long-term trends on homogenized time series.

- Time series are affected not only by the instrument type or inlet changes, but also by replacement by instruments of the same type and by shifts in calibrations.

- A rotation between instruments in a network (e.g., to enable repairs) will decrease potential missed data losses but has a potential to increase breakpoints in the time series, particularly in the wavelength dependence of the parameters.

- The scattering and backscattering coefficients and the backscattering fraction and the scattering Ångström exponents are very sensitive to the humidity conditions in the nephelometers due to the hygroscopic growth of particles even at low RH. The nephelometer humidity sensors should be better checked and characterized in order to assess long-term trends of dry particles.

- Long-term trend analysis should not be computed on time series shorter than 10 y, since short datasets lead to a larger probability of false trend detection because of the low number of elements in the time series.

- Stations with long-term records have to be sustained and their funding should be assured in order to study the future impact of aerosol on climate change. Stations maintenance as well as new station creation in regions with a low spatial coverage (Africa, South America, Asia and Oceania) should be particularly encouraged.

*Data availability*: Almost all datasets are available as level 2 NASA/AMES files at EBAS (http://ebas.nilu.no/) at an hourly resolution. The screened datasets used for this study aggregated as daily medians can be found at : https://doi.org/10.21336/c4dy-yw57

*Author contribution*: CLM, YL and EA gathered datasets and applied additional QC to the time series. MCC did a further QC, computed the long-term trends and analysed the results. MCC and EA wrote the manuscript. NB, JH, PL, CLM, MP, and PZ extensively contributed to the revision of the manuscript. All the other co-authors contribute to the measurements of aerosol optical properties at the 52 stations and to the manuscript reviewing.

*Competing interests*: The authors declare that they have no conflict of interest.

**Acknowledgements**

The authors would like to thank the numerous, but unfortunately unnamed, technical and scientific staff members of the stations as well as many students included in these analyses, whose dedication to quality for decades have made this paper possible. Provision of data from this study have mainly been acquired in the framework of NOAA-FAN (https://www.esrl.noaa.gov/gmd/aero/net/); ACTRIS, under the ACTRIS-2 (Aerosols, Clouds, and Trace gases Research InfraStructure) project supported by European Union (grant agreement no. 654109); and IMPROVE (http://vista.cira.colostate.edu/Improve/). Some European sites and measurements were also supported by the Co-operative Programme for Monitoring and Evaluation of the Long-range Transmission of Air pollutants in Europe (EMEP) under UNECE. The authors also gratefully acknowledge the following persons and organizations:

- AMY: the Korea Meteorological Administration Research and Development Program "Development of Monitoring and Analysis Techniques for Atmospheric Composition in Korea" under Grant (KMA2018-00522).
- APP: Appalachian State College of Arts and Sciences, electronics technician Michael Hughes, machinist Dana Greene
- BEO: projects ACTRIS2 and ACTRIS-BG
- BIR: the project no 80026 Arctic Monitoring and Assessment Programme (AMAP) under the EU-action "Black Carbon in the Arctic". Aerosol optical/physical properties at Birkenes II are financed by the Norwegian Environment Agency
- CGO: the Australian Bureau of Meteorology for their long term and continued support of the Cape Grim Baseline Air Pollution Monitoring Station, and all the staff from the Bureau of Meteorology and CSIRO, particularly John Gras who instigated the measurements of aerosol scattering and absorption
- CPR: Para La Naturaleza and the nature reserve of Cabezas de San Juan and the support of grants AGS 0936879 and EAR-1331841.
- GSN: the Basic Science Research Program through the National Research Foundation of Korea (2017R1D1A1B06032548).
- SMR: the European Union Seventh Framework Programme under grant agreement No 262254. This research has also received funding from the European Union's Horizon 2020 research and innovation programme under grant agreement No 654109 via project ACTRIS-2 and grant agreement No 689443 via project iCUPE. The work was also funded by the Academy of Finland (project. No 307331).
- IMPROVE: IMPROVE is a collaborative association of state, tribal, and federal agencies, and international partners. Support for IMPROVE nephelometers comes from the National Park Service. The assumptions, findings, conclusions, judgments, and views presented herein are those of the authors and should not be interpreted as necessarily representing the National Park Service (stations: ACA, BBE, CRG, GBN, GLR, GSM, HGC, MCN, MRN, MZW, NCC, RMN, SCN, SHN).
- IZO: Measurement Programme within the Global Atmospheric Watch (GAW) Programme at the Izaña Atmospheric Research Centre, financed by AEMET.
- JFJ: Urs Baltensperger, Günther Wehrle, Erik Herrmann; the International Foundation High Altitude Research Station Jungfraujoch and Gornergrat (HSFJG), the Swiss contributions (GAW-CH and GAW-CH-Plus) to the Global Atmosphere Watch programme of the World Meteorological Organization (WMO) which are coordinated by MeteoSwiss; in the context of ACTRIS: the European Union's Horizon 2020 research and innovation programme under grant agreement no. 654109 (ACTRIS-2 project) and no. 739530 (ACTRIS-PPP project), as well as the Swiss State Secretariat for Education, Research and Innovation, SERI, under contract number 15.0159-1 (ACTRIS-2 project). The opinions expressed and arguments employed herein do not necessarily reflect the official views of the Swiss Government.
- LLN: the Taiwan Environmental Protection Administration for their continued support of the Lulin Atmospheric Background Station (LABS); the Ministry of Science and Technology for the support to individual PIs' research funding.

- MSY: the European Union's Horizon 2020 research and innovation programme under grant agreement no. 654109, ACTRIS (project no. 262254), ACTRIS-PPP (project no. 739530). Measurements at MSY (Montseny) station was supported by the MINECO (Spanish Ministry of Economy, Industry and Competitiveness) and FEDER funds under the PRISMA project (CGL2012-39623-C02/00) and under the HOUSE project (CGL2016-78594-R), by the MAGRAMA (Spanish Ministry of Agriculture, Food and Environment) and by the Generalitat de Catalunya (AGAUR 2014 SGR33, AGAUR 2017 SGR41 and the DGQA). Marco Pandolfi is funded by a Ramón y Cajal Fellowship (RYC-2013-14036) awarded by the Spanish Ministry of Economy and Competitiveness.

- MUK: the Ministry of Foreign Affairs of Finland, project grants (264242, 268004, 284536, and 287440) received from Academy of Finland; Business Finland and DBT, India sponsored project (2634/31/2015), the Centre of Excellence in Atmospheric Science funded by the Finnish Academy of Sciences (307331), and an esteemed collaboration of FMI and TERI.

- NOAA stations (BND, BRW, MLO, SGP, SPO, SUM, THD): Derek Hageman for all his programming efforts for NOAA and NFAN stations, John Ogren for initiating the expanded NFAN measurements and NOAA's Climate Program Office for funding

- PAY: the Swiss Federal Office for the Environment (FOEN).

- PUY: the staff of OPGC and LaMP, INSU-CNRS and the University Clermont Auvergne, and the financial support from ACTRIS-France National Research infrastructure and CNRS-INSU long-term observing program..

- SGP: the U.S. Department of Energy Atmospheric Radiation Measurement Program via Argonne National Laboratory, the DOE SGP ARM Climate Research Facility staff and scientists.

- TIK: The Aethalometer was supplied by Russ Schnell; Tiksi overall logistics and operations by Taneil Uttal and Sara Morris (NOAA/ESRL/PSD, Boulder, CO, USA).

- UGR: the European Union's Horizon 2020 research and innovation program through project ACTRIS-2 (grant agreement No 654109), the Spanish Ministry of Economy and Competitiveness through projects CGL2016-81092-R, CGL2017-90884-REDT and RTI2018-101154-A-I00.

– WLG: China Meteorological Administration for their continued support to Waliguan Atmospheric Baseline Station; National Scientific Foundation of China (41675129), National Key Project of Ministry of Science and Technology of the People's Republic of China (2016YFC0203305 & 2016YFC0203306), Basic Research Project of Chinese Academy of Meteorological of Sciences (2017Z011). It was also supported by the Innovation Team for Haze-fog Observation and Forecasts of China Meteorological Administration.

- ZEP: the Swedish EPA's (Naturvårdsverket) Environmental monitoring program (Miljöövervakning), the Knut-and-Alice-Wallenberg Foundation within the ACAS project (Arctic Climate Across Scales, project no. 2016.0024), the research engineers Tabea Henning, Ondrej Tesar and Birgitta Noone from ACES and the staff from the Norwegian Polar Institute (NPI), NPI for substantial long-term support in maintaining the measurements, Maria Burgos and Dominic Heslin-Rees (ACES) for preparing the data.

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

**Tables**

**Table 1**: List of observatories included in this study, arranged alphabetically by GAW acronyms, including their names, countries, coordinates and elevation, site environmental characteristic (geographical category and footprint), size cut, type of nephelometer and absorption filter photometer deployed, time period used, and nephelometer RH percentiles.

| GAW code | Station Name | Country | GPS coordinates | Site Characteristics [1] | Size cut [2] | $\sigma_{sp}$ Period [3] | $\sigma_{ap}$ Period [4] | Sample RH 5th;50th;95th percentile |
|---|---|---|---|---|---|---|---|---|
| ACA | Acadia NP[5] | US | 44.38°N, 68.26°W, 122 m | Coast, RB | - | O,1994-2018 | -- | 46;75;98 |
| ALT | Alert | CA | 82.50°N, 62.34°W, 210 m | P, P | PM10 & PM1 | T, 2005-2017 | P, 2005-2014 C, 2014-2017 | 0;4;23 |
| AMY | Anmyeon-do | KR | 36.54°N, 126.33°E, 46 m | Coast, RB | PM10 | T, 2008-2018 | AE16, 2008-2009 AE31, 2010-2018 | 8;26;66 |
| APP | Appalachian | US | 36.21°N, 81.69°W, 1076 m | Con, RB | PM10 & PM1 | T, 2010-2018 | P, 2010-2016 C, 2016-2018 | 0;17;41 |
| BBE | Big Bend NP | US | 29.30°N, 103.18°W, 1052 m | Con, DE | - | O, 1998-2015 | -- | 14;41;78 |
| BEO | Moussala | BG | 42.18°N, 23.59°E, 2925 m | Mt, Mix | TSP | T, 2008-2017 | -- | 4;15;26 |
| BIR | Birkenes | NO | 58.38°N, 8.25°E, 220 m | Con, F | PM10 | T, 2010-2018 | P, 2010-2018 | 11;21;38 |
| BND | Bondville | US | 40.05°N, 88.37°W, 213 m | Con, RB | PM10 & PM1 | T, 1995-2018 | P, 1998-2012 C, 2012-2018 | 5;22;47 |
| BRW | Barrow | US | 71.32°N, 156.61°W, 11 m | Polar, Coast, P | TSP- PM10 &PM1 | R, 1978-1997 T, 1997-2018 | P, 1998-2014 C, 2014-2018 | 0;7;26 |
| CGO | Cape Grim | AU | 40.68°S, 144.69°E, 94 m | Coast, RB | PM10 | E, 2006-2018 | M, 2008-2018 | 0;9;23 |
| CMN | Monte Cimone | IT | 44.17°N, 10.68°E, 2165 m | Mt, Mix | TSP | -- | M, 2008-2018 | 14;34;57 |
| CPR | Cape San Juan | PR | 18.38°N, 65.62°W, 65 m | Coast, F | PM1 & PM10 | T, 2005-2016 | P, 2007-2014 C, 2014-2016 | 31;48;70 |
| CPT | Cape point | ZA | 34.35°S, 18.49°E, 230 m | Coast, Mix | PM1 & PM10 | T, 2006-2014 | P, 2006-2014 | 25;36;51 |
| CRG | Columbia River George | US | 45.66°N, 121.00°W, 178 m | Con, RB | - | O, 1994-2004 | -- | 35;63;92 |
| EGB | Egbert | CA | 44.23°N, 79.78°W, 255 m | Con, RB | PM10 | T, 2010-2018 | P, 2010-2018 | 6;23;60 |
| FKL | Finokalia | GR | 35.34°N, 25.67°E, 150 m | Coast, RB | TSP-PM10 PM10 PM10 | -- | AE21, 2004-2010 AE31, 2011-2014 AE33, 2015-2018 | 29;64;90 |
| GBN | Great Basin NP | US | 39.01°N, 114.22°W, 2065 m | Mt, DE | - | O, 2008-2018 | -- | 14;41;80 |
| GLR | Glacier NP | US | 48.51°N, 114.00°W, 976 m | Con, F | - | O, 2008-2018 | -- | 48;78;95 |
| GSM | Great Smoky Mountain NP | US | 35.63°N, 83.94°W, 810 m | Con, F | - | O, 1994-2018 | -- | 40;73;98 |
| GSN | Gosan | KR | 33.28°N, 126.17°E, 72 m | Coast, RB | TSP-PM10 & PM1 | T, 2008-2015 | AE31, 2008-2015 | 14;30;64 |

| HGC | Grand Canyon NP | US | 35.97°N, 111.98°W, 2267 m | Con, F | - | O, 1998-2018 | -- | 18;45;91 |
|-----|-----------------|-----|---------------------------|--------|---|------------|----|----------|
| HPB | Hohenpeissenberg | DE | 47.80°N, 11.01°E, 985 m | Mt, RB | TSP-PM10 | T, 2006-2017 | M, 2004-2017 | 9;23;44 |
| IPR | Ispra | IT | 45.80°N, 8.63°E, 209 m | Con, U | PM10 | T, 2004-2017 | AE31, 2004-2017 | 6;24;56 |
| IZO | Izana | ES | 28.31°N, 16.50°W, 2373 m | Mt, Mix | PM10 | T, 2009-2018 | M, 2007-2018 | 5;15;33 |
| JFJ | Jungfraujoch | CH | 46.55°N, 7.99°E, 3580 m | Mt, Mix | TSP | T, 1996-2018 | AE31, 2002-2018 | 0;7;16 |
| KPS | K-Pustza | HU | 46.97°N, 19.58°E, 125 m | Con, RB | PM10 | T, 2008-2017 | P, 2008-2012 C, 2012-2018 | 11;24;44 |
| LLN | Lulin | TW | 23.47°N, 120.87°E, 2862 m | Mt, F, Mix | PM10 & PM1 | T, 2009-2018 | P, 2009-2011 C, 2012-2018 | 5;16;44 |
| MCN | Mammoth Caves NP | US | 37.13°N, 86.15°W, 235 m | Con, RB | - | O, 1993-2018 | -- | 48;78;99 |
| MEL | Melpitz | DE | 51.53°N, 12.93°E, 86 m | Con, U | PM10 | T, 2007-2017 | M, 2007-2017 | 7;20;34 |
| MLO | Mauna Loa | US | 19.54°N, 155.58°W, 3397 m | Mt, Mix | TSP-PM10 & PM1 | T, 2000-2018 | MRI, 1988-1999 P, 2000-2013 C, 2013-2018 | 0;6;18 |
| MRN | Mount Rainier NP | US | 46.76°N, 122.12°W, 439 m | Con, F | - | O, 1993-2018 | -- | 68;92;100 |
| MSY | Montseny | ES | 41.78°N, 2.36°E, 700 m | Mt, RB | PM10 | E, 2010-2018 | M, 2009-2018 | 15;26;43 |
| MUK | Mukteshwar | IN | 29.44°N, 79.62°E, 2180 m | Mt, Mix | PM2.5-PM10 | E, 2006-2013 | AE31, 2006-2015 | 2;7;15 |
| MZW | Mt. Zirkel Wilderness | US | 40.54°N, 106.68°W, 3243 m | Mt, F | - | O, 1994-2008 | -- | 28;65;92 |
| NCC | National Capitol Central | US | 38.90°N, 77.04°W, 514 m | Con, U | - | O, 2004-2015 | -- | 38;63;90 |
| NMY | Neumayer | DE | 70.67°S, 8.27°W, 42 m | Polar, Coast, Mix | PM10 | T, 2009-2018 | M, 2007-2018 | 0;2;10 |
| PAL | Pallas | FI | 67.97°N, 24.12°E, 560 m | Polar, Pristine | PM5-PM2.5-PM10 | T, 2000-2018 | M, 2008-2018 | 4;12;32 |
| PAY | Payerne | CH | 46.81°N, 6.94°E, 490 m | Con, RB | PM2.5 | -- | AE31, 2009-2018 | ?? |
| PUY | Puy de Dôme | FR | 45.77°N, 2.97°E, 1465 m | Mt, Mix | TSP | T, 2009-2018 | M, 2009-2017 | 10;26;49 |
| RMN | Rocky Mountain NP | US | 40.28°N, 105.55°W, 2760 m | Mt, RB | - | O, 2008-2018 | -- | 21;48;88 |
| SCN | Sycamore Canyon | US | 35.14°N, 111.97°W, 2046 m | Con, F | - | O, 1999-2009 | -- | 17;50;97 |
| SGP | Southern Great Plains | US | 36.60°N, 97.50°W, 318 m | Con, RB | PM10 & PM1 | T, 1997-2017 | P, 2007-2017 | 5;25;53 |
| SHN | Shenandoah | US | 38.52°N, 78.44°W, 1074 m | Con, F | - | O, 1997-2018 | -- | 41;78;100 |
| SMR | Hyytiala | FI | 61.85°N, 24.29°E, 181 m | Con, F | TSP-PM10 | T, 2007-2017 | AE31, 2007-2017 | 4;14;47 |
| SPO | South Pole | US | 90.00°S, 24.80°W, 2841 m | Polar, P | TSP | T, 1979-2018 | -- | 0;0;0 |
| SUM | Summit | DK | 72.58°N, 38.48°W, 3238 m | Polar, P | TSP | -- | AE16, 2006-2016 C, 2016-2018 | 0;0;5 |
| THD | Trinidad Head | US | 41.05°N, 124.15°W, 107 m | Coast, RB | PM10 & PM1 | T, 2003-2016 | P, 2003-2013 C, 2013-2016 | 16;27;38 |
| TIK | Tiksi | RU | 71.59°N, 128.92°E, 8 m | Polar, Coast, RG | PM10 | -- | AE31, 2010-2018 | ?? |
| UGR | Granada | ES | 37.16°N, 3.61°W, 680 m | Con, U | TSP | T, 2006-2018 | M, 2006-2018 | 14;29;46 |
| WLG | Mount Waliguan | CN | 36.29°N, 100.90°E, 3810 m | Mt, Mix | PM10 & PM1 | T, 2008-2018 | P, 2008-2018 | 0;8;23 |
| ZEP | Zeppelin Mountain | NO | 78.91°N, 11.89°E, 475 m | Polar, Mt, P | PM10 | T, 2005-2016 | P, 2005-2018 AE31 | 0;7;17 |
| ZSF | Zugspitze-Schneefernerhaus | DE | 47.42°N, 10.58°E, 2671 m | Mt, Mix | TSP & PM10 | -- | M, 2009-2018 | 4;13;24 |

[1] *Geographical category:* Mountain=**Mt**, Polar=**P**, Continental=**Con**, Coastal =**Coast**

   *Footprint:* Rural background=**RB**, Forest=**F**, Desert=**DE**, (Sub-)Urban=**U,** Pristine**= P** Mixed: **Mix**

[2] the mention of two size cuts separated by "-" corresponds to a modification of inlet during the tim series,  whereas the "&" corresponds to measurements at two size cuts.

[*3] T=TSI nephelometer, O=Optec nephelometer, R=Radiance Research nephelometer;E3=Ecotech nephelometer Aurora 3000, E4=Ecotech nephelometer Aurora 4000

[*4] AE16($C_{ref}$ =1.8)/AE22 ($C_{ref}$ =1.8)/AE31 ($C_{ref}$ =3.5)/AE33 ($C_{ref}$ =3.5)=Aethalometer  , P1=1-wavelength or P3=3-wavelength PSAP, M=MAAP, C=NOAA CLAP , ET=ES95L Thermo 5012==M

**Table 2**: MK trends for all parameters in units/y for the last 10 y, 15 y and 20 y of measurements ending in 2016-2018. The ss trends are given in bold. Results in %/y are given in Table S1.

| Station | $\sigma_{sp}$ 10 y | 15 y | 20 y | $\sigma_{bsp}$ 10 y | 15 y | 20 y | $\sigma_{ap}$ 10 y | 15 y | 20 y | $\omega_{o}$ 10 y | 15 y | 20 y | b 10 y | 15 y | 20 y | $\mathring{a}_{sp}$ 10 y | 15 y | 20 y | $\mathring{a}_{ap}$ 10 y | 15 y |
|---|---|---|---|---|---|---|---|---|---|---|---|---|---|---|---|---|---|---|---|---|
| **Africa** | | | | | | | | | | | | | | | | | | | | |
| IZO | -0.106 | | | -0.009 | | | -0.008 | | | -0.000 | | | **0.000** | | | -0.001 | | | | |
| **Asia** | | | | | | | | | | | | | | | | | | | | |
| AMY | -0.741 | | | -0.068 | | | -0.007 | | | **-0.000** | | | 0.000 | | | **0.012** | | | | |
| LLN | -0.109 | | | -0.010 | | | **-0.049** | | | **0.003** | | | **0.001** | | | **0.018** | | | **0.004** | |
| WLG | -0.428 | | | 0.017 | | | **-0.057** | | | **0.000** | | | **0.002** | | | 0.011 | | | **0.024** | |
| **Europe** | | | | | | | | | | | | | | | | | | | | |
| BEO | -0.052 | | | 0.023 | | | | | | | | | **-0.001** | | | **-0.019** | | | | |
| BIR | **-0.144** | | | **-0.020** | | | **-0.020** | | | 0.001 | | | 0.000 | | | **-0.013** | | | | |
| CMN | | | | | | | -0.011 | | | | | | | | | | | | | |
| FKL | | | | | | | -0.000 | **-0.001** | | | | | | | | | | | | |
| HPB | **-0.414** | | | **-0.047** | | | **-0.069** | | | **0.000** | | | | | | | | | | |
| SMR | **-0.193** | | | **-0.022** | | | **-0.038** | | | **0.002** | | | **0.001** | | | **0.011** | | | **-0.003** | |
| IPR | **-2.454** | | | **-0.317** | | | **-0.124** | | | **-0.006** | | | **0.001** | | | **-0.007** | | | 0.001 | |
| JFJ | **-0.092** | **-0.062** | **-0.031** | **-0.007** | **-0.006** | **-0.004** | **-0.011** | **-0.004** | | **-0.002** | **-0.001** | | | | | **-0.029** | **-0.008** | 0.004 | **-0.007** | **-0.006** |
| KPS | -0.285 | | | -0.023 | | | -0.019 | | | **0.001** | | | -0.000 | | | 0.000 | | | | |
| MPZ | **-1.015** | | | **-0.145** | | | -0.121 | | | 0.000 | | | -0.000 | | | 0.004 | | | | |
| MSY | **-1.155** | | | **-0.095** | | | -0.027 | | | **-0.003** | | | **0.002** | | | 0.004 | | | | |
| PAL | **0.064** | 0.013 | | **0.012** | 0.003 | | -0.004 | | | **0.002** | | | 0.000 | 0.000 | | 0.007 | 0.000 | | | |
| PAY | | | | | | | -0.235 | | | | | | | | | | | | | |
| PUY | -0.147 | | | -0.012 | | | **-0.017** | | | **0.002** | | | **0.002** | | | **-0.021** | | | | |
| UGR | 0.330 | | | 0.062 | | | **-0.031** | | | **0.001** | | | **0.001** | | | **0.008** | | | | |
| ZSF | | | | | | | -0.036 | | | | | | | | | | | | | |
| **North America** | | | | | | | | | | | | | | | | | | | | |
| ACA | **-0.522** | **-0.301** | **-0.267** | | | | | | | | | | | | | | | | | |
| APP | **-0.627** | | | **-0.074** | | | **-0.092** | | | **0.000** | | | **0.001** | | | **-0.008** | | | -0.000 | |
| BND | **-0.787** | **-0.526** | **-0.413** | **-0.107** | **-0.071** | **-0.056** | **-0.055** | **-0.065** | **-0.025** | **-0.000** | **-0.000** | **-0.000** | 0.000 | **0.000** | **0.000** | **0.006** | 0.001 | **0.003** | | |
| CPR | **0.394** | | | **0.037** | | | **-0.010** | | | **0.001** | | | **-0.000** | | | **-0.018** | | | **0.088** | |

| | | | | | | | | | | | | | | | | | | | |
|---|---|---|---|---|---|---|---|---|---|---|---|---|---|---|---|---|---|---|---|
| EGB | 0.093 | | | **0.041** | | | 0.008 | | | 0.000 | | | **0.003** | | | **-0.027** | **-0.022** | | |
| GBN | **-0.168** | | | | | | | | | | | | | | | | | | |
| GLR | 0.147 | | | | | | | | | | | | | | | | | | |
| HGC | **-0.152** | **-0.158** | **-0.061** | | | | | | | | | | | | | | | | |
| MCN | **-1.321** | **-1.161** | **-0.821** | | | | | | | | | | | | | | | | |
| MRN | -0.026 | **-0.104** | **-0.208** | | | | | | | | | | | | | | | | |
| RMN | **-0.011** | | | | | | | | | | | | | | | | | | |
| SGP | **-0.294** | **-0.299** | **-0.318** | **-0.036** | **-0.047** | **-0.036** | -0.009 | | | **-0.001** | | | **0.001** | 0.000 | 0.000 | -0.008 | -0.007 | -0.006 | 0.017 |
| SHN | **-0.712** | **-0.673** | **-0.539** | | | | | | | | | | | | | | | | |
| THD | **-0.636** | | | **-0.071** | | | -0.006 | | | **-0.000** | | | 0.000 | | | 0.013 | | | 0.003 |
| **South Pacific** | | | | | | | | | | | | | | | | | | | |
| CGO | **0.124** | | | | | | 0.000 | | | 0.000 | | | | | | | | | |
| MLO | **-0.015** | 0.000 | **0.003** | -0.001 | 0.001 | | **-0.003** | 0.001 | | **0.002** | **-0.002** | | **0.004** | 0.001 | | 0.004 | **-0.019** | **-0.010** | **0.081** |
| **Polar regions** | | | | | | | | | | | | | | | | | | | |
| ALT | **-0.005** | | | 0.000 | | | 0.002 | | | **0.001** | | | **0.002** | | | **0.012** | | | **0.008** |
| BRW | **-0.241** | -0.068 | **-0.054** | **-0.016** | -0.004 | -0.003 | **-0.007** | -0.001 | -0.002 | 0.000 | -0.000 | -0.000 | **0.002** | 0.001 | 0.000 | 0.010 | 0.004 | **-0.002** | -0.004 |
| NMY | 0.008 | | | -0.001 | | | **-0.000** | | | 0.000 | | | -0.001 | | | -0.004 | | | |
| SPO | **0.006** | 0.000 | **0.004** | **0.003** | 0.000 | | | | | | | | **0.006** | 0.000 | | -0.019 | **-0.023** | **-0.028** | |
| SUM | | | | | | | -0.000 | 0.000 | | | | | | | | | | | |
| TIK | | | | | | | **-0.006** | | | | | | | | | | | | -0.002 |
| ZEP | **0.050** | | | **0.007** | | | -0.000 | | | **0.001** | | | **0.000** | 0.000 | | -0.018 | -0.003 | -0.018 | 0.003 |

**Table 3**: Number of trends analyzed for each parameters, of ss cases for each trend analysis methods, of trends with similar statistical significance in MK, GLS/day and LMS/log, of trends with similar statistical significance for all the five methods, of trends with at least GLS/day or LMS/log ss similar to MK, of trends with none agreement between these two methods and MK ss.

| Number of | $\sigma_{sp}$ | $\sigma_{bsp}$ | $\sigma_{ap}$ | $\omega_0$ | b | $å_{sp}$ | $å_{ap}$ |
|---|---|---|---|---|---|---|---|
| Time series | 37 | 28 | 33 | 27 | 26 | 27 | 14 |
| ss MK | 25 | 17 | 21 | 20 | 19 | 19 | 9 |
| ss GLS/day | 27 | 19 | 24 | 22 | 20 | 21 | 7 |
| ss GLS/month | 22 | 16 | 21 | 12 | 17 | 12 | 9 |
| ss LMS/log | 25 | 18 | 17 | 12 | 17 | 14 | 5 |
| ss LMS/lin | 22 | 17 | 21 | 12 | 17 | 13 | 7 |
| MK, GLS/day and LMS/log identical | 30 | 24 | 23 | 12 | 17 | 19 | 8 |
| all 5 methods identical | 27 | 23 | 21 | 10 | 15 | 16 | 6 |
| MK+ GLS/day or LMS/log identical | 4 | 3 | 7 | 12 | 7 | 7 | 2 |
| MK different from GLS/day or LMS/log | 3 | 1 | 3 | 3 | 2 | 1 | 4 |

**Table 4**: Overview of the aerosol optical properties decadal MK trends ending between 2016-2018 for all the stations and per continent/region of the world.

| Regions (nb stations/nb ss trends) | Mean trend for all stations [%/y] (std) | Mean ss trend [%/y] (std) |
|---|---|---|
| Scattering coefficient | | |
| all (37/25) | -2.19 (3.20) | -2.80 (3.53) |
| Africa (1/0) | -4.6 | |
| Asia (3/0) | -1.79 (0.16) | |
| Europe (12/8) | -3.23 (3.32) | -4.32 (3.44) |
| N.-America (14/11) | -2.54 (2.74) | -3.41 (2.36) |
| Pacific (2/2) | 0.73 (4.01) | 0.73 (4.01) |
| Polar regions (5/4) | 0.30 (4.00) | 0.13 (4.60) |
| Backscattering coefficient | | |
| all (28/17) | -0.97 (3.42) | -1.46 (4.16) |
| Africa (1/0) | -2.31 | |
| Asia (3/0) | -0.69 (1.04) | |
| Europe (12/8) | -1.82 (3.43) | -3.12 (3.2) |
| N. America (6/6) | -1.32 (3.33) | -1.32 (3.33) |
| Pacific (1/0) | -0.66 | |
| Polar regions (5/3) | 1.52 (4.64) | 2.7 (6.14) |
| Absorption coefficient | | |
| all (33/21) | -3.05 (3.26) | -4.42 (3.09) |
| Africa (1/0) | -3.84 | |
| Asia (3/2) | -3.66 (3.63) | -5.35 (3.02) |
| Europe (15/12) | -3.87 (3.34) | -4.48 (3.38) |
| N. America (6/3) | -2.23 (2.14) | -3.77 (1.37) |
| Pacific (2/1) | -1.91 (3.79) | -4.6 |
| Polar regions (6/3) | -1.73 (4.29) | -4.19 (4.97) |
| Single scattering albedo | | |
| all (27/20) | 0.02 (0.28) | 0.01 (0.32) |
| Africa (1/0) | -0.02 | |
| Asia (3/3) | 0.13 (0.25) | 0.13 (0.25) |
| Europe (11/9) | -0.03 (0.41) | -0.06 (0.45) |
| N. America (6/5) | 0.00 (0.13) | -0.03 (0.14) |
| Pacific (2/1) | 0.14 (0.18) | 0.27 |

| | | |
|---|---|---|
| Polar regions (4/2) | 0.07 (0.076) | 0.12 (0.00) |
| Backscattering fraction | | |
| all (26/19) | 1.02 (1.46) | 1.39 (1.54) |
| Africa (1/1) | 0.41 | 0.41 |
| Asia (3/2) | 1.23 (0.70) | 1.06 (0.36) |
| Europe (10/6) | 0.49 (0.99) | 0.81 (1.18) |
| N. America (6/5) | 0.82 (0.98) | 0.95 (1.03) |
| Pacific (1/1) | 3.40 | 3.40 |
| Polar regions (5/4) | 1.82 (2.57) | 2.42 (2.52) |
| Scattering Ångström exponent | | |
| all (27/19) | -0.21 (1.71) | -0.32 (1.95) |
| Africa (1/0) | -0.17 | |
| Asia (3/2) | 1.37 (0.18) | 1.28 (0.13) |
| Europe (11/8) | -0.23 (0.94) | -0.45 (1.02) |
| N. America (6/6) | -1.03 (2.98) | -1.03 (2.98) |
| Pacific (1/0) | 0.36 | |
| Polar regions (5/3) | -0.22 (1.62) | 0.39 (1.89) |
| Absorption Ångström exponent | | |
| all (14/9) | 1.26 (2.42) | 2.01 (2.78) |
| Asia (2/2) | 1.37 (1.31) | 1.37 (1.31) |
| Europe (3/2) | -0.35 (0.43) | -0.59 (0.21) |
| N. America (4/2) | 2.20 (3.12) | 4.26 (3.49) |
| Pacific (1/1) | 6.48 | 6.48 |
| Polar regions (4/2) | 0.16 (0.73) | 0.74 (0.50) |

## Figures

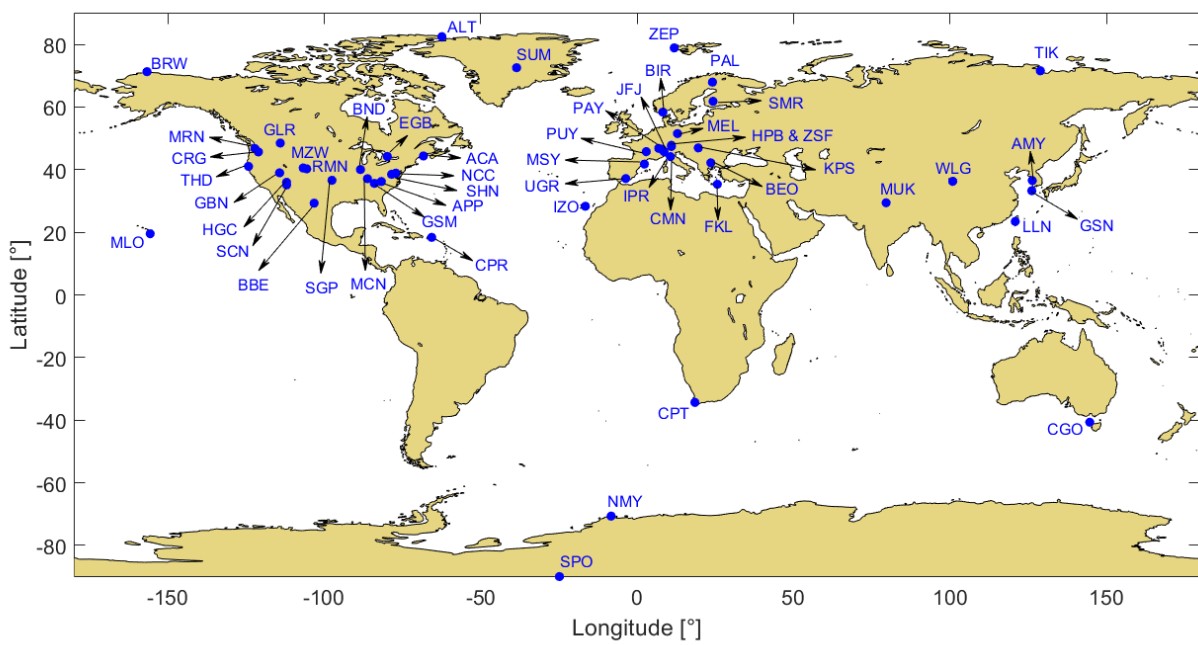

**Figure 1**: Map of stations with their GAW acronyms

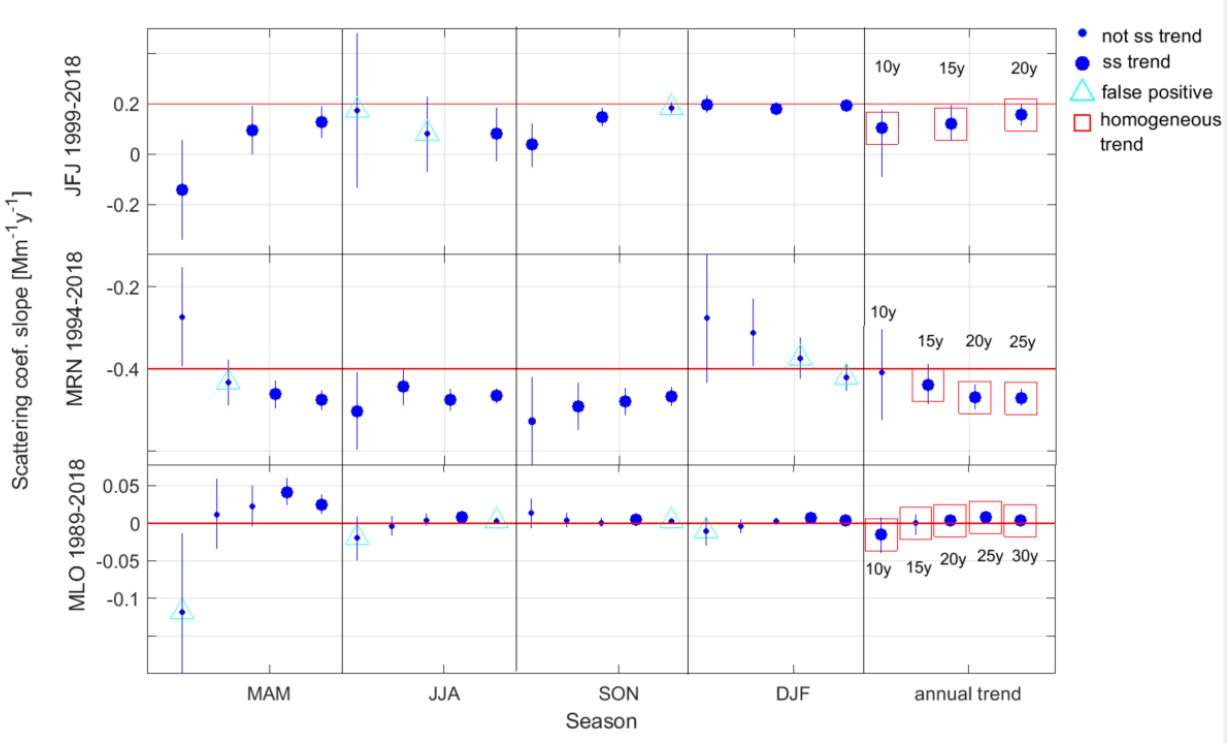

**Figure 2**: Seasonal MK results for $\sigma_{sp}$ trend for three stations with long time series: JFJ, MRN and MLO. The trends are plotted for the last 10 years period (2009-2018) as well as for all possible longer periods (15 y=2004-2018 to 30 y=1989-2018). The seasons correspond to meteorological seasons (MAM= March-April-May, JJA= June-July-August, SON=September-October-November and DJF= December-January-February). The dots correspond to the slope, large dots being ss at 95% confidence level whereas small dot are not ss trends. The cyan

triangles correspond to false positive trends (with type I error). Red squares correspond to annual trends where the seasonal results are homogeneous.

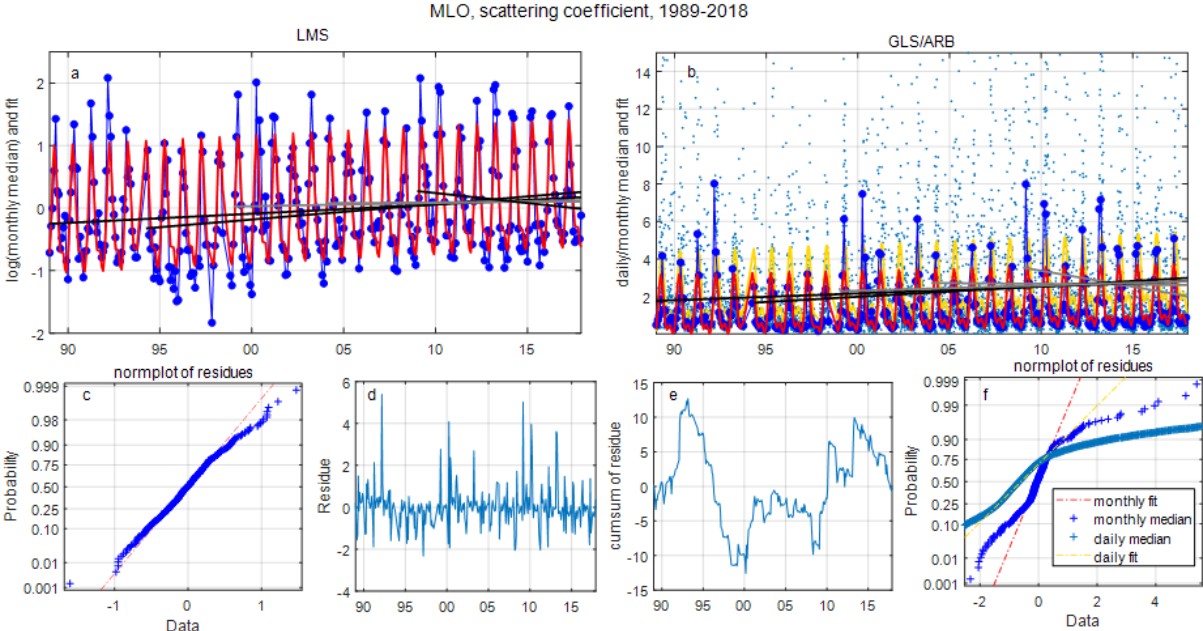

**Figure 3**: LMS and GLS/ARB results of MLO σsp: a) Logarithm of the monthly medians (blue circles), LMS fit (red) and the 10 y to 30 y slopes (ss slopes are plotted in black and not ss slopes in grey), b) daily medians (light blue dots) and their GLS/ARB fit (orange line), monthly medians (blue circles) and their GLS/ARB fit (red) and the 10 y to 30 y slopes, c) normplot of LMS residues, d) monthly medians of the GLS/ARB residues, e) cumulative summation of monthly median GLS/ARB residues and f) normplot of GLS/ARB residues for daily medians (light blue crosses and orange line) and monthly medians (blue crosses and red line).

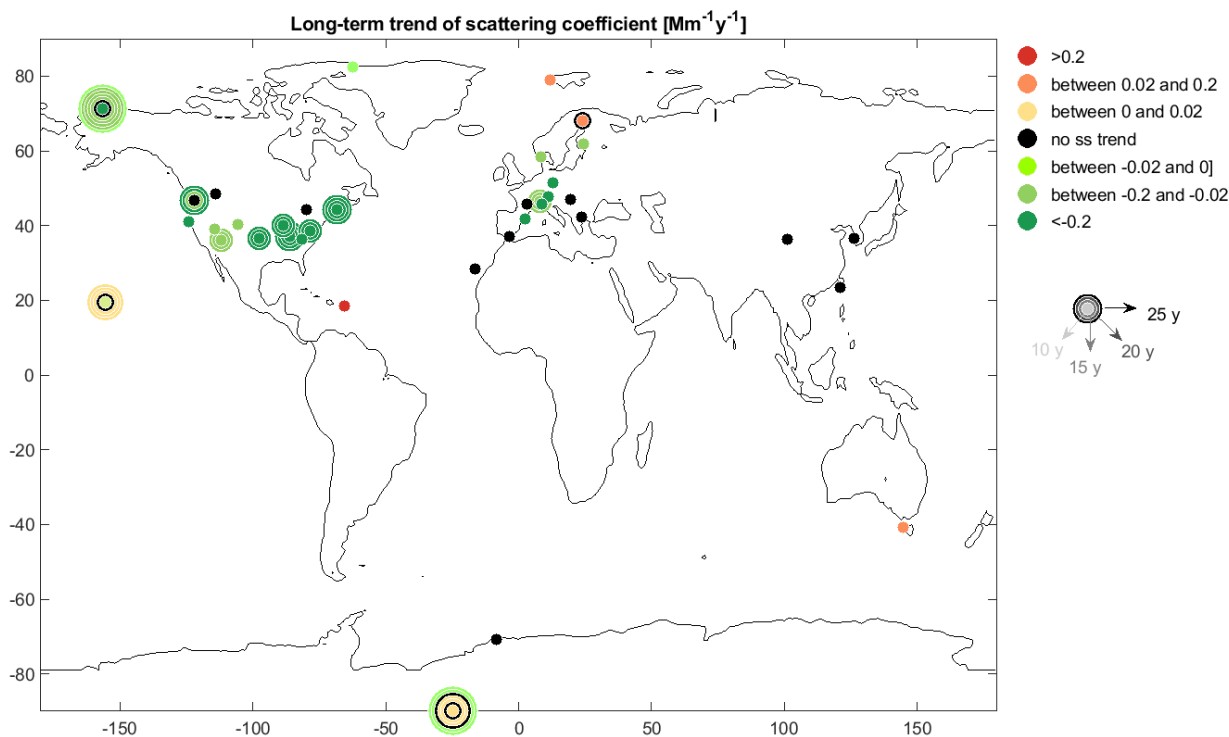

**Figure 4**: MK trends results for the scattering coefficient. Black symbols correspond to stations with no significant trends. Green and orange symbols correspond to ss negative and positive trends, respectively. The magnitude of the trends (slope) is given by the colors as stipulated in the legend. The size of the circles is proportional to the length of the data sets with the central dots representing the most recent 10 y trend ending in 2016, 2017 or 2018. If possible, trends for longer time periods were calculated and the larger circles denote the trends for 15 y to 40 y in 5 y increments.

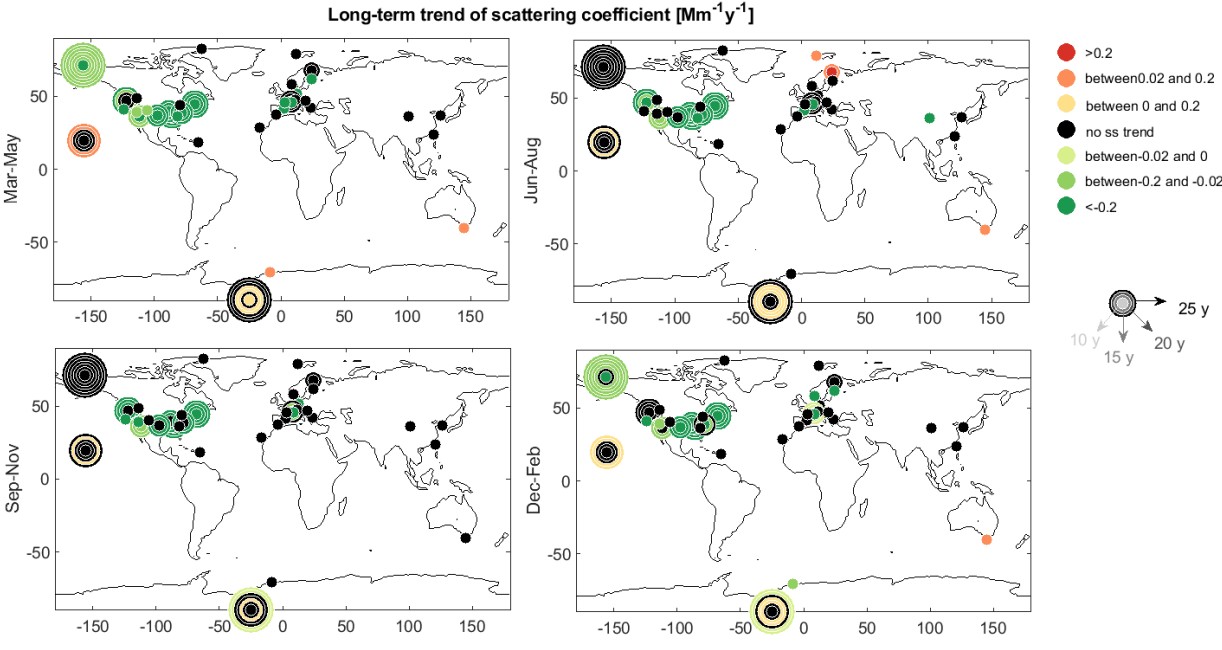

**Figure 5**: Seasonal results of the MK trend of the scattering coefficient. Other details same as Fig. 4.

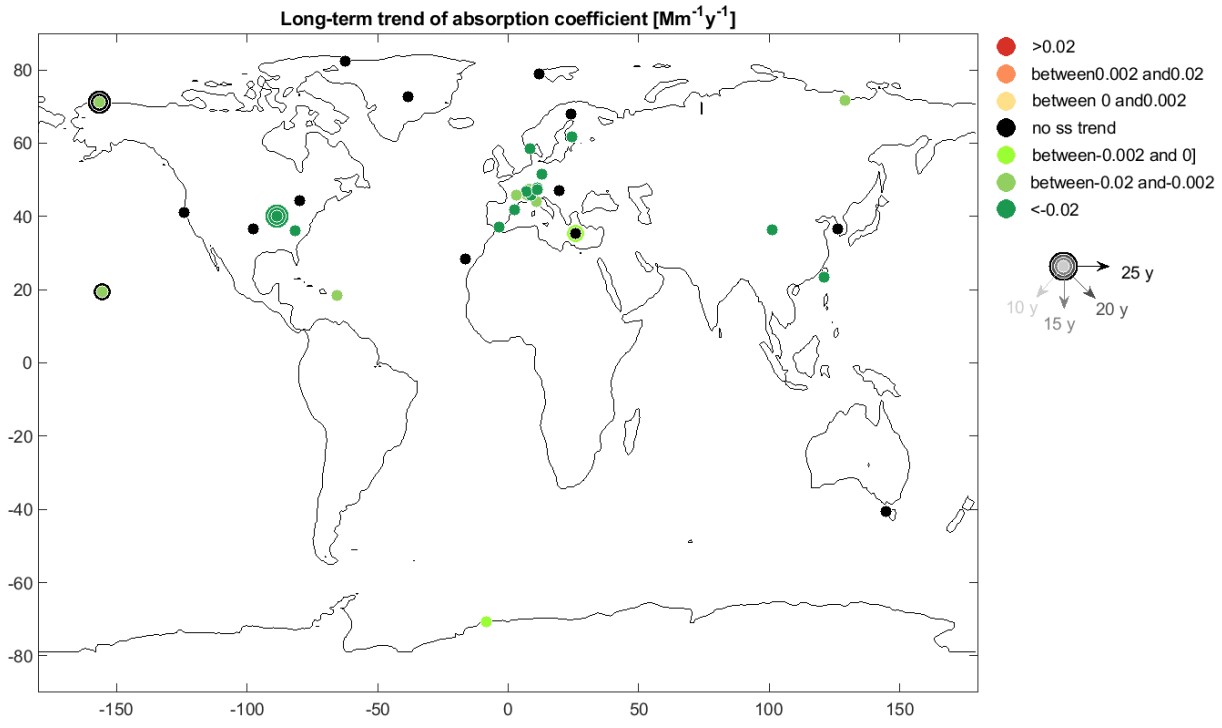

**Figure 6**: MK trends results for the absorption coefficient. Other details same as Fig. 4.

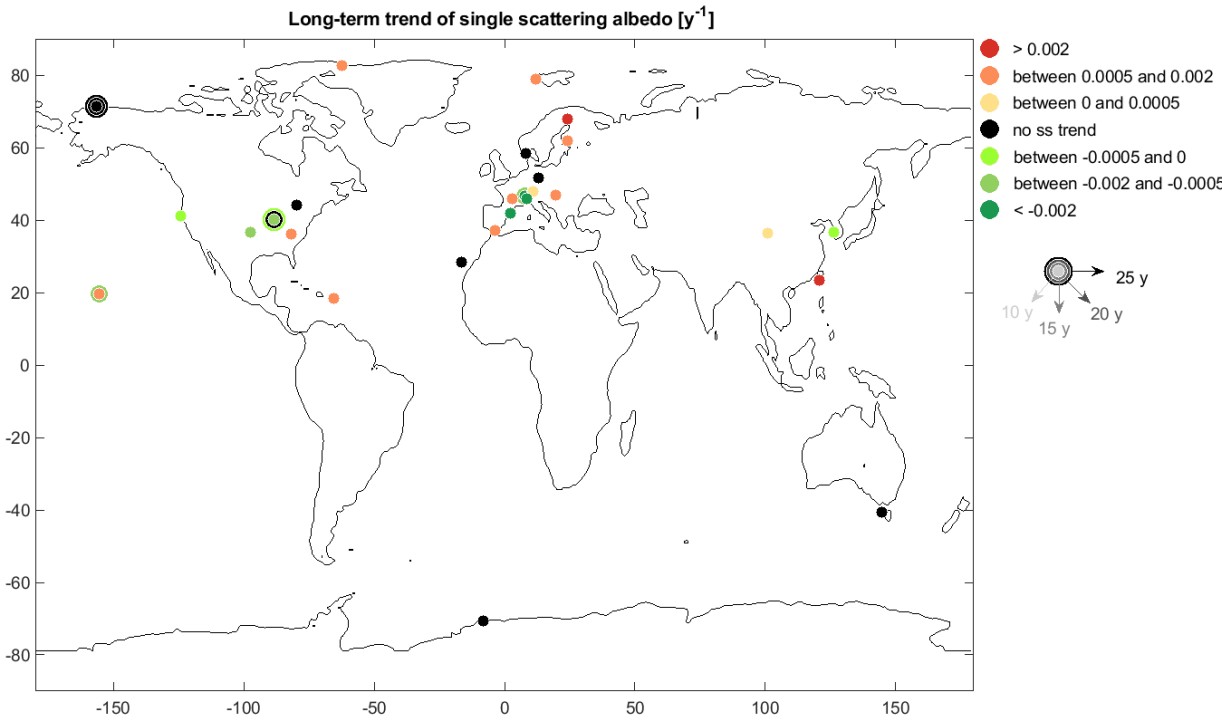

**Figure 7**: MK trend results for the single scattering albedo. Other details same as Fig. 4

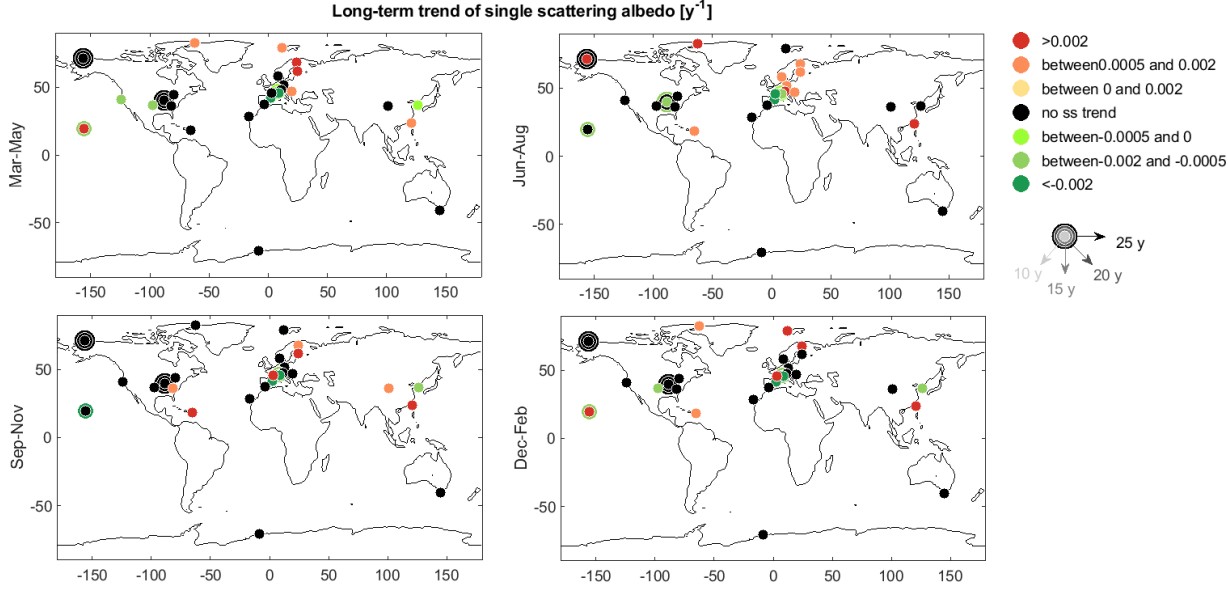

**Figure 8**: Seasonal results of the MK trend of the single scattering albedo. Other details same as Fig. 4.

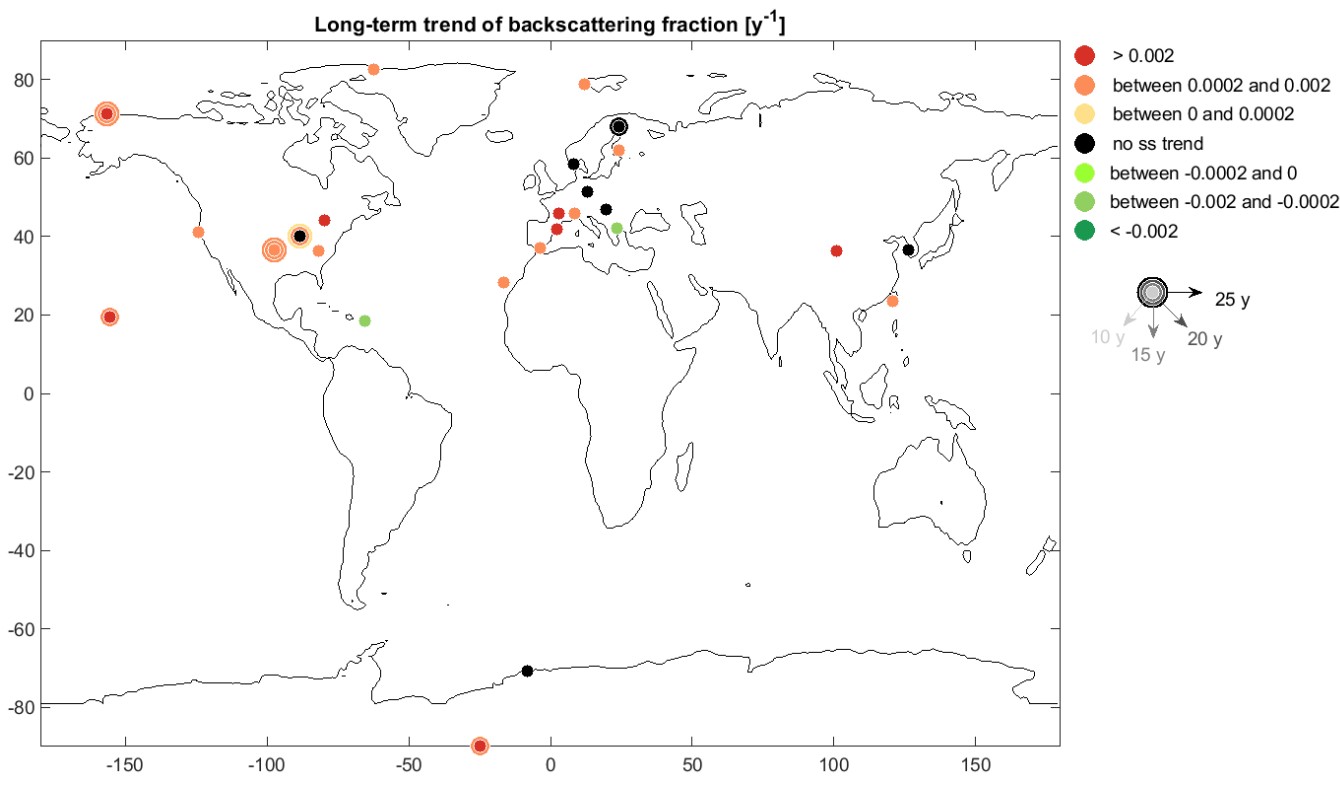

**Figure 9**: MK trends results for the backscattering fraction. Other details same as Fig. 4

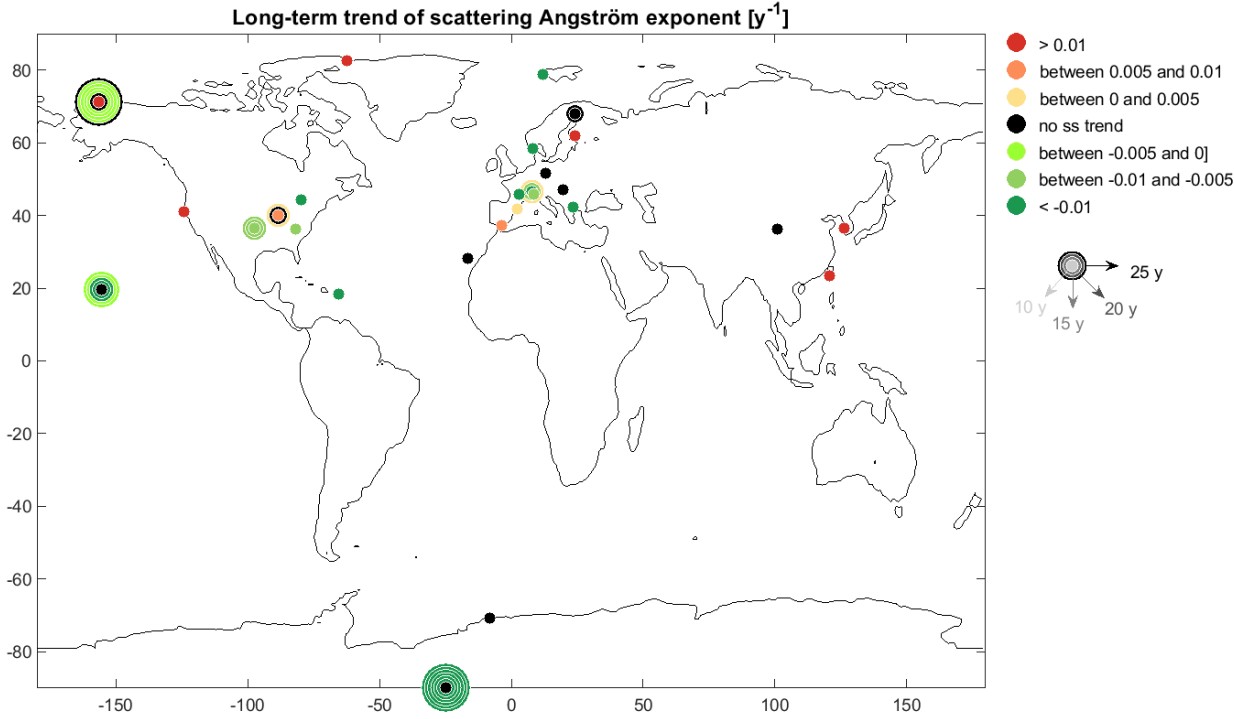

**Figure 10**: MK trend results for the scattering Ångström exponent. Other details same as Fig. 4.

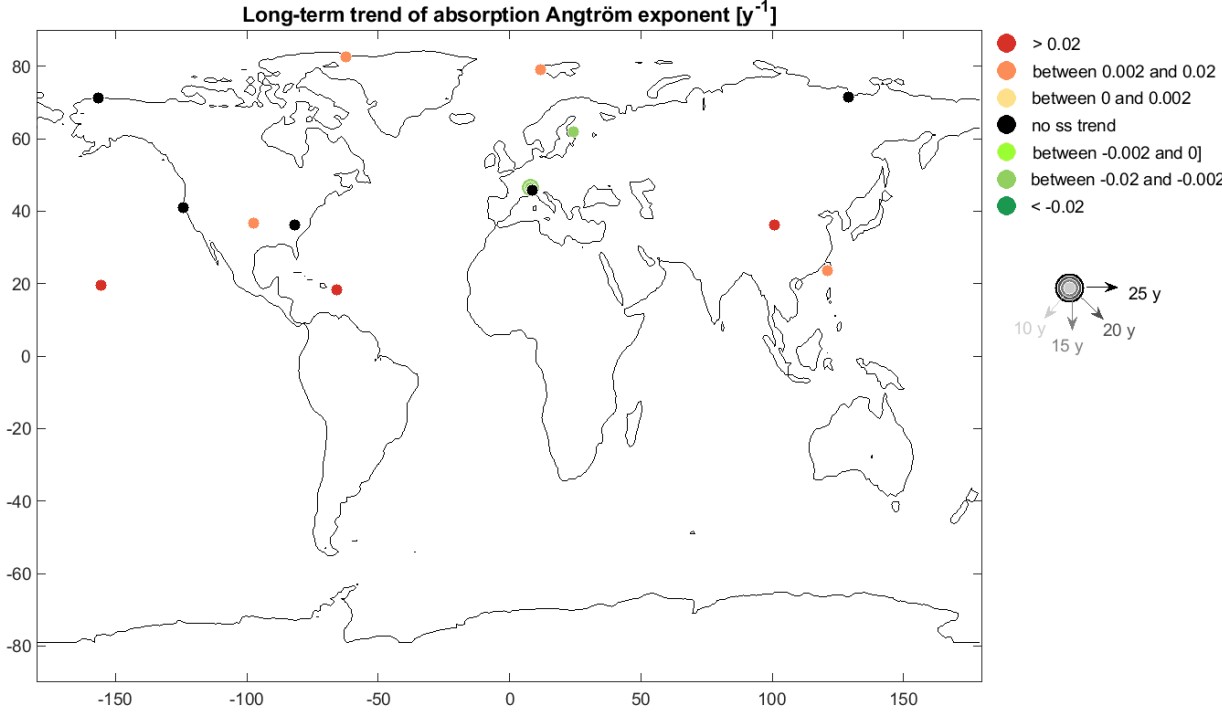

**Figure 11**: MK trends results for the absorption Ångström exponent. Other details same as Fig. 4.

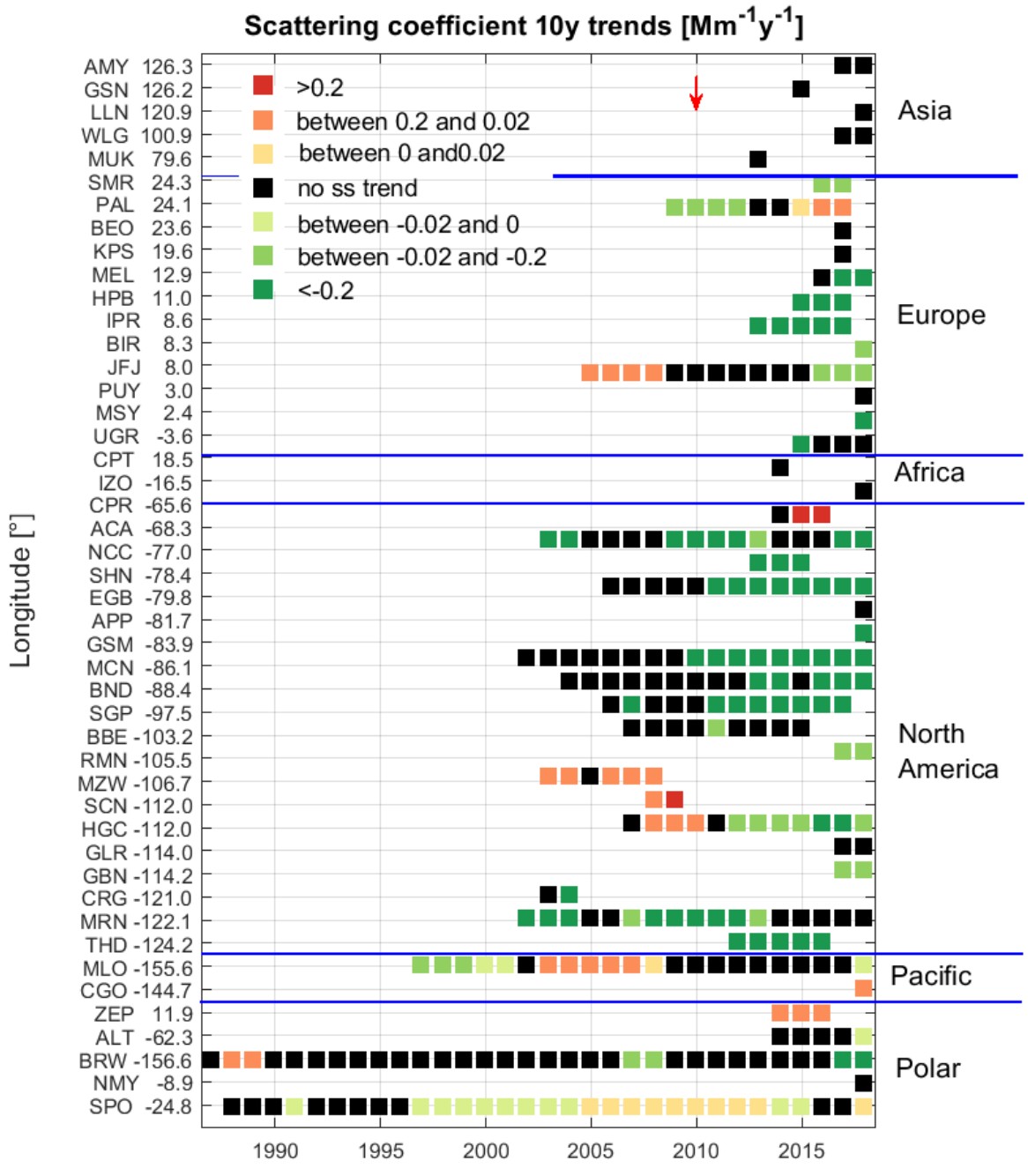

**Figure 12:** Time series of sequential 10 y $\sigma_{sp}$ trends as a function of station longitude. Stations in the South Pacific and in polar regions were grouped for clarity. The red arrow indicates the end of the time periods covered in CC2013.

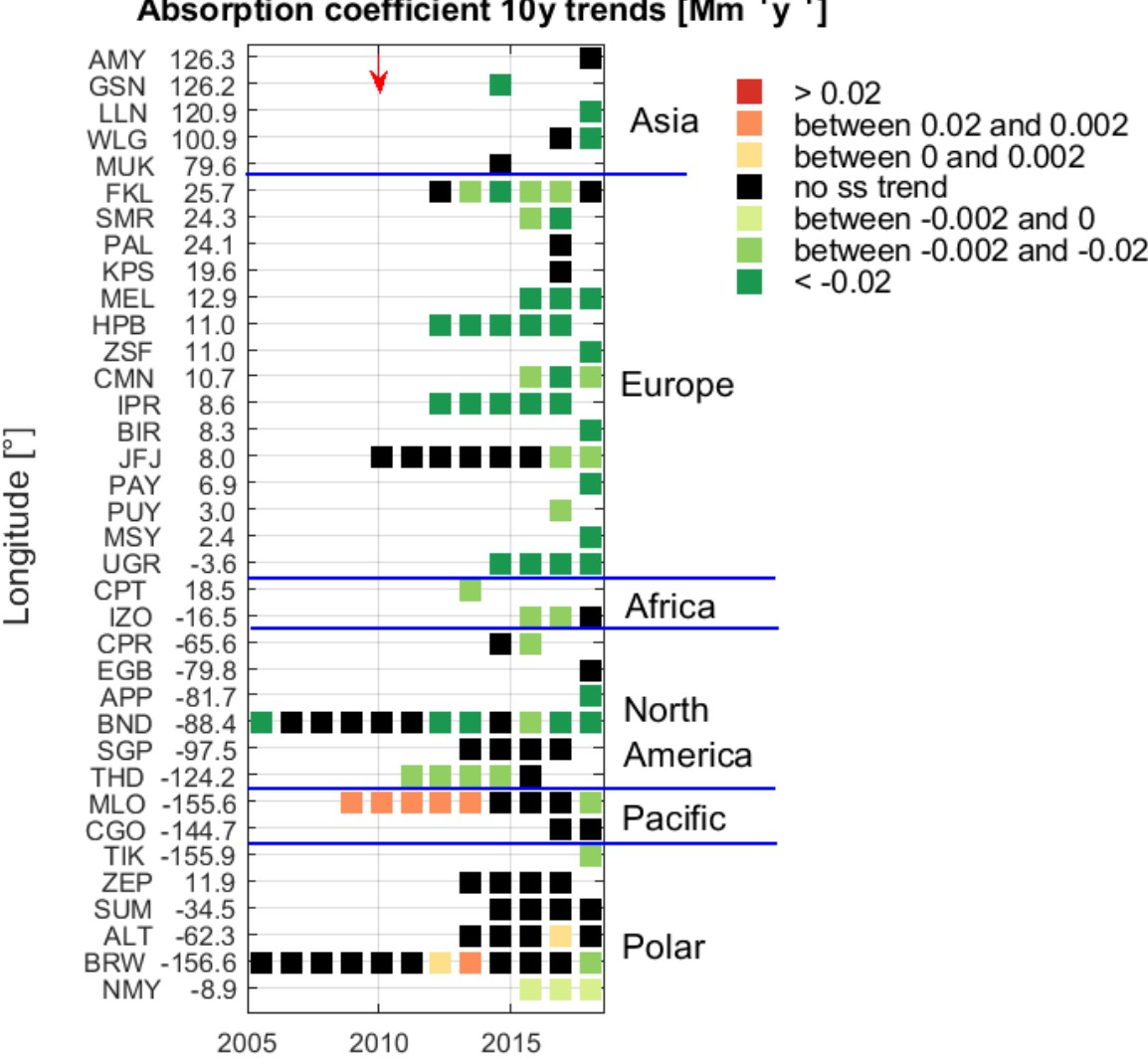

**Figure 13:** Time series of sequential 10 y $\sigma_{ap}$ trends as a function of station longitude. The red arrow indicates the end of the time period covered in CC2013.

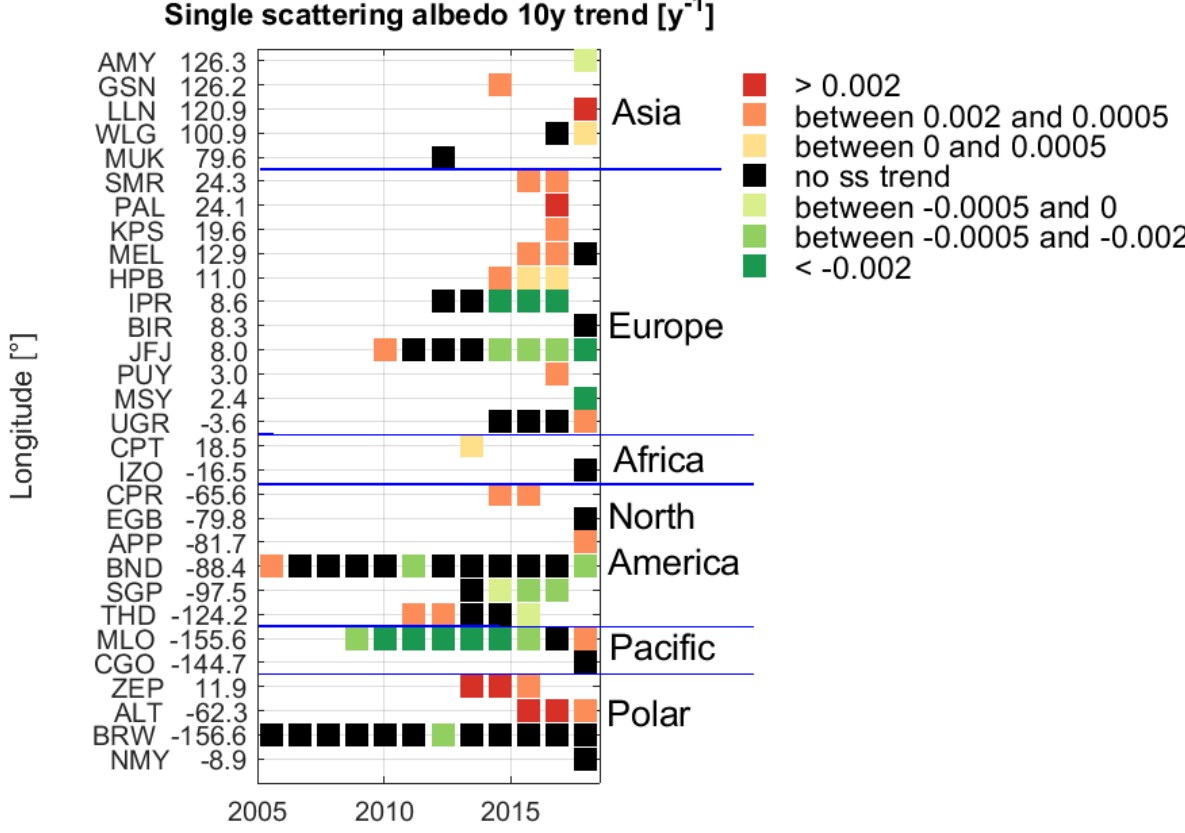

**Fig 14:** Time series of sequential 10 y $\omega_0$ trends as a function of station longitude.

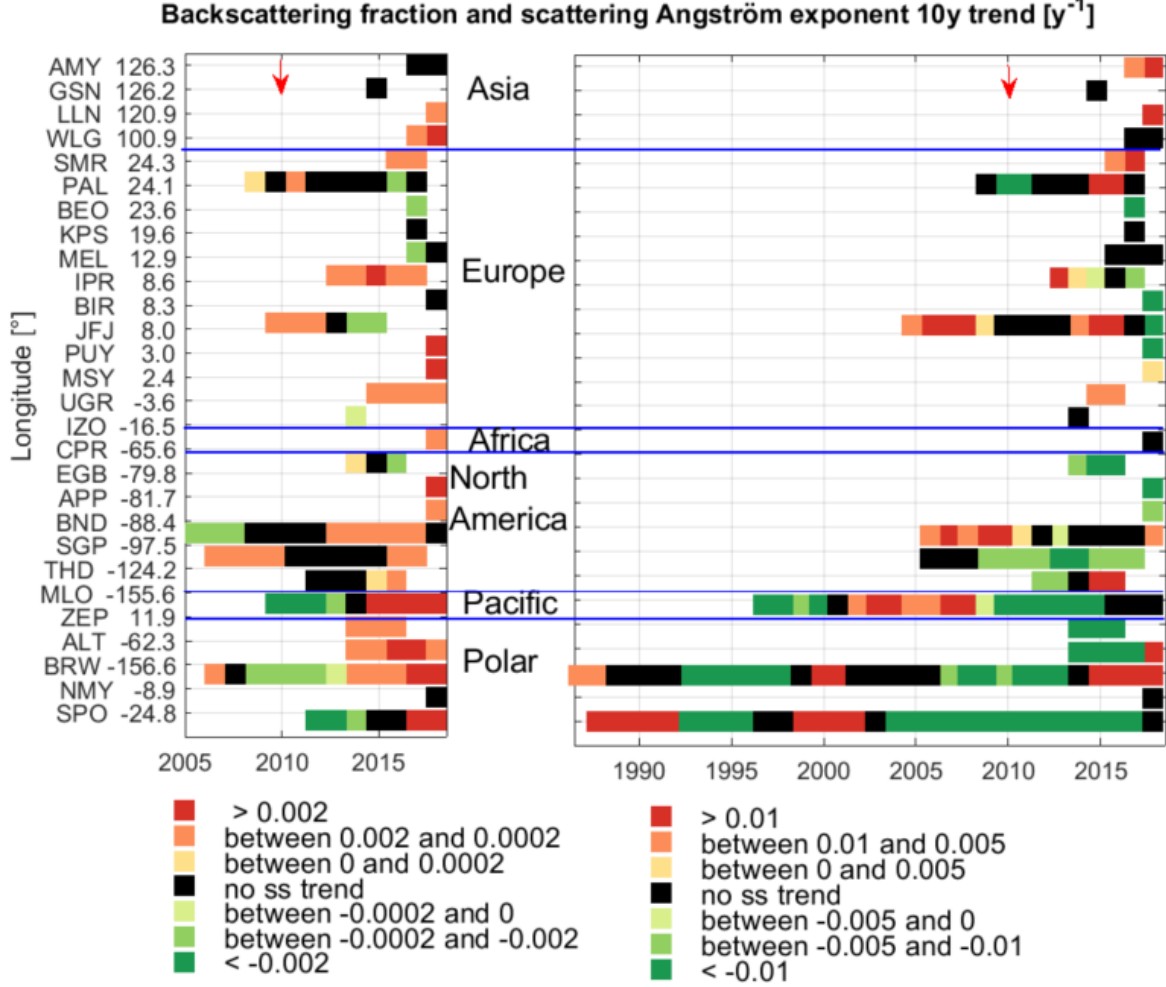

**Figure 15**: Time series of sequential 10 y b and å_sp trends as a function of station longitude. The red arrows indicate the end of the time period covered in CC2013.

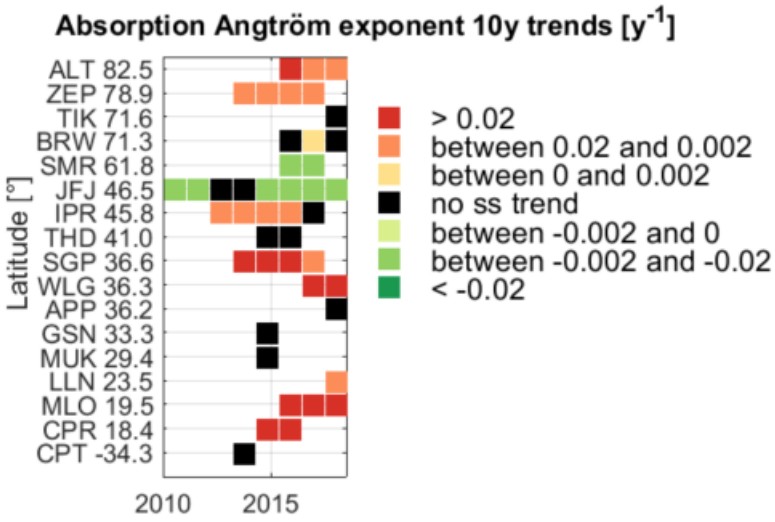

**Figure 16:** Time series of sequential 10 y å_ap trends as a function of station latitude.

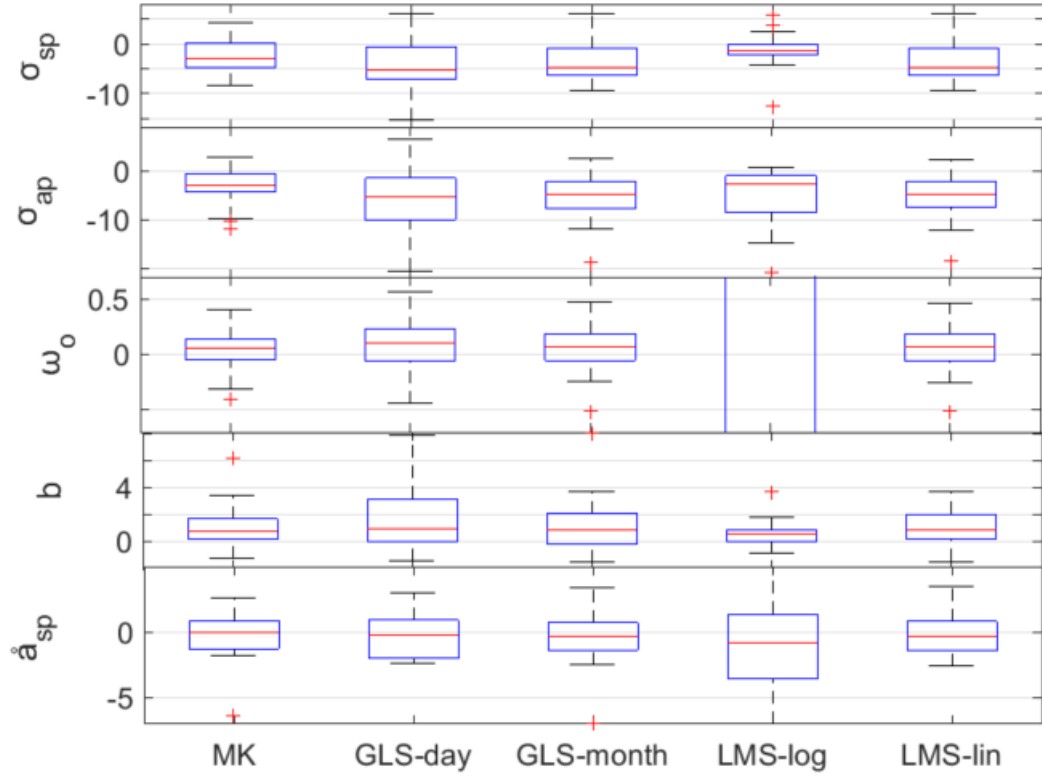

**Figure 17**: Median, interquartile ranges and whiskers of the slopes in %/y computed by the five methods for the scattering coefficient, the absorption coefficient, the single scattering albedo, the backscattering fraction and the scattering Angström exponent. The outliers are not always visible in the figure for the purpose of clarity.

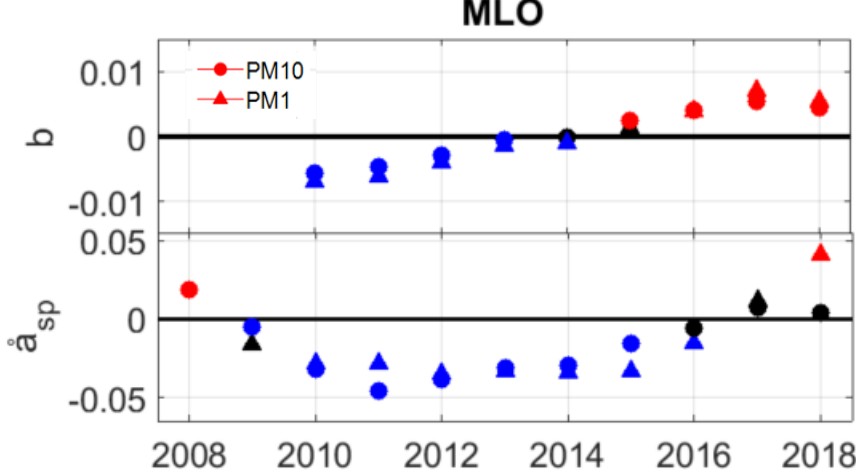

Fig 18: 10 y slopes of b and å$_{sp}$ at MLO. Ss negative and positive trends are plotted in blue and red, respectively. Not ss trends are plotted in black. The dots correspond to PM10 and the upwards triangles to PM1.