# Peer review of "Multidecadal trend analysis of in-situ aerosol radiative properties around the world"

_Atmospheric Chemistry and Physics, 2019_

## Referee Comment (RC1) · Anonymous Referee #1 · 12 Feb 2020

Multidecadal trend analysis of aerosol radiative properties at a global scale
Coen et al.

This manuscript collates data from a number of ground stations making aerosol measurements around the world. The authors are to be commended on the effort of making these data as consistent as possible. Doing this work is extremely important. Most of the major conclusions about trends are already known but it is a solid piece of work.

I have two major concerns with the manuscript and some other general concerns. The first major concern is overstatement about the global scale of these data. The second is a lack of the context provided by satellite data.

First, the manuscript claims in the title to be at a global scale and makes statements about global trends (for example page 25 line 5 "leading to global positive median trend of 0.02%/y"). There is just no way that the stations in the manuscript represent the global scale. In figure 12, there are 46 stations, 32 of which are in North America or Europe. Other figures are very similar. That means about 2/3 of the data are from less than 7% of the area of the Earth. In Figures 4 through 11 there are no stations in South America, none in the vast majority of Africa, and none on the main continent of Australia. You can't claim a global scale when entire continents are missing. Furthermore, the station locations are probably biased to regions with decreasing trends. Regions with recent decreasing trends in aerosols, such as North America and Europe, are heavily represented. On the other hand, regions with increasing trends in aerosol in the last decade or two, such as India and the Mideast, are not represented in the figures.

One specific statement is the conclusion (page 24 line 40) "Results from this study provide evidence that the aerosol load has significantly decreased over the last two decades in the regions represented by the 52 stations" is very misleading and should be changed. It should read something more like "The stations considered confirm decreasing trends in North America and Europe. Trends elsewhere are scattered, with too few stations to understand global trends." The rest of the manuscript should be similarly less definite about a global scale.

Second, the manuscript perpetuates an unfortunate situation in the literature that the in-situ and satellite researchers rarely make use of the other. I often tell satellite researchers they need to consider the in-situ data. Here the in-situ researchers need to consider the satellite data. Why should this manuscript be the one to do that? It claims a "global scale", and that definitely means including some satellite data.

In the present manuscript, satellite data are dismissed saying the ground stations have longer records (line 16, page 3). This is mostly untrue. MISR and MODIS both have 20 years of data, making their record longer than all but a handful of ground stations (and almost all of those handful are in the United States). SeaWifs has an even longer data record (Hsu et al., 2012). for the entirety of satellite data, the manuscript has only one oddly chosen reference about measurements in South Korea.

MISR and the newer MODIS retrievals provide aerosol optical depth over land as well as ocean. They measure more than optical depth. MODIS measures the Angstrom coefficient for

scattering. MISR has a measure of the single scattering albedo, with some difficulties in the measurement but good enough for trends in some locations.

I am not asking for a major review of how satellite data relates to long-term, ground-based measurements. I do think it is reasonable to ask you to show a figure with a map of satellite-derived trends in optical depth, Angstrom coefficient, and possibly aerosol absorption (optional, since the satellite absorption data are a bit trickier). The period could be something like 2009-2018 or 2004-2018 to match most of the ground sites. Then use that figure to put your ground stations in context. It isn't that hard to produce such figure. Out of 40+ authors there should be somebody who has experience using satellite data. If there isn't, it says something about our field. I'm copying one of your figures next to some satellite context below.

For discussing the context from satellite data, one important reference is Zhao et al. (Environ. Res. Lett, 2017) because it shows trends in not only optical depth but also detailed optical properties such as single scattering albedo for the Eastern US, Europe, and China. At a quick glance, those trends seem consistent with the ground stations; you can do a better analysis.

*Environ. Res. Lett.* **12** (2017) 054021

[Figure]

Figure 2. Annual mean AOD during 2001–2015 observed by MISR, MODIS/Terra, and MODIS/Aqua in three target regions. The dashed lines and the numbers in the legends represent trendlines and slopes (yr$^{-1}$) determined by linear regression, respectively. The slopes are marked with a star (∗) if they are statistically significant at the 0.05 level based on Student's t-test. The trends and their statistical significance determined by the Mann–Kendall test associated with Sen's slope are shown in table S1.

[Figure]

Figure 5. Seasonal mean fraction of the large-size AOD and single-scattering albedo during 2001–2015 observed by MISR in three target regions. The fraction of large-size AOD: (a) EUS, (b) WEU, and (c) ECC. Single-scattering albedo: (d) EUS (e) WEU, and (f) ECC. The dashed lines and the numbers in the legends represent trendlines and slopes (yr$^{-1}$) determined by linear regression, respectively. The slopes are marked with a star (∗) if they are statistically significant at the 0.05 level based on Student's t-test. The trends and their statistical significance determined by the Mann–Kendall test associated with Sen's slope are shown in table S3.

Alfaro-Contreres et al. (2017), Wei et al. (2019), and Murphy (2013) show the context that the region from the Mideast to India (with no ground stations) has had increasing trends in aerosol.

[Figure]

**Figure 4.** The deseasonalized, monthly and regionally averaged AOTs for eight selected regions utilizing MODIS C6 DT and MISR aerosol products. Straight lines are linear fits to the monthly data.

[Figure]

**b**

Zonal average

Annual average trend in 670 nm optical depth (yr$^{-1}$)

$-0.005$   $0.000$   $0.005$   $0.010$

(e) MOD08

(f) MYD08

(g) MISR

(h) SeaWiFS

$-0.01$   $-0.005$   $0.0$   $0.005$   $0.01$

**Figure 14.** Linear trend based on deseasonalized monthly AOD$_S$ anomalies from 2003 to 2010. Units are AOD yr$^{-1}$. Black dots indicate a significant trend at the 95 % confidence level ($p < 0.05$).

Long-term trend of scattering coefficient [Mm$^{-1}$y$^{-1}$]

> 0.2
between 0.02 and 0.2
between 0 and 0.02
no ss trend
between -0.02 and 0]
between -0.2 and -0.02
< -0.2

25 y
10 y   20 y
15 y

**igure 4**: MK trends results for the scattering coefficient. Black symbols correspond to stations

You can also look at Mehta et al., Remote Sensing of the Environment, 2016 Kahn and Gaitley JGR 2015, Hsu et al., ACP, 2020, "Global and regional trends…", Wei et al., ACP, 2019, "Intercomparison in spatial distribution and temporal trends". This is not a comprehensive list.

Finally, I have two lesser general concerns. The first is to think about how extreme events can affect trends. There are aerosol events that are so large they can change even a decade-long trend with a single event. I would bet that the recent Australian fires were big enough to change the trends for that region, indeed large swaths of the Southern Hemisphere, for an entire decade. The 1997-98 Indonesian fires also were big enough. There may be other such events in your data series. This doesn't disprove the validity of your statistical analysis, just look at the time series and comment if appropriate.

The other general comment is that most of section 4.4 is speculation without supporting evidence.

---

## Referee Comment (RC2) · Wenche Aas (Referee) · 14 Feb 2020

This manuscript gives a thorough and comprehensive overview of the long term measurements of aerosols optical- and physical properties. The analysis is robust with a thorough screening of the data for inconsistencies, evaluation of the QA/QC done and comparing different statistical tool. It gives confident in the assessment. I have some comments, which you may take into account:

P7, line 35. Why are the trends based on daily medians and not mean? It would have been nice to include a sentence of the choice of aggregation.

Page 11. If as stated in line 39 that the TFPW rejection rate is too high, why is the criteria as stated in line 35 significant trend considered only when both PW procedures

gave ss?

Page 11 line 49. What does it actually mean that the seasons are homogeneous?

Figure 2. Spelling mistake in caption. RMN should be MRN

Page 12. Line 40. GLS/ARB trends for MLO: "the longer periods exhibit ss negative trends". Should read positive trends if to be similar as MK, and it also looks like positive trends in Figure 3.

Figure 3. The fitted curves are for the longest periods I assume, (small) different fit for shorter periods?

Figure 4. ALT is missing on the map. The site is present in Figure 5.

P 12, L 43-45. If the seasonality fit is better using daily median compared to the GLS/ARB data, why not also use daily data for the LMS analysis. That would make all three methods more comparable to the data used in the MK test?

P13, 42-44. The paragraph of MLO seems a bit station specific, while the rest of the bullet point represent regions. Maybe add a sentence that this site represents the Pacific as in table 4 and then maybe include CGO?

P14, line 18. GLR does not turn up as ss positive in Figure 12 or 4, though in Table 2

P14, lin20. "PM10 trends are five times larger than PM1", maybe rewrite to "scattering trends of PM10 aerosols are five times larger than PM1". Do not find separate results for the different size cut off. Could they be included in Table 2 and Figure 12? And maybe indicate which size are chosen for Figure 4 and 5?

P14 line 27-31. Do not understand the possible "enhanced" NPF. The references only describes that you may have more NPF at high altitudes but not the possible trends. If the NPF should mask the trend we would expect more NPF in present day than earlier, or?

P14 line 43. Not only ALT with positive trend, also NMY.

P15 line 8. WLG show negative trend in Figure 6, Table 2 and Figure 13.

P15 line 11-12. NMY (and ALT) show positive trend

Figure 6. Seems like several of the sites miss information (outside circles) of longer trends than the last 10 years. I.e. JFJ, FKL, IPR, UGR etc.

There are two Figure S7. The first with backscatter trend should maybe be in the paper?

P15, line 33. Not sure about "Mostly decreasing". The mean trend is decreasing maybe (Table 4), but only 1 site with ss negative trend (SGP)

P16 line 9-10. Why not mentioned that the polar sites ALT and ZEP show positive trends?

Chapter 3.2. When I read chapter 3.1, I used the Figures developed for 3.2 since they are connected and give a more complete picture. Think the presentation of trend results would have benefited to combine these chapters.

P17 line 36. I don't find the trends in Figure 12 that much scattered, and I am not sure if one can track the differences back to abatement strategies. The UGR site is urban and influence by Saharan dust and is not representative for detecting general trends in Europe. The increase in SMR the latter periods might be do to increase emissions of BVOCs from the boreal forest? The trends in observed and modelled chemical composition in Europe are non linear due to changes in atmospheric chemistry. Maybe refer to some model/observation studies Europe trends for comparison. E.g:

Banzhaf, S., Schaap, M., Kranenburg, R., Manders, A. M. M., Segers, A. J., Visschedijk, A. J. H., Denier van der Gon, H. A. C., Kuenen, J. J. P., van Meijgaard, E., van Ulft, L. H., Cofala, J., and Builtjes, P. J. H.: Dynamic model evaluation for secondary inorganic aerosol and its precursors over Europe between 1990 and 2009, Geosci.

[Figure]

Model Dev., 8, 1047–1070, https://doi.org/10.5194/gmd-8-1047-2015, 2015.

Ciarelli, G., Theobald, M. R., Vivanco, M. G., Beekmann, M., Aas, W., Andersson, C., Bergström, R., Manders-Groot, A., Couvidat, F., Mircea, M., Tsyro, S., Fagerli, H., Mar, K., Raffort, V., Roustan, Y., Pay, M.-T., Schaap, M., Kranenburg, R., Adani, M., Briganti, G., Cappelletti, A., D'Isidoro, M., Cuvelier, C., Cholakian, A., Bessagnet, B., Wind, P., and Colette, A.: Trends of inorganic and organic aerosols and precursor gases in Europe: insights from the EURODELTA multi-model experiment over the 1990–2010 period, Geosci. Model Dev., 12, 4923–4954, https://doi.org/10.5194/gmd-12-4923-2019, 2019.

P26 line 26. Does it have to be anthropogenic sources? Changes in natural sources, typically BVOC may contribute? May a change in atmospheric composition contribute to smaller aerosols, i.e. less sulfate aerosols and more ammonium nitrate which potentially might be smaller?

P22 line 46- Bodhain and Dutton, 1993 is a quite old reference, maybe add e.g. Hand et al 2012 for a longer trend analysis for especially sulfate. Further, one should probably mention that the in Asia the decrease started rapidly after 2013 when the China's Clean Air Action was implemented. Maybe add Paulot et al (2018), which gives a nice global overview of trends using satellites and models

Hand, J. L., Schichtel, B. A., Malm, W. C., and Pitchford, M. L.: Particulate sulfate ion concentration and SO2 emission trends in the United States from the early 1990s through 2010, Atmos. Chem. Phys., 12, 10353–10365, https://doi.org/10.5194/acp-12-10353-2012, 2012.

Paulot, F., Paynter, D., Ginoux, P., Naik, V., and Horowitz, L. W.: Changes in the aerosol direct radiative forcing from 2001 to 2015: observational constraints and regional mechanisms, Atmos. Chem. Phys., 18, 13265–13281, https://doi.org/10.5194/acp-18-13265-2018, 2018.

P23. Line 49 Pandolfi et al (2016) only show PM trends in NE Spain. There are several other national PM trend studies in Europe (e.g. Germany, France, Switzerland). For a complete overview of Europa it is possible to refer to EMEP TFMM assessment report showing PM trends from both EMEP and AIRBASE for 2002-2012: Colette et al 2016: https://projects.nilu.no//ccc/reports/cccr1-2016.pdf (chapter 3.6.1)

---

## Referee Comment (RC3) · Anonymous Referee #3 · 10 Mar 2020

The manuscript covers a comprehensive overview of long-term in-situ measurements of optical and physical properties of aerosols comprising data from numerous stations significant on global scale for the documentation of inert evolutions of radiative properties. It includes preparation and quality assurance of data, as well as several statistical methods used for trend analysis.

Few comments from my side:

P3, L10 and L11: there is a space between - and 0.45 (leading to a newline between)

P8, L15: Assuming an Absorption Angström exponent of one for SSA calculation could cause further dependence on changes of size distribution or chemical composition. What is the impact of this assumption?

[Figure]

P8, L24 – L28: is part of data preparation and thus could be moved to section 2.4

Section 2.4 is missing a paragraph on assessment for nephelometer artefacts

P12, L15: do the monthly medians fit the log-normal distribution and what was the procedure to deal with negatives or zero values? What was the reason for median as aggregation method?

P16, L13: could add "Backscatter fraction (b)" for readability

P20, L15 and L17: "derived parameters" would be more specific (instead of "computed parameters")

P23, L18 and L19: the intention of "Ideally, abatement policy. . ." is not clear and vague

P27, L15: "due because"

---

## Short Comment (SC1) · 11 Mar 2020

Studying long time series is important part of atmospheric research and this work has significant potential to increase our knowledge on evolution of aerosol parameters affecting climate change. However, I see some methodological details that should be addressed before drawing final conclusions from the data.

My comments:

Using a pre-whitening method always loses information from the data and because there is no information on the applied method (Collaud Coen et al., in preparation), it is impossible to see how much information is lost. Thus, results of this work cannot be evaluated before the method is available for inspection.

[Figure]

There exists time series analysis methods which do not require pre-whitening, why the authors are not considering them? For example, dynamic linear models (DLM) have been shown to be good tools for atmospheric data e.g. in Laine et al. (2014), Dunne et al. (2015) and Mikkonen et al. (2015). with DLM, it is possible to model time-varying trends in measured time series and at the same time take account structural dependencies, e.g. seasonality and autocorrelation, in the data.

In addition, it shows from Figure 3 that the trends in the data cannot be described with one linear slope. With DLM the shape of the trend is not limited to straight line but the trend can change its value continuously and it can be analyzed directly if the time series contains changepoints and where they most likely are.

DLM method is nicely described in Laine (2020) and supported by freely distributed Matlab toolbox in https://mjlaine.github.io/dlm/index.html

References

Dunne, E. M., Mikkonen, S., Kokkola, H., and Korhonen, H.: A global process-based study of marine CCN trends and variability, Atmos. Chem. Phys., 14, 13631–13642, https://doi.org/10.5194/acp-14-13631-2014, 2014.

Laine, M., Latva-Pukkila, N., and Kyrölä, E.: Analysing time-varying trends in strato-spheric ozone time series using the state space approach, Atmos. Chem. Phys., 14, 9707–9725, https://doi.org/10.5194/acp-14-9707-2014, 2014.

Laine, M. : Introduction to Dynamic Linear Models for Time Series Analysis. in book: Geodetic Time Series Analysis in Earth Sciences, pp 139-156, doi: 10.1007/978-3-030-21718-1_4, 2020

Mikkonen S., Laine M., Mäkelä H. M., Gregow H., Tuomenvirta H., Lahtinen M., Laak-sonen A.: Trends in the average temperature in Finland, 1847-2013 Stochastic Environmental Research and Risk Assessment 29 1521-1529. doi:10.1007/s00477-014-0992-2, 2015

---

## Author Comment (AC1) · 22 May 2020

**Answers to Santtu Mikkonen:**

The authors thank Santtu Mikkonen for his comments and suggestions about the statistical treatment of the trend analysis.

During the review process, the routines for MK trend analysis were translated into R and an error was found in the selection of data for north hemispheric winter season. This error was corrected in the original matlab routines leading to minor changes in slope absolute values for most of the stations, but also sometimes to modification of the statistical significance. The more important changes are:

- ALT was the only station with ss trend in absorption coefficient and this was the only case where there is a strong discrepancy among the analysing methods, MK being ss positive, LMS/log not ss and GLS/day ss negative. The correction leads to MK not ss trend in absorption coefficient at ALT and remove therefore the solely strong discrepancy between the methods.
- MLO has a ss negative trend in scattering coefficient for the last 10 y, leading to a better agreement between scattering and absorption trends. The evolution from positive to negative ss trends is now well established.
- Some other not ss present-day trends are now ss negative (RMN scattering coefficient, CPR absorption coefficient, THD single scattering albedo) or ss positive (PUY single scattering albedo, MSY scattering Ångström exponent, LLN absorption Ångström exponent).
- Some ss trends are now not ss: IZO absorption coefficient,
- One trend (JFJ scattering Ångström exponent for the 20y period) change from ss negative to ss positive trend.
- The statistical significance of some of the 10 y trends of the time evolution analysis (Sect. 3.2) is also modified, but these changes do not impact the results.

The revised manuscript and all tables and figures were corrected in order to take into account the new results.

**Answers to specific comments:**

1. Using a pre-whitening method always loses information from the data and because there is no information on the applied method (Collaud Coen et al., in preparation), it is impossible to see how much information is lost. Thus, results of this work cannot be evaluated before the method is available for inspection.

   *The referee is absolutely right. The applied methodology should have been available at the same time than this paper. Anyhow, this paper results from an international initiative in order to published this trend analysis of all in-situ aerosol optical parameters over the world so that it can be taken into account for the next release of the IPCC report. The paper on the applied methodology was then written thereafter but is submitted since three*

*weeks to Atmos. Meas. Techn. Discussion and I hope that it will be published there before the acceptation of this paper describing the results of the trend analysis.*

2. There exists time series analysis methods which do not require pre-whitening, why the authors are not considering them? For example, dynamic linear models (DLM) have been shown to be good tools for atmospheric data e.g. in Laine et al. (2014), Dunne et al. (2015) and Mikkonen et al. (2015). with DLM, it is possible to model timevarying trends in measured time series and at the same time take account structural dependencies, e.g. seasonality and autocorrelation, in the data. In addition, it shows from Figure 3 that the trends in the data cannot be described with one linear slope. With DLM the shape of the trend is not limited to straight line but the trend can change its value continuously and it can be analyzed directly if the time series contains changepoints and where they most likely are.

   *This analysis not only presents the non-parametric Mann-Kendall method for long-term trend analysis but also LMS and GLS/ARB results that do not require prewhitening. As described in sect. 2.5 and particularly in subsect 2.5.1, the distribution of the aerosol parameter is strongly skewed resulting in not normally distributed residues after LMS or GLS/ARB tests, so that non-parametric long-term trend analyses are required. Due to the high autocorrelation in the time series, a prewhitening method is also necessary in order to decrease the rate of rejection of the null hypothesis of no trend in the absence of a trend. The authors are well aware of the detrimental effect of prewhitening methods (see submitted manuscript Collaud Coen et al., 2020) but tried to apply the most adequate methodology.*

   *The authors are also aware of the DLM method: DLM is however a parametric method and should, thus, not be used if the residues of the fit are not normally distributed. The applied Mann-Kendall test was instead chosen and their results were compared to the LMS and GLS/ARB parametric methods. In order to have an insight into the change of the trend with time, all possible 10 y trends in the time series were computed and the results are described in Sect. 3.2. This procedure allows maintaining the rule of applying long-term trend analysis on periods of at least 10 y and can be considered, to some extent, as a differential non-parametric trend analysis method.*

---

## Author Comment (AC2) · 22 May 2020

The authors thank the referee for their detailed review of the manuscript and for all their comments and suggestions about comparison with satellite measurements. The inclusion of these airborn REM trend analyses allows a clear improvement of the paper with a more global view of aerosol modifications around the world.

During the review process, the routines for MK trend analysis were translated into R and an error was found in the selection of data for north hemispheric winter season. This error was corrected in the original matlab routines leading to minor changes in slope absolute values for most of the stations, but also sometimes to modification of the statistical significance. The more important changes are:

- ALT was the only station with ss trend in absorption coefficient and this was the only case where there is a strong discrepancy among the analysing methods, MK being ss positive, LMS/log not ss and GLS/day ss negative. The correction leads to MK not ss trend in absorption coefficient at ALT and remove therefore the solely strong discrepancy between the methods.
- MLO has a ss negative trend in scattering coefficient for the last 10 y, leading to a better agreement between scattering and absorption trends. The evolution from positive to negative ss trends is now well established.
- Some other not ss present-day trends are now ss negative (RMN scattering coefficient, CPR absorption coefficient, THD single scattering albedo) or ss positive (PUY single scattering albedo, MSY scattering Ångström exponent, LLN absorption Ångström exponent).
- Some ss trends are now not ss: IZO absorption coefficient,
- One trend (JFJ scattering Ångström exponent for the 20y period) change from ss negative to ss positive trend.
- The statistical significance of some of the 10 y trends of the time evolution analysis (Sect. 3.2) is also modified, but these changes do not impact the results.

The revised manuscript and all tables and figures were corrected in order to take into account the new results.

**Answers to specific comments:**

I have two major concerns with the manuscript and some other general concerns. The first major concern is overstatement about the global scale of these data. The second is a lack of the context provided by satellite data.

1. First, the manuscript claims in the title to be at a global scale and makes statements about global trends (for example page 25 line 5 "leading to global positive median trend of 0.02%/y"). There is just no way that the stations in the manuscript represent the global scale. In figure 12, there are 46 stations, 32 of which are in North America or Europe. Other figures are very similar. That means about 2/3 of the data are from less than 7% of the area of the Earth. In Figures 4 through 11 there are no stations in South America, none in the vast majority of Africa, and none on the main continent of Australia. You can't claim a global scale when entire continents are missing. Furthermore, the station locations are probably biased to regions with

decreasing trends. Regions with recent decreasing trends in aerosols, such as North America and Europe, are heavily represented. On the other hand, regions with increasing trends in aerosol in the last decade or two, such as India and the Mideast, are not represented in the figures.

*The referee is completely right. The word "global" was used for two purposes: 1) this study represents the best "globality" that can be reach with in-situ aerosol measurement and 2) the word "global" is used instead of the word "annual" in the result section. These two inadequate usages of the word "global" were modified in the manuscript. The title is now "Multidecadal trend analysis of in-situ aerosol radiative properties around the world".*

2.  One specific statement is the conclusion (page 25 line 40) "Results from this study provide evidence that the aerosol load has significantly decreased over the last two decades in the regions represented by the 52 stations" is very misleading and should be changed. It should read something more like "The stations considered confirm decreasing trends in North America and Europe. Trends elsewhere are scattered, with too few stations to understand global trends." The rest of the manuscript should be similarly less definite about a global scale.

*The authors agree that the results of this study cannot be considered as global due to the low representativity of stations in Asia, South America, Africa and Australia. The manuscript was consequently modified:*
*"Results from this study provide evidence that the aerosol load has significantly decreased over the last two decades in North America and Europe. The low number of stations in the other continents means global tendencies cannot be assessed and the results are more variable."*

3.  Second, the manuscript perpetuates an unfortunate situation in the literature that the in-situ and satellite researchers rarely make use of the other. I often tell satellite researchers they need to consider the in-situ data. Here the in-situ researchers need to consider the satellite data. Why should this manuscript be the one to do that? It claims a "global scale", and that definitely means including some satellite data.

In the present manuscript, satellite data are dismissed saying the ground stations have longer records (line 16, page 3). This is mostly untrue. MISR and MODIS both have 20 years of data, making their record longer than all but a handful of ground stations (and almost all of those handful are in the United States). SeaWifs has an even longer data record (Hsu et al., 2012). For the entirety of satellite data, the manuscript has only one oddly chosen reference about measurements in South Korea.

MISR and the newer MODIS retrievals provide aerosol optical depth over land as well as ocean. They measure more than optical depth. MODIS measures the Angstrom coefficient for scattering. MISR has a measure of the single scattering albedo, with some difficulties in the measurement but good enough for trends in some locations.

I am not asking for a major review of how satellite data relates to long-term, ground-based measurements. I do think it is reasonable to ask you to show a figure with a map of satellite derived trends in optical depth, Angstrom coefficient, and possibly aerosol absorption (optional, since the satellite absorption data are a bit trickier). The period could be something like 2009-2018 or 2004-2018 to match most of the ground sites. Then use that figure to put your ground stations in context. It isn't that hard to produce such figure. Out of 40+ authors there should be somebody who has experience using satellite data. If there isn't, it says something about our field. I'm copying one of your figures next to some satellite context below.

For discussing the context from satellite data, one important reference is Zhao et al. (Environ.Res. Lett, 2017) because it shows trends in not only optical depth but also detailed optical properties

such as single scattering albedo for the Eastern US, Europe, and China. At a quick glance, those trends seem consistent with the ground stations; you can do a better analysis.

Environ. Res. Lett. **12** (2017) 054021

[Figure]

**Figure 2.** Annual mean AOD during 2001–2015 observed by MISR, MODIS/Terra, and MODIS/Aqua in three target regions. The dashed lines and the numbers in the legends represent trendlines and slopes (yr⁻¹) determined by linear regression, respectively. The slopes are marked with a star (∗) if they are statistically significant at the 0.05 level based on Student's t-test. The trends and their statistical significance determined by the Mann–Kendall test associated with Sen's slope are shown in table S1.

[Figure]

**Figure 5.** Seasonal mean fraction of the large-size AOD and single-scattering albedo during 2001–2015 observed by MISR in three target regions. The fraction of large-size AOD: (a) EUS, (b) WEU, and (c) ECC. Single-scattering albedo: (d) EUS, (e) WEU, and (f) ECC. The dashed lines and the numbers in the legends represent trendlines and slopes (yr⁻¹) determined by linear regression, respectively. The slopes are marked with a star (∗) if they are statistically significant at the 0.05 level based on Student's t-test. The trends and their statistical significance determined by the Mann–Kendall test associated with Sen's slope are shown in table S3.

Alfaro-Contreres et al. (2017), Wei et al. (2019), and Murphy (2013) show the context that the region from the Mideast to India (with no ground stations) has had increasing trends in aerosol.

[Figure]

**Figure 4.** The deseasonalized, monthly and regionally averaged AOTs for eight selected regions utilizing MODIS C6 DT and MISR aerosol products. Straight lines are linear fits to the monthly data.

[Figure]

[Figure]

**Figure 14.** Linear trend based on deseasonalized monthly $AOD_S$ anomalies from 2003 to 2010. Units are $AOD\,yr^{-1}$. Black dots indicate a significant trend at the 95 % confidence level ($p < 0.05$).

[Figure]

You can also look at Mehta et al., Remote Sensing of the Environment, 2016 Kahn and Gaitley JGR 2015, Hsu et al., ACP, 2020, "Global and regional trends…", Wei et al., ACP, 2019, "Intercomparison in spatial distribution and temporal trends". This is not a comprehensive list.

*The authors thank the referee for pointing this lack of results from satellite measurements, for the proposed references and do agree about the unfortunate statement of the relative confinement of both research domains. The authors do further agree with the referee that the in-situ measurements of aerosol properties cannot be called "global" since large*

*domains of world are under-represented (e.g. Asia) or even not represented at all (e.g. Africa, South America, middle East). The world "global" was then removed from the manuscript.*

*After a literature review, guided in part by the reviewer's suggestions, we did not find an already published global long-term trend analysis from satellite measurements with the same (or similar) aerosol radiative properties used in our study. Despite the number of co-authors, none has a sufficiently experience to extract aerosol radiative properties from satellite measurements and compute global trends. Moreover, the authors consider that a necessary condition of good comparison between in-situ and satellite aerosol trends is the used of the same trend analysis methodology. If such an analysis would be very relevant, the amount of necessary work to achieve it is too large to be done in the restricted lapse of time imposed by the next IPCC report deadline. The authors are however open for a further collaboration in this domain.*

*To respond to the referee's requirement, sect. 4.4 was completely re-written in order to include some results from already published long-term trend analysis from satellite measurements:*

[revised manuscript text omitted]

4. Finally, I have two lesser general concerns. The first is to think about how extreme events can affect trends. There are aerosol events that are so large they can change even a decade-long trend with a single event. I would bet that the recent Australian fires were big enough to change the trends for that region, indeed large swaths of the Southern Hemisphere, for an entire decade. The 1997-98 Indonesian fires also were big enough. There may be other such events in your data series. This doesn't disprove the validity of your statistical analysis, just look at the time series and comment if appropriate.

*Extreme events have an increasing occurrence caused by the global warming. It would have been very interesting to study the long-term trends of the extremes of in-situ aerosol parameters. Some cited studies in the western USA (McClure and Jaffe, 2018) show that the average $PM_{2.5}$ were not affected by the increasing biomass burning events but that the extremes (98 percentiles) were largely affected. This study was not designed to study regional effects but to bring together all the in-situ aerosol long-term trends results in a solely and homogeneous study. The time delay and the already large size of the paper do not allow us to study the trends of the extreme events. Similarly, trend in coarse mode aerosol ($PM_{10}$-$PM_1$) would also be very interesting, but were skipped for the same reason.*

5. The other general comment is that most of section 4.4 is speculation without supporting evidence.

   *As already mentioned as an answer to comments 3, section 4.4 was completely re-written in order to include more satellite and model studies. Sect 4.4 is now divided in sub-sections dealing with 1) the comparison with other in-situ results, 2) the comparison with remote sensing instruments and finally 3) the causality. Model studies are mostly cited in the last sub-section. These modifications allow to better comparing the results of this study with global studies.*

---

## Author Comment (AC3) · 22 May 2020

**Answers to Wenche Aas:**

The authors thank Wenche Aas for her detailed review of the manuscript, for the meticulous pointing of incoherencies between tables and figures, as well as for all their comments and suggestions allowing a clear improvement of the paper.

During the review process, the routines for MK trend analysis were translated into R and an error was found in the selection of data for north hemispheric winter season. This error was corrected in the original matlab routines leading to minor changes in slope absolute values for most of the stations, but also sometimes to modification of the statistical significance. The more important changes are:

- ALT was the only station with ss trend in absorption coefficient and this was the only case where there is a strong discrepancy among the analysing methods, MK being ss positive, LMS/log not ss and GLS/day ss negative. The correction leads to MK not ss trend in absorption coefficient at ALT and remove therefore the solely strong discrepancy between the methods.
- MLO has a ss negative trend in scattering coefficient for the last 10 y, leading to a better agreement between scattering and absorption trends. The evolution from positive to negative ss trends is now well established.
- Some other not ss present-day trends are now ss negative (RMN scattering coefficient, CPR absorption coefficient, THD single scattering albedo) or ss positive (PUY single scattering albedo, MSY scattering Ångström exponent, LLN absorption Ångström exponent).
- Some ss trends are now not ss: IZO absorption coefficient,
- One trend (JFJ scattering Ångström exponent for the 20y period) change from ss negative to ss positive trend.
- The statistical significance of some of the 10 y trends of the time evolution analysis (Sect. 3.2) is also modified, but these changes do not impact the results.

The revised manuscript and all tables and figures were corrected in order to take into account the new results.

**Answers to specific comments:**

1. P7, line 35. Why are the trends based on daily medians and not mean? It would have been nice to include a sentence of the choice of aggregation.

   *The mean is the usual averaging method for normal distribution. In case of skewed distribution, the median is used in order to minimize the effect of outliers. The measured in-situ aerosol parameters all have distributions that strongly diverge from normal distribution and can be best fitted by Johnson distribution (Sect 2.5). Most of the calculated parameters are also not normally distributed. They are sometimes almost normally distributed but only for some stations or, e.g., one of the size cuts. To ensure the*

*homogeneity of the results, the median was applied in order to minimize the effects of extremes values.*

*The following sentence was added to the manuscript: "The median was chosen to minimize the effect of extreme values on the average since the measured parameters are strongly not normally distributed and most of the calculated parameters also do not follow a normal distribution".*

2. Page 11. If as stated in line 39 that the TFPW rejection rate is too high, why is the criteria as stated in line 35 significant trend considered only when both PW procedures gave ss?

*The PW rejection rate is very low since this method has a very low type 1 error. PW has however a lower power than the TFPW. If both PW and TFPW detect a ss trend, the low rejection rate (low type 1 error) is ensured by the PW and the trend can be considered as ss. If TFPW leads to ss trends but not PW, the trend is considered as a false positive result as the no-trend hypothesis is rejected by TFPW but accepted by PW. At line 33 (p.11), "PW" was misleading used instead of "prewhitening" so that the reader could not differentiate between the method called PW (von Storch, 1995) and the general used of a prewhitening method that comprises both PW and TFPW.*

*The manuscript was consequently modified and the mention of "false positive" was added at line 38 in order to clarify the explanation : "The standard pre-whitening (PW) by removing the first lag autocorrelation (von Storch, 1995) has a very low type 1 error but also a low test power, whereas the so-called trend-free pre-whitening procedure published by Yue et al. (2002) (called TFPW-Y in Collaud Coen et al., submitted, 2020) restores the test power at the expense of the type 1 error. Both these prewhitening procedures were applied prior to the MK test to assess the statistical significance of the trend. A trend was then considered as ss only if both PW and TFPW-Y were ss at the 95% confidence level or if PW is ss but not TFPW-Y (false negative). Among the trends of all parameters at all stations calculated for this paper, none was ss for the PW but not for the TFPW-Y, meaning that the PW procedure was always powerful enough. In contrast, many trends were not ss when PW was applied, but were ss with the TFPW-Y procedure, leading to false positives and showing that the TFPW_Y rejection rate of the no-trend hypothesis is effectively too high."*

3. Page 11 line 49. What does it actually mean that the seasons are homogeneous?

*Yes, annual trend are only considered is seasons are homogeneous. If $Z_i$ is the Z Mann-Kendall statistical for the i season,*

$$X^2_{total} = \sum_{i=1}^{m} Z_i^2 = X^2_{homogeneous} + X^2_{trend} = X^2_{homogeneous} + m * \bar{Z}^2$$

*where $\bar{Z} = \frac{1}{m} * \sum_{i=1}^{m} Z_i$*

*Then    $X^2_{homogeneous} = \sum_{i=1}^{m} Z_i^2 - m * \bar{Z}^2$*

*Both X statistic have a chi-squared distribution with m-1 degrees of freedom, m being the number of seasons. Their statistical significance can then be assessed (see e.g. Table p.*

*114 of Gilbert, 1987). Trends between seasons are homogeneous if $X^2_{homogeneous}$ is ss. We chose a ss confidence level of 90% for this study (see sect. 2.5.1, p. 12). Since the test for homogeneity between seasons is extensively explained in cited publications and due to the length of the manuscript, the authors does not consider a complete description in the paper as necessary. Gilbert (1987) and Sirois (1998) are however mentioned at p. 12 line 1 in the revised version: "The annual trends were considered only if the slopes of the four seasons were homogeneous at the 90% confidence level (Gilbert, 1987; Sirois, 1998)."*

*Homogeneous trends are mentioned with red squares on Figure 2.*

4. Figure 2. Spelling mistake in caption. RMN should be MRN

   *Both stations (RMN and MRN) exist. Figure 2 relates to MRN and not to RMN.*

5. Page 12. Line 40. GLS/ARB trends for MLO: "the longer periods exhibit ss negative trends". Should read positive trends if to be similar as MK, and it also looks like positive trends in Figure 3.

   *Yes. The text was modified accordingly.*

6. Figure 3. The fitted curves are for the longest periods I assume, (small) different fit for shorter periods?

   *The total fit (in red and orange) corresponds effectively to the longest period whereas the fitted linear trend (in black and grey) are plotted for varying periods. The fitted seasonal cycle is to some extent different for shorter periods, since the amplitude of the cycle is not allowed to change with time. For example, the GLS/ARB fit of the last 10 y with a negative linear trend will better correspond to the original time series minima than the plotted fitted with the positive linear trend for the last 30 y.*

7. Figure 4. ALT is missing on the map. The site is present in Figure 5.

   *ALT is not missing but in light green corresponding to ss negative trend with slope smaller than -0.02. The color is perhaps not well visible so that it was changed to a stronger one.*

8. P 12, L 43-45. If the seasonality fit is better using daily median compared to the GLS/ARB data, why not also use daily data for the LMS analysis. That would make all three methods more comparable to the data used in the MK test?

   *The monthly median for the LMS analysis was first chosen so that the present analysis can be compared to CC2013. The second strong argument concern the presence of negatives: for some stations sampling very low aerosol concentrations, there is a large amount of negative daily medians, particularly for the absorption coefficient. Since the logarithm of the data is taken for the LMS analysis, daily medians would result in a corresponding amount of missing values that concentrate in the season with the lowest aerosol concentration. For some stations and parameters, this can also lead to a distortion of the comparison with the other methods.*

9. P13, 42-44. The paragraph of MLO seems a bit station specific, while the rest of the bullet points represent regions. Maybe add a sentence that this site represents the Pacific as in table 4 and then maybe include CGO?

*The referee remark is pertinent. CGO is now associated to MLO as representative of the Pacific region and, due to the lower amount of stations in the pacific, this bullet point was shifted lower. The present text is:*

*" The only two stations representing the Pregion are MLO and CGO. The recent MLO 10 y $\sigma_{sp}$ trend is ss decreasing, the $\sigma_{sp}$ 15 y trend is not ss, whereas the trends for the longer time periods (20-30 y) are ss positive (see Fig. 2). In the previous decadal trend paper (CC2013), MLO exhibited a ss positive trend for the 10 y period ending in 2010. MLO $\sigma_{sp}$ trends changed then from previously ss positive to nowadays ss negative trends. The recent 10y at CGO is found to be positive and quite homogeneous with the seasons, with fall being the only season without a ss trend."*

10. P14, line 18. GLR does not turn up as ss positive in Figure 12 or 4, though in Table 2

*Thanks for pointing at this incoherence. The GLR scattering coefficient has a period of about 2 years with higher minima in the middle of the time series that cannot be explained by any instrumental change or station history. The removal of this period lead to not ss positive trend, whereas keeping this period lead to ss positive trend. The authors and the station manager finally decided to present the result with the removal of that period, leading unfortunately to these inconstancies between the text, Table 2 and the figures. This is however corrected in the revised version with the correction of Table 2 and the removal of the mention of the GLR ss positive trend at p. 14 line 18.*

11. P14, lin20. "PM10 trends are five times larger than PM1", maybe rewrite to "scattering trends of PM10 aerosols are five times larger than PM1". Do not find separate results for the different size cut off. Could they be included in Table 2 and Figure 12? And maybe indicate which size are chosen for Figure 4 and 5?

*The manuscript was modified:" At CPR, the largest scattering trend is found in summer and the scattering trend of PM10 trend is five times larger than the PM1 trend".*

*As stipulated at Sect. 2.4, the paper presents the TSP/PM10 trends and PM1 results are only presented if they are different from PM10 results. This occurs quite rarely, but CPR is one of this case as mentioned at p. 14 line 20. All figures and tables consequently present TSP/PM10 trends. If the values of the slope are usually lightly different for PM1 trends, an inclusion of these trends for all aerosol parameters in Table 2 and in all figures would contribute to dilute the main results without bringing much more valuable information. The differences between both size cuts are so small that the authors did even not consider showing them in the supplement for clarity purpose.*

12. P14 line 27-31. Do not understand the possible "enhanced" NPF. The references only describes that you may have more NPF at high altitudes but not the possible trends. If the NPF should mask the trend we would expect more NPF in present day than earlier, or?

*Exactly. The NPF are "enhanced" at high altitudes compared to low altitudes NPF rate. This means that the ratio of new particle over the total particles concentration is larger at high altitude and can potentially mask the trends observed at lower altitudes. The manuscript was modified leading hopefully to less confusion:*

*"New particle formation (NPF) and growth are favored at high altitudes (> 1000 m and up to 5000 m) due to low temperatures, high solar radiation and low pre-existing particle concentrations leading to limited condensational sinks for nucleation precursor gases (Sellegri et al., 2019). This higher frequency of nucleation at high altitude leads to a high contribution of secondary particles to the total number concentration that largely contributes to the total scattering coefficient. The decreasing $\sigma_{sp}$ trends from anthropogenic pollution in the planetary boundary layer can, consequently, be masked by the presence of NFP at high altitude stations."*

13. P14 line 43. Not only ALT with positive trend, also NMY.

   *Sorry, this is a typo on Table 2, Figure 6 was right. NMY has a ss tiny negative trend.*

   *Anyhow, the MK trend analysis was recomputed due to the error in the selection in winter season and ALT trend remains positive but not ss. The text, figures and tables were consequently modified.*

14. P15 line 8. WLG show negative trend in Figure 6, Table 2 and Figure 13.

   *You're right. This is an analysis that come from a previous version without 2017-2018 data for WLG. The manuscript was modified:*

   *"In Asia, both the high altitude station of LLN in Taiwan and WLG in China exhibits annual ss decreasing $\sigma_{ap}$ trends. The south Korean coastal station of AMY has no ss annual trend."*

15. P15 line 11-12. NMY (and ALT) show positive trend

   *Similarly to answer at comment 13, there is a typo in Table 2, figure 6 being right. NMY has a ss negative trend. ALT was positive but the recomputed trends lead now to not ss trend for ALT.*

16. Figure 6. Seems like several of the sites miss information (outside circles) of longer trends than the last 10 years. I.e. JFJ, FKL, IPR, UGR etc.

   *The JFJ trend is masked by other stations since I always plot the longest trend first in order not to mask stations with shorter trends. FKL is present but in light green that is not very visible. This color was changed. IPR and UGR have only 10 y trend since their time series are shorter than 15 y.*

17. There are two Figure S7. The first with backscatter trend should maybe be in the paper?

   *No, this is a typo in the supplement, the last figure name was changed to S8 and the text (p. 15 line 27) was also adapted (S7 to S8). Thanks for this observation.*

18. P15, line33. Not sure about "Mostly decreasing". The mean trend is decreasing maybe (Table 4), but only 1 site with ss negative trend (SGP)

   *In North America, there are three ss decreasing trends (BND, SGP and THD), two ss increasing trends and one not ss trend. Anyhow, this is visible in Figure 7 but Table 2 missed the minus sign for BND and THD, which is a mistake. Table 2 was modified.*

19. P16 line 9-10. Why not mentioned that the polar sites ALT and ZEP show positive trends?

*You are right, the positive trends at ALT and ZEP are missing. The manuscript was consequently modified:*

*"The Arctic stations of ALT and ZEP have ss positive $\omega_0$ annual trends, which are due to ss positive trends from March to August for ALT and from December to May for ZEP."*

20. Chapter 3.2. When I read chapter 3.1, I used the Figures developed for 3.2 since they are connected and give a more complete picture. Think the presentation of trend results would have benefited to combine these chapters.

*It was difficult to decide if the present-day trends and the time evolution of the trends should be presented together for each aerosol parameters or treated separately. We decided to treat them separately for clarity purpose, because i) there are some stations, which end their measurements before 2016 leading to no present-day trends and ii) there are not many stations with more than 15 y of measurement (scattering coefficient: 17, absorption coefficient: 5, single scattering albedo: 4) allowing a complete discussion of the trend evolution. We recognize that this choice also has inconveniences but they seem to us, however, not sufficient to merge sections 3.1 and 3.2.*

21. P17 line 36. I don't find the trends in Figure 12 that much scattered, and I am not sure if one can track the differences back to abatement strategies. The UGR site is urban and influence by Saharan dust and is not representative for detecting general trends in Europe. The increase in SMR the latter periods might be do to increase emissions of BVOCs from the boreal forest? The trends in observed and modelled chemical composition in Europe are non linear due to changes in atmospheric chemistry. Maybe refer to some model/observation studies Europe trends for comparison. E.g: Banzhaf, S., Schaap, M., Kranenburg, R., Manders, A. M. M., Segers, A. J., Visschedijk, A. J. H., Denier van der Gon, H. A. C., Kuenen, J. J. P., van Meijgaard, E., van Ulft, L. H., Cofala, J., and Builtjes, P. J. H.: Dynamic model evaluation for secondary inorganic aerosol and its precursors over Europe between 1990 and 2009, Geosci. Model Dev., 8, 1047–1070, https://doi.org/10.5194/gmd-8-1047-2015, 2015.

Ciarelli, G., Theobald, M. R., Vivanco, M. G., Beekmann, M., Aas, W., Andersson, C., Bergström, R., Manders-Groot, A., Couvidat, F., Mircea, M., Tsyro, S., Fagerli, H., Mar, K., Raffort, V., Roustan, Y., Pay, M.-T., Schaap, M., Kranenburg, R., Adani, M., Briganti, G., Cappelletti, A., D'Isidoro, M., Cuvelier, C., Cholakian, A., Bessagnet, B., Wind, P., and Colette, A.: Trends of inorganic and organic aerosols and precursor gases in Europe: insights from the EURODELTA multi-model experiment over the 1990–2010 period, Geosci. Model Dev., 12, 4923–4954, https://doi.org/10.5194/gmd-12-4923-2019, 2019.

*The term "homogeneous" is effectively not correctly chosen. The sentence was modified:*

*"The evolution of the European $\sigma_{sp}$ 10 y trends does not show a clear time for trend modification like in North America, probably due to delays in abatement policies in each individual country."*

*Both publications of Banzhaf and Ciarelli are very relevant to discuss the trends in in-situ aerosol parameters, so that they were introduced in the discussion section 4.2 (see next comment 22) and 4.3:*

*Sect 4.3:" Moreover, emission changes can lead to modification of the atmosphere chemistry. Banzhaf et al. (2015) shows, for example, that the sulfate and nitrate formation have increased in efficiency by factors between 20-25% between 1990 and 2009 leading to lower trends in sulfate and total nitrate concentrations than the trends in precursor emissions and concentrations. "*

22. P22 line 26. Does it have to be anthropogenic sources? Changes in natural sources, typically BVOC may contribute? May a change in atmospheric composition contribute to smaller aerosols, i.e. less sulfate aerosols and more ammonium nitrate which potentially might be smaller?

    *This is a very interesting comment and these criteria were introduced:*

    *"Trends towards smaller particle size might be due to an increase of near anthropogenic sources of pollution, to an increase in new particle formation, to a decrease of long-range transport of anthropogenic pollution, to increased scavenging of larger particles due to changes in atmospheric conditions, to a modification of atmospheric chemistry (Banzhaf et al., 2015) or to a change in both primary and secondary natural aerosol (e.g. an increase of biogenic secondary aerosols and their precursors as demonstrated by Ciarelli et al., 2019)."*

23. P22 line46-Bodhain and Dutton, 1993 is a quite old reference, maybe add e.g. Hand et al 2012 for a longer trend analysis for especially sulfate. Further, one should probably mention that the in Asia the decrease started rapidly after 2013 when the China's Clean Air Action was implemented. Maybe add Paulot et al (2018), which gives a nice global overview of trends using satellites and models

    Hand, J. L., Schichtel, B. A., Malm, W. C., and Pitchford, M. L.: Particulate sulfate ion concentration and SO2 emission trends in the United States from the early 1990s through 2010, Atmos. Chem. Phys., 12, 10353–10365, https://doi.org/10.5194/acp12-10353-2012, 2012.

    Paulot,F.,Paynter,D.,Ginoux,P.,Naik,V.,andHorowitz,L.W.: Changes in the aerosol direct radiative forcing from 2001 to 2015: observational constraints and regional mechanisms, Atmos. Chem. Phys., 18, 13265–13281, https://doi.org/10.5194/acp18-13265-2018, 2018.

    *The old reference of Bodhaine and Dutton was used on purpose to emphasized the fact that the sulphate decrease was recognize since a long time. Hand et al., (2012) was also added as a more recent citation as well as Paulot et al., 2018).*

    *"The beginning of the decrease of the aerosol burden varies with region; the earliest decrease is found in Europe in the 1980's (Tørseth et al., 2012), followed by North America in the 1990's (Bodhaine and Dutton, 1993, Hand et al., 2012) and by Asia some 10-15 years ago (Sogacheva et al., 2019, Zhao et al., 2019, Paulot et al., 2018)."*

24. P23. Line 49 Pandolfi et al (2016) only show PM trends in NE Spain. There are several other national PM trend studies in Europe (e.g. Germany, France, Switzerland). For a complete overview of Europa it is possible to refer to EMEP TFMM assessment report showing PM trends from both EMEP and AIRBASE for 2002-2012: Colette et al 2016: https://projects.nilu.no//ccc/reports/cccr1-2016.pdf (chapter 3.6.1)

*Thanks for this reference. Sect 4.4 was improved by clearly differentiate sub-chapters devoted to comparison with other in-situ trends, comparison with ground-based and air-born remote sensing measurements trends and leading to a more global view, and, finally, the causality for the detected trends that are often based on models. New references were then introduced:*

[revised manuscript text omitted]

---

## Author Comment (AC4) · 22 May 2020

**Answers to anonymous referee 3:**

The authors thank the referee for their detailed review of the manuscript and for all their comments and suggestions allowing a clear improvement of the paper.

During the review process, the routines for MK trend analysis were translated into R and an error was found in the selection of data for north hemispheric winter season. This error was corrected in the original matlab routines leading to minor changes in slope absolute values for most of the stations, but also sometimes to modification of the statistical significance. The more important changes are:

- ALT was the only station with ss trend in absorption coefficient and this was the only case where there is a strong discrepancy among the analysing methods, MK being ss positive, LMS/log not ss and GLS/day ss negative. The correction leads to MK not ss trend in absorption coefficient at ALT and remove therefore the solely strong discrepancy between the methods.
- MLO has a ss negative trend in scattering coefficient for the last 10 y, leading to a better agreement between scattering and absorption trends. The evolution from positive to negative ss trends is now well established.
- Some other not ss present-day trends are now ss negative (RMN scattering coefficient, CPR absorption coefficient, THD single scattering albedo) or ss positive (PUY single scattering albedo, MSY scattering Ångström exponent, LLN absorption Ångström exponent).
- Some ss trends are now not ss: IZO absorption coefficient,
- One trend (JFJ scattering Ångström exponent for the 20y period) change from ss negative to ss positive trend.
- The statistical significance of some of the 10 y trends of the time evolution analysis (Sect. 3.2) is also modified, but these changes do not impact the results.

The revised manuscript and all tables and figures were corrected in order to take into account the new results.

**Answers to specific comments:**

1. P3, L10 and L11: there is a space between - and 0.45 (leading to a newline between)

   *Thank you, the space was removed.*

2. P8, L15: Assuming an Absorption Angström exponent of one for SSA calculation could cause further dependence on changes of size distribution or chemical composition. What is the impact of this assumption?

   *Yes, this assumption can lead to a SSA departing from the true values. Let consider a range of absorption Ångström exponents between -0.5 and -2 and an often encountered ratio between scattering and absorption value of 10. An adjustment from blue (470 nm) to*

*green wavelength (570 nm) would lead to an error of 10% and -18% for $å_{ap}$ =-0.5 and -2, respectively. An adjustment from red (660 nm) to green wavelength (570 nm) would lead to an error of -7% and 16% for $å_{ap}$ =-0.5 and -2, respectively. This will induce a maximum error of ± 1.6 % on the SSA values. A similar difference for the scattering Ångström exponent would lead to a maxiuml error of ± 2 % on the SSA values considering the TSI wavelengths (450, 550 and 700 nm). A combination of maximal error on both $å_{ap}$ and $å_{sp}$ leads to a maximum cumulative error of 6% on SSA. Considering the large errors usually estimated to approximately 30% of the absorption coefficient and of 10-20% on the scattering exponent, the error induced by the 1/λ dependence can be considered as negligible.*

3. P8, L24 – L28: is part of data preparation and thus could be moved to section 2.4

   *This was done in the revised version.*

4. Section 2.4 is missing a paragraph on assessment for nephelometer artefacts

   *The truncation error and the ways it is considered for the various instruments used in this study are described in sect. 2.2, second §. The artifacts bounded to the humidity percentage during the measurement are described in sect. 2.2, third §, in sect. 2.4 (p. 9 lines 45) as well as discussed in sect. 4.1 (p. 21, line 37). The way to handle the wavelength dependence, including the computed parameters, is described in sect. 2.3. A new § was added to Sect. 2.4 in order to describe the potential effects of the truncation correction on the trend analysis:*

   *"4) Nephelometer truncation correction artefacts: as explained in Sect. 2.2, the various types of nephelometer measure at different truncated angular ranges that were corrected by several algorithms or even not corrected. The absence of truncation correction leads to lower scattering and backscattering coefficients than the true values and the correction algorithm effects are known to increase with particle size. The most important requirement that was verified for this trend analysis is the coherent treatment of nephelometer data for each time series. The bias leading to a higher contribution of Aitken and accumulation modes than the coarse mode is difficult to estimate, but the minimal differences in PM1 and PM10 results (see Sect 4.2) suggest this artefact is small. The effect of the humidity on the nephelometer measurements is regarded as the most significant artefact."*

5. P12, L15: do the monthly medians fit the log-normal distribution and what was the procedure to deal with negatives or zero values? What was the reason for median as aggregation method?

   *The monthly median can be considered (at least for part of the time series) as lognormally distributed (see normal probability plot thereafter). None of the values (e.g. negatives, zeros, very low values) were removed before computing the monthly medians. The monthly median aggregation leads to very few negatives that were discarded before taking the logarithm of the data. Aerosol time series do not have zeros. Absorption, scattering and backscattering coefficients have very low values that could be considered as below detection limit values, but no peculiar treatment was applied to very low values.*
   *Since most of the parameters analyzed in this study are not normally distributed, the median was chosen to minimize the effect of extreme values on the average (see sect.*

*2.3 first §). This is the usually recommended method for aggregation in case of not-normally distribution.*

[Figure]

6. P16, L13: could add "Backscatter fraction (b)" for readability

    *This was done in the revised version.*

7. P20, L15 and L17: "derived parameters" would be more specific (instead of "computed parameters")

    *This was modified in the revised version.*

8. P23, L18 and L19: the intention of "Ideally, abatement policy..." is not clear and vague

    *The abatement policy mentioned here concern the governmental regulation to decrease atmospheric pollutants and comprise both gaseous and particle emissions. The manuscript was modified to clarify this point: "Ideally, abatement policy aimed at decreasing atmospheric pollutant levels would take into account both climate and health impacts."*

9. P27, L15: "due because"

*Thanks, "due" was removed.*